# LEARNING IN CIRCLES: ROTATIONAL DYNAMICS IN COMPETITIVE REINFORCEMENT LEARNING

## ABSTRACT

Optimization in competitive reinforcement learning (RL) differs from standard minimization. Actor–critic methods, in single- and multi-agent (MARL) settings, involve coupled objectives, so optimizing them jointly requires finding an equilibrium rather than performing independent descent. Through operator-theoretic viewpoint, we show that actor–critic models inherently exhibit rotational dynamics during learning, cycling around equilibria, thereby explaining in part the instability often observed in practice. Through the variational inequality (VI) framework for studying equilibrium seeking problems, we adopt the Lookahead method for VIs, which suppresses these rotations in actor–critic RL. Building on this, we introduce *Lookahead-(MA)RL (LA-(MA)RL)* to efficiently mitigate rotational dynamics. Across classical two-player games and multi-agent benchmarks, including *Rock–paper–scissors*, *Matching pennies*, and *Multi-Agent Particle environments*, LA-MARL consistently improves convergence and stability. Our results highlight optimization as a critical yet underexplored lever in RL: by rethinking the equilibrium-seeking dynamics, one can achieve substantial stability and performance gains.

## 1 INTRODUCTION

Competitive reinforcement learning (RL)—including single-agent actor–critic methods and multi-agent RL (MARL)—are widely used due to their ability to address complex challenges (see, for example, Omidshafiei et al., 2017; Vinyals et al., 2017; Spica et al., 2018; Zhou et al., 2021; Bertsekas, 2021). However, these methods are considerably harder to train than standard minimization problems. A key difficulty is the *interdependence of parameters*: for example, actors and critics update each other's loss landscapes, so the optimization problem itself changes during training. Numerous algorithmic fixes have been proposed, such as target networks, regularizers, and alternative learning objectives, yet instabilities and reproducibility issues remain, especially in competitive multi-agent settings. Recent reports of large seed-to-seed performance variability in widely used MARL benchmarks highlight the scope of this challenge (Bettini et al., 2024b; Gorsane et al., 2022).

One mechanism rising from this interdependence in competive RL algorihtms is *rotational learning dynamics*, where iterates cycle around equilibria instead of converging (Mescheder et al., 2018; Balduzzi et al., 2018). For instance, the Gradient Descent (GD) method for the $\min_{z_1 \in \mathbb{R}^{d_1}} \max_{z_2 \in \mathbb{R}^{d_2}} z_1 \cdot z_2$ game, rotates around the equilibrium $(0, 0)$ for infinitesimally small learning rates, and diverges away for practical choices of its value. Consequently, GD—and its adaptive variants like *Adam* (Kingma & Ba, 2015)—fail to converge for a broad class of equilibrium-seeking problems.

Figure 1 illustrates this phenomenon in a simple two-player RL setting: under standard gradient descent the joint strategies rotate around the mixed-strategy equilibrium without convergence, while the *Lookahead* (Zhang et al., 2019b; Chavdarova et al., 2021) game optimization method (described in Section 2) contracts the trajectories and reaches equilibrium. Such learning dynamics may not be the only cause of instability in competitive RL, but offer a tractable lens through which we can study their cause.

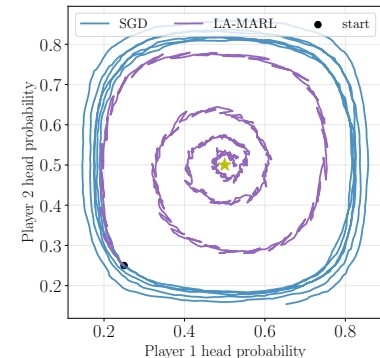

Figure 1: **Rotational dynamics in learning: Matching Pennies.** Joint probabilities of playing *Heads* for both players. $x$-axis: Player 1's probability; $y$-axis: Player 2's. SGD iterates (blue) circle around the mixed-strategy equilibrium (yellow star) without converging, while Lookahead trajectories contract and converge.

Hence, in this work, we ask: *to what extent are these instabilities due to the choice of optimization method, rather than the design of the algorithm itself?* Many RL algorithms implicitly aim to solve an equilibrium problem. Our goal is therefore to keep the high-level algorithm and objectives fixed, but to disentangle the effect of the optimization (learning) dynamics on training outcomes. To do so, we adopt insights from the *variational inequality (VI)* framework, which is a natural mathematical model for equilibrium problems. This motivates two central questions:

*(i) Do competitive RL methods such as actor–critic indeed exhibit rotational or game-like dynamics, and why?*

*(ii) When keeping all other choices fixed, how much can VI-based optimization improve stability and learning outcomes in competitive RL?*

Our objective is to provide systematic evidence that the choice of optimization method itself significantly impacts competitive RL outcomes. MARL serves as an especially relevant setting, since agents not only interact through coupled objectives but also through data collection, making optimization dynamics even more critical. Particularly, MARL problems are modeled with *stochastic games* (Littman, 1994); refer to Section 2. Three main MARL learning paradigms are commonly used:

- *value-based learning*—focuses on estimating so-called *value functions* (e.g., $Q$-learning, Deep $Q$-Networks (Mnih et al., 2015)) to learn action-values first and infer a policy based on it,

- *policy-based learning*—directly optimizes the *policy* (e.g., *REINFORCE (Williams, 1992)*) by adjusting action probabilities without explicitly learning the value functions, and

- *actor-critic methods*—combines value-based and policy-based approaches where an actor selects actions, and a critic evaluates them.

Furthermore, MARL can be broadly categorized into *centralized* and *independent* learning approaches. In centralized MARL, a global critic or shared value function leverages information from all agents to guide learning, improving coordination (Sunehag et al., 2017; Lowe et al., 2017; Yu et al., 2021). In contrast, independent MARL treats each agent as a separate learner, promoting scalability but introducing non-stationarity as agents continuously adapt to each other's evolving policies (Matignon et al., 2012; Foerster et al., 2017). In this work, we focus on *Centralized Training Decentralized Execution* (CTDE) approaches, specifically the ones with centralized critics. Several of the new algorithms in MARL belong to this category such as (MADDPG, Lowe et al., 2017), (MATD3, Ackermann et al., 2019), (MAPPO, Yu et al., 2021), and (COMA, Foerster et al., 2018). In this work, we primarily focus on the CTDE actor-critic MARL learning paradigm, and adopt VI approaches leveraging a (combination of) *nested-Lookahead-VI* (Chavdarova et al., 2021) and *Extragradient* (Korpelevich, 1976) methods for iteratively solving variational inequalities (VIs).

**Contributions.** This paper:

- Analyzes standard actor–critic methods through their Jacobian and shows that, even in a single-agent setting, the joint actor–critic updates generically induce a non-symmetric operator with a rotational (game-like) component, which amplifies in centralized multi-agent extensions. To make this concrete, we introduce a minimal theoretical setting that isolates this phenomenon and validate it with numerical experiments (Section 3).

- Adopts the variational inequality (VI) viewpoint to cast common policy-gradient and centralized actor–critic MARL methods (e.g., MADDPG, MATD3) as explicit VI problems in the joint parameter space, providing concrete operators that match their practical implementations.

- We propose *LA-(MA)RL* (Algorithm 1), a scalable approach for neural network-based agents. While presented for actor-critic systems, the method generalizes to other MARL settings. *LA-MARL* is computationally efficient, making it well-suited for large-scale optimization tasks.

- We evaluate these methods on classic matrix games (rock–paper–scissors, matching pennies) and in benchmark multi-agent particle environments (Lowe et al., 2017), showing improved stability over standard optimization.

As a side contribution, we illustrate the limitations of reward-based performance metrics in competitive MARL and advocate distance-to-equilibrium and cross-play style evaluations as more informative alternatives in these settings.

Code implementation: https://anonymous.4open.science/r/VI-marl-47A4.

**Brief related works discussion.** In mathematics and numerical optimization, equilibrium-finding problems can be modeled using several frameworks, most notably the *Variational Inequality* (VIs, Stampacchia, 1964; Facchinei & Pang, 2003b) framework (see Section 2 for a formal definition). Recent advances in solving variational inequalities (VIs) have been heavily influenced by challenges observed in training generative adversarial networks (GANs, Goodfellow et al., 2014). This progress spans both theoretical developments—with new convergence guarantees (e.g., Golowich et al., 2020b; Daskalakis et al., 2020b; Gorbunov et al., 2022)—and practical algorithms for large-scale optimization (Diakonikolas, 2020; Chavdarova et al., 2021).

A large body of work has studied optimization in actor–critic (AC) and centralized MARL. In the single-agent case, numerous AC algorithms have been proposed (Lillicrap et al., 2015; Schulman et al., 2015; 2017) and their convergence properties analyzed (Mnih et al., 2016; Konda & Tsitsiklis, 1999; Holzleitner et al., 2020; Kumar et al., 2025; Chen & Zhao, 2025). Recent works emphasize their adversarial, game-like structure, connecting AC to GANs (Pfau & Vinyals, 2017) and showing that rotations can be interpreted as Stackelberg dynamics (Zheng et al., 2021). In MARL, research has focused on two-player zero-sum settings with regret guarantees (Bai & Jin, 2020; Xie et al., 2020), as well as extensions to linear quadratic games (Kalman, 1960; Fazel et al., 2018; Bu et al., 2019; Zhang et al., 2021; Mazumdar et al., 2020; Hambly et al., 2023), where convergence remains significantly harder in general-sum cases. Broader formulations such as polymatrix games further expose the limits of gradient descent and motivate alternative algorithms (Ma et al., 2021; Janovskaja, 1968). Despite progress, optimization and convergence in multi-agent actor–critic methods (Bettini et al., 2024a) remain open challenges, with non-stationarity and complex interactions continuing to drive instability.

We provide a detailed related works discussion in Appendix A.

## 2 PRELIMINARIES

**Notation.** We use bold lowercase for vectors, curly capitals for sets, lowercase for real-valued functions, and capitals for operators $\mathcal{Z} \mapsto \mathcal{Z}$ (e.g., $F$). We write $[n] = \{1, \ldots, n\}$ and $\succeq$ for positive semidefinite matrices. Let $\mathcal{Z}$ be a convex, compact subset of Euclidean space equipped with inner product $\langle \cdot, \cdot \rangle$. We follow standard MARL notation in what follows.

**MARL.** *Markov games* (MGs, or *stochastic games*, Shapley, 1953; Littman, 1994) generalize *Markov Decision Processes* (MDPs Puterman, 1994) to a multi-agent setting. An MG is given by:

$$\left(n, \mathcal{S}, \{\mathcal{A}_i\}_{i=1}^n, p, \{r_i\}_{i=1}^n, \gamma\right), \tag{MG}$$

with $n$ agents sharing a state space $\mathcal{S}$. Each agent $i \in [n]$ receives observation $\boldsymbol{o}_i \in \mathcal{O}$ of the current state $\boldsymbol{s} \in \mathcal{S}$ of the environment. In the most general case, agent $i$'s observation $\boldsymbol{o}_i = f(\boldsymbol{s})$, where $f \colon \mathcal{S} \to \mathcal{O}_i$. For instance, $f$ can be an identity or coordinate-selection map with $\mathcal{O}_i \subseteq \mathcal{S}$, or $f$ can be a nontrivial mapping. Based on its policy $\pi_i \colon \mathcal{O}_i \to \mathcal{A}_i$, each agent $i \in [n]$ selects an action $a_i \in \mathcal{A}_i$, where $\mathcal{A}_i$ is its finite action set. The joint actions of all agents are represented as $\boldsymbol{a} \triangleq (a_1, \ldots, a_n)$, and the joint action space as $\mathcal{A} \triangleq \mathcal{A}_1 \times \cdots \times \mathcal{A}_n$.

The environment transitions to a new state $\boldsymbol{s}' \in \mathcal{S}$ according to a *transition function* $p \colon \mathcal{S} \times \mathcal{A} \to \Delta(\mathcal{S})$, where $\Delta(\mathcal{S})$ is the space of probability distributions over $\mathcal{S}$ (non-negative $|\mathcal{S}|$-dimensional vector summing to 1). The function $p$ specifies the probability distribution of the next state $\boldsymbol{s}'$, given the current state $\boldsymbol{s}$ and the joint action $\boldsymbol{a}$. Each agent $i \in [n]$ receives a reward $r_i$, where the reward function $r_i \colon \mathcal{S} \times \mathcal{A} \times \mathcal{S} \to \mathbb{R}$ depends on the current state, the joint action, and the resulting next state. The importance of future rewards is governed by the *discount factor* $\gamma \in [0, 1)$.

MGs generalize MDPs and *repeated games* (Aumann, 1995) by introducing non-stationary dynamics, where agents learn their policies jointly and adaptively. Each agent $i \in [n]$ aims to maximize its expected cumulative reward (return), defined as:

$$v_i^{\pi_i, \boldsymbol{\pi}_{-i}}(\boldsymbol{s}) = \mathbb{E}\left[\sum_{t=0}^{\infty} \gamma^t r_i(\boldsymbol{s}_t, \boldsymbol{a}_t, \boldsymbol{s}_{t+1}) \mid \boldsymbol{s}_0 \sim \boldsymbol{\rho}, \boldsymbol{a}_t \sim \boldsymbol{\pi}(\boldsymbol{s}_t)\right], \tag{MA-Return}$$

with $\boldsymbol{\pi} \triangleq (\pi_1, \ldots, \pi_n)$ the joint policy of all agents, $\boldsymbol{\pi}_{-i}$ the policies of all agents except agent $i$, and $\boldsymbol{\rho}$ the initial state distribution. Agents' interaction introduces non-stationarity (due to evolving policies) and reward interdependencies, leading to a distinct optimization landscape. *Nash equilibria* serve as solution concepts, where no agent can improve its return by unilaterally altering its policy.

**MADDPG.** *Multi-agent deep deterministic policy gradient (MADDPG, Lowe et al., 2017)*, extends *Deep deterministic policy gradient* (DDPG, Lillicrap et al., 2015) to multi-agent setting, leveraging a centralized training with decentralized execution paradigm. Each agent $i \in [n]$ has: (i) *critic* network $Q_i \colon \mathcal{O}_i \times \cdots \times \mathcal{O}_n \times \mathcal{A} \to \mathbb{R}$, parametrized by $\boldsymbol{w}_i \in \mathbb{R}^{d_i^Q}$: acts as a centralized action-value function, evaluating the expected return of joint actions $\boldsymbol{a}$ in state $\boldsymbol{s}$ and (ii) *actor* network $\mu_i \colon \mathcal{O}_i \to \Delta(\mathcal{A}_i)$, parametrized by $\boldsymbol{\theta}_i \in \mathbb{R}^{d_i^\mu}$: represents the agent's policy, mapping agents' observation of states $s$ to a probability distribution over actions $a_i$.

MADDPG uses *target* networks for stability, which are delayed versions of the critic and actor networks: *target critic* $\bar{Q}_i$, is parametrized by $\bar{\boldsymbol{w}}_i \in \mathbb{R}^{d_i^Q}$, and *target-actor* $\bar{\mu}_i$, parametrized by $\bar{\boldsymbol{\theta}}_i$. The target networks are updated using a soft update, with $\tau \in (0, 1]$ as:

$$\bar{\boldsymbol{w}}_i \leftarrow \tau \boldsymbol{w}_i + (1 - \tau)\bar{\boldsymbol{w}}_i, \quad \text{(Target-Critic)} \qquad \bar{\boldsymbol{\theta}}_i \leftarrow \tau \boldsymbol{\theta}_i + (1 - \tau)\bar{\boldsymbol{\theta}}_i. \quad \text{(Target-Actor)}$$

**MATD3.** *Multi-agent TD3* (Ackermann et al., 2019) extends MADDPG with: (i) *dual critics*: each agent uses two critics $Q_{i,1}, Q_{i,2}$ and the smaller target value to reduce overestimation bias; (ii) *delayed actor updates*: policies are updated every $c$ critic steps for stability. Gaussian noise is also added to target actions for exploration. These modifications improve robustness over MADDPG in multi-agent settings.

**Variational Inequality** (Stampacchia, 1964; Facchinei & Pang, 2003b). Variational Inequalities (VIs) extend beyond standard minimization problems to encompass a broad range of equilibrium-seeking problems. The connection to such general problems can be understood from the optimality condition for convex functions: a point $\boldsymbol{z}^\star$ is an optimal solution if and only if $\langle \boldsymbol{z} - \boldsymbol{z}^\star, \nabla f(\boldsymbol{z}^\star) \rangle \geq 0, \forall \boldsymbol{z} \in \text{dom} f$. In the framework of VIs, this condition is generalized by replacing the gradient field $\nabla f$ with a more general vector field $F$, allowing for the modeling of a wider class of problems. Formally, the VI goal is to find an equilibrium $\boldsymbol{z}^\star$ from the domain of continuous strategies $\mathcal{Z}$, such that:
$$\langle \boldsymbol{z} - \boldsymbol{z}^\star, F(\boldsymbol{z}^\star) \rangle \geq 0, \quad \forall \boldsymbol{z} \in \mathcal{Z}, \tag{VI}$$

where $F \colon \mathcal{Z} \to \mathbb{R}^d$, referred to as the *operator*, is continuous, and $\mathcal{Z}$ is a subset of the Euclidean $d$-dimensional space $\mathbb{R}^d$. VIs are thus characterized by the tuple $(F, \mathcal{Z})$, denoted herein as VI($F$, $\mathcal{Z}$). For a more comprehensive introduction to VIs, including examples and applications, see Appendix B.1.

**VI methods.** The *gradient descent* method straightforwardly extends for the VI problem as follows:
$$\boldsymbol{z}_{t+1} = \boldsymbol{z}_t - \eta F(\boldsymbol{z}_t), \tag{GD}$$

where $t$ denotes the iteration count, and $\eta \in (0, 1)$ the step size or learning rate.

*Extragradient* (Korpelevich, 1976) is a modification of GD, which uses a "prediction" step to obtain an extrapolated point $\boldsymbol{z}_{t+\frac{1}{2}}$ using GD: $\boldsymbol{z}_{t+\frac{1}{2}} = \boldsymbol{z}_t - \eta F(\boldsymbol{z}_t)$, and the gradients at the *extrapolated* point are then applied to the *current* iterate $\boldsymbol{z}_t$ as follows:
$$\boldsymbol{z}_{t+1} = \boldsymbol{z}_t - \eta F\left(\boldsymbol{z}_{t+\frac{1}{2}}\right). \tag{EG}$$

Unlike gradient descent, EG converges in some simple game instances, such as in games linear in both players (Korpelevich, 1976).

The *nested-Lookahead-VI* (LA) algorithm for VI problems (Alg. 3, Chavdarova et al., 2021), is a general wrapper of a "base" optimizer $B \colon \mathbb{R}^n \to \mathbb{R}^n$ where, after every $k$ iterations with $B$, $\boldsymbol{z}_{t+1} = B(\boldsymbol{z}_t)$ an averaging step is performed as follows:
$$\boldsymbol{z}_{t+k} \leftarrow \boldsymbol{z}_t + \alpha(\boldsymbol{z}_{t+k} - \boldsymbol{z}_t), \quad \alpha \in [0, 1]. \tag{LA}$$

For this purpose a copy (snapshot) of the iterate after the averaging step is stored for the next LA update. See Appendix B.1.1 for an alternative view.

This averaging can be applied recursively across multiple levels $l$, when using LA as base optimizer, typically with $l \in [1, 3]$. In Algorithm 6, the parameter $k^{(j)}$ at level $j \in [l]$ is defined as the multiple of $k^{(j-1)}$ from the previous level $j - 1$, specifically $k^{(j)} = c_j \cdot k^{(j-1)}$. For $l = 1, k = 2$, LA has connections to EG (Chavdarova et al., 2023), however for higher values of $k$ and $l$ the resulting operator exhibits stronger contraction (Chavdarova et al., 2021; Ha & Kim, 2022), which effectively addresses rotational learning dynamics.

# 3 ACTOR–CRITIC REINFORCEMENT LEARNING AS A GAME: BEYOND STANDARD MINIMIZATION

We begin with a *single-agent discounted MDP* and show that, even in this simplest case, the coupled actor–critic updates form a game in parameter space. This toy model isolates the interaction between actor and critic and will serve as the basis for our multi-agent extension in Section **??**.

**Single-agent setting.** Consider the following setting with one unrolled state $s'$ per trajectory as follows.

**Setting 1** (off-policy deterministic AC setting). *Action $a \in \mathbb{R}$, state $s \in \mathbb{R}^d$. Features $\phi(s, a) = f_\phi(s) + a \, m(s) \in \mathbb{R}^c$ with $m : \mathbb{R}^d \to \mathbb{R}^c$. Critic $Q_{\boldsymbol{w}}(s, a) = \langle \boldsymbol{w}, \phi(s, a) \rangle$, actor $\pi_{\boldsymbol{\theta}}(s) = \langle \boldsymbol{\theta}, s \rangle$, batch $\{(s_i, a_i, r_i, s'_i)\}_{i=1}^{B}$, discount $\gamma \in [0, 1)$. Furthermore, let $\phi_i \triangleq \phi(s_i, a_i)$, $a'_i \triangleq \pi_{\boldsymbol{\theta}}(s'_i)$, $\phi'_i \triangleq \phi(s'_i, a'_i)$, $\delta_i \triangleq \langle \boldsymbol{w}, \phi_i \rangle - r_i - \gamma \langle \boldsymbol{w}, \phi'_i \rangle$.*

**Lemma (informal) 1** (complex eigenvalues generically). *Setting 1 with $\gamma > 0$, has operator $F = (F_{\boldsymbol{w}} ; F_{\boldsymbol{\theta}})^\intercal$ with $F_{\boldsymbol{w}} = \frac{2}{B} \sum_i \delta_i (\phi_i - \gamma \phi'_i)$, $F_{\boldsymbol{\theta}} = -\frac{1}{B} \sum_i s_i \langle \boldsymbol{w}, m(s_i) \rangle$. Its Jacobian has block form $J \equiv (J_{\boldsymbol{ww}}, J_{\boldsymbol{w\theta}} ; J_{\boldsymbol{\theta w}}, \boldsymbol{0})^\intercal$ with:*

$$J_{\boldsymbol{ww}} = \frac{2}{B} \sum_i (\phi_i - \gamma \phi'_i)(\phi_i - \gamma \phi'_i)^\intercal \succeq 0,$$

$$J_{\boldsymbol{w\theta}} = -\frac{2\gamma}{B} \sum_i \left[ (\phi_i - \gamma \phi'_i) \, s'^{\intercal}_i \, \boldsymbol{w}^\intercal m(s'_i) + m(s'_i) \, s'^{\intercal}_i \, \delta_i \right], \ \text{ and } J_{\boldsymbol{\theta}, \boldsymbol{w}} = -\frac{1}{B} \sum_i s_i m(s_i)^\intercal.$$

*Although the underlying environment is a* single-agent *discounted MDP, the learning problem is an equilibrium-seeking problem between two coupled objectives: the critic seeks to minimize $F_{\boldsymbol{w}}(\boldsymbol{w}; \boldsymbol{\theta})$, while the actor seeks to maximize $F_{\boldsymbol{\theta}}(\boldsymbol{w}; \boldsymbol{\theta})$. We therefore interpret the pair $(\boldsymbol{w}, \boldsymbol{\theta})$ as two players in a game in parameter space with joint dynamics governed by $F$. Furthermore, the Jacobian $J$ is typically non-symmetric and, for generic data/parameters, has a non-real conjugate eigenpair (i.e., the linearized dynamics include a rotational component). All eigenvalues are real only under special symmetry/degeneracy (e.g., $\gamma = 0$ or $J_{\boldsymbol{w\theta}} = J_{\boldsymbol{\theta w}}^\intercal$, which in this model requires $\delta_i \equiv 0$ and $\boldsymbol{w}^\intercal m(s'_i) \equiv 0$ for all $i$).*

The above lemma shows setting 1 has *game structure*: except in special symmetry/degeneracy cases (*e.g.*, $\gamma = 0$ or $J_{\boldsymbol{w\theta}} = 0$), the Jacobian is non-symmetric and generically has complex conjugate eigenpair. This is a hallmark of game (rotation component), hence departs from standard minimization where in contrast, $J$—called Hessian in that context—is always symmetric. Full statement and proof deferred to Appendix C.1.

Since the operator of the problem is rotational, running the gradient descent method (GD) produces cyclic trajectories around an equilibrium. Without appropriate learning dynamics, the joint model $\boldsymbol{w}, \boldsymbol{\theta}$ may diverge away. Figure 2 (top) depicts (GD) on a concrete numerical instance of the problem class of setting 1. (GD) does not converge, but rather the joint last iterate $(\boldsymbol{w}, \boldsymbol{\theta})$ cycles around the equilibrium point $(\boldsymbol{w}^\star, \boldsymbol{\theta}^\star)$. Lookahead LA on the other hand, shown in Figure 2 (bottom), increases the contraction (see Appendix D.1) of the base (GD) and converges to the equilibrium. The example game details are given in appendix section C.1.1.

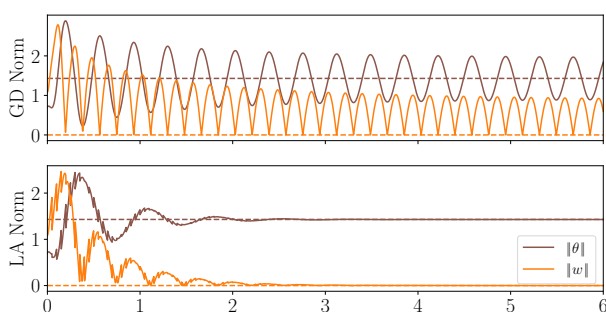

Figure 2: **Rotational learning dynamics**–on Setting 1 instance; see Appendix C.1.1. $x$-axis: training iterations, $y$-axis: $\|\boldsymbol{w}\|$ and $\|\boldsymbol{\theta}\|$ (times $10^3$); the separate norms at equilibrium— $\|\boldsymbol{w}^\star\|, \|\boldsymbol{\theta}^\star\|$—are depicted with dashed lines in orange and brown, resp. *Top*: gradient descent; *bottom*: Lookahead.

**Extension to the multi-agent setting.** With $n$ actors and $n$ centralized critics (each seeing the full state and joint action), the Jacobian keeps the single-agent saddle/block character but now aggregates $n$ actor–critic couplings; the effective coupling (the product of the actor→critic and critic→actor blocks) sums over agents and is generically sign-indefinite, so a non-real conjugate eigenpair persists

and the spectral radius typically *increases* with $n$ (at least matching the strongest single-agent term), while only degenerate triangular/symmetric cases (e.g., $\gamma = 0$ or vanishing cross-couplings) yield an all-real spectrum. Appendix C includes the corresponding lemma and proof for this setting.

While gradient descent might converge in problems with only a mild rotational component, our primary interest is the more competitive settings of (multi-agent) RL. This motivates the following Section 4, where we reinterpret existing MARL methods through the lens of variational inequalities (VIs) and introduce a general algorithmic family that incorporates the insights above.

# 4 COMPETITIVE RL THROUGH VARIATIONAL INEQUALITIES: EXISTING METHODS AND A GENERAL ALGORITHM

In this section, we reinterpret competitive RL methods through the lens of variational inequalities (VIs). We first unify existing approaches—including multi-agent policy gradient and actor–critic methods—under a common operator framework. Since single-agent actor–critic is a special case of the multi-agent setting, our focus is on the latter in the presentation, which captures a broader set of methods. Building on the insights from Section 3, we then introduce a general algorithmic family that augments these operators with Lookahead (LA) and/or Extragradient (EG) updates, applied directly in the joint parameter space of all agents. The pseudocode formulation naturally subsumes single-agent actor–critic and multi-agent policy gradient as special cases.

## 4.1 MARL OPERATORS

**General MARL.** Policy-based learning solves (MA-Return), where agents optimize their policies to maximize return. The associated operator $F_{\text{MAR}}$ (*multi-agent-return*), with $\mathcal{Z} \equiv \mathcal{A}$, is:

$$F_{\text{MAR}}\left(\begin{bmatrix} \vdots \\ \boldsymbol{\pi}_i \\ \vdots \end{bmatrix}\right) \triangleq \begin{bmatrix} \vdots \\ \nabla_{\boldsymbol{\pi}_i} v_i^{\pi_i, \boldsymbol{\pi}_{-i}} \\ \vdots \end{bmatrix} \equiv \begin{bmatrix} \vdots \\ \nabla_{\boldsymbol{\pi}_i}\left(\mathbb{E}\left[\sum_{t=0}^{\infty} \gamma^t r_i(\boldsymbol{s}_t, \boldsymbol{a}_t, \boldsymbol{s}_{t+1}) | \boldsymbol{s}_0 \sim \boldsymbol{\rho}, \boldsymbol{a}_t \sim \boldsymbol{\pi}(\boldsymbol{s}_t)\right]\right) \\ \vdots \end{bmatrix}. \qquad (F_{\text{MAR}})$$

**Actor-critic (MA)RL.** We denote by $\boldsymbol{x}$ the full state representation, from which each agent's observation $\boldsymbol{o}_i$ is derived. Each agent $i \in [n]$ has a centralized critic $\mathbf{Q}_i^{\boldsymbol{\mu}}(\boldsymbol{x}_t, \boldsymbol{a}_t; \boldsymbol{w}_i)$ and a policy $\boldsymbol{\mu}_i(\boldsymbol{o}_i; \boldsymbol{\theta}_i)$. Given a batch $\mathcal{B} = \{(\boldsymbol{x}^j, \boldsymbol{a}^j, \boldsymbol{r}^j, \boldsymbol{x}'^j)\}_{j=1}^{|\mathcal{B}|}$ from the replay buffer $\mathcal{D}$, equilibrium is sought by solving

(VI) with operator:
$$F_{\text{MAAC}}\left(\begin{bmatrix} \vdots \\ \boldsymbol{w}_i \\ \boldsymbol{\theta}_i \\ \vdots \end{bmatrix}\right) \equiv \begin{bmatrix} \vdots \\ \nabla_{\boldsymbol{w}_i}\left(\frac{1}{|\mathcal{B}|} \sum_{j=1}^{|\mathcal{B}|} \ell_i^{\boldsymbol{w}}(\cdot; \boldsymbol{w}_i, \boldsymbol{\theta}_i)\right) \\ \nabla_{\boldsymbol{\theta}_i}\left(\frac{1}{|\mathcal{B}|} \sum_{j=1}^{|\mathcal{B}|} \ell_i^{\boldsymbol{\theta}}(\cdot; \boldsymbol{w}_i, \boldsymbol{\theta}_i)\right) \\ \vdots \end{bmatrix}, \qquad (F_{\text{MAAC}})$$

with parameter space $\mathcal{Z} \equiv \mathbb{R}^d$, $d = \sum_{i=1}^{n}(d_i^Q + d_i^{\mu})$. *MAAC* stands for *multi-agent actor–critic*. Even for $n{=}1$, the actor–critic interaction forms a game: $\boldsymbol{w}_i$ updates depend on $\boldsymbol{\theta}_i$ and vice versa, a coupling that drives the dynamics in MARL, see Section 3.

**MADDPG (Lowe et al., 2017)** As an example, we expand ($F_{\text{MAAC}}$) for MADDPG. The critic and actor losses are:

$$\ell_i^{\boldsymbol{w}}(\cdot; \boldsymbol{w}_i, \boldsymbol{\theta}_i) = \left(y_i - \mathbf{Q}_i^{\boldsymbol{\mu}}(\boldsymbol{x}^j, \boldsymbol{a}^j; \boldsymbol{w}_i)\right)^2, y_i = r_i^j + \gamma \mathbf{Q}_i^{\bar{\boldsymbol{\mu}}}\left(\boldsymbol{x}'^j, \boldsymbol{a}'; \boldsymbol{w}_i'\right)\big|_{\boldsymbol{a}' = \bar{\boldsymbol{\mu}}(\boldsymbol{o}'^j)}. \qquad (\ell_{\text{MADDPG}}^{\boldsymbol{w}})$$

$$\ell_i^{\boldsymbol{\theta}}(\cdot; \boldsymbol{w}_i, \boldsymbol{\theta}_i) = \boldsymbol{\mu}_i(\boldsymbol{o}_i^j; \boldsymbol{\theta}_i) \nabla_{a_i} \mathbf{Q}_i^{\boldsymbol{\mu}}(\boldsymbol{x}^j, a_1^j, \dots, a_i, \dots, a_n^j; \boldsymbol{w}_i)\big|_{a_i = \boldsymbol{\mu}_i(\boldsymbol{o}_i^j)}. \qquad (\ell_{\text{MADDPG}}^{\boldsymbol{\theta}})$$

Here, the critic minimizes the Bellman error while the actor maximizes the $Q$-value, highlighting their coupled dynamics. Operators for other algorithms are deferred to Appendix D.

## 4.2 LA-(MA)RL PSEUDOCODE

To solve the VI problem with MARL operators such as ($F_{\text{MAR}}$) or ($F_{\text{MAAC}}$), we propose two methods: *LA-MARL* and *EG-MARL*.

---

**Algorithm 1** LA–(MA)RL Pseudocode

---

1: **Input:** Env $\mathcal{E}$; agents $n$; episodes $T$; actor nets $\{\mu_i\}_{i=1}^n$ (params $\boldsymbol{\theta}$); critic nets $\{Q_i\}_{i=1}^n$ (params $\boldsymbol{w}$); targets $\{\bar{\mu}_i, \bar{Q}_i\}$; VI operator $F$; base optimizer $B$; discount $\gamma$; soft update $\tau$; random steps $t_{\text{rand}}$; learn interval $t_{\text{learn}}$; LA hyperparams $\mathcal{L} = (l, \{k^{(j)}\}, \alpha_{\boldsymbol{\theta}}, \alpha_{\boldsymbol{w}})$.
2: **Init:** Replay buffer $\mathcal{D} \leftarrow \varnothing$; set targets $\bar{\theta} \leftarrow \theta, \bar{w} \leftarrow w$; LA state $\boldsymbol{\phi}$.
3: **for** $e = 1$ **to** $T$ **do**
4:   reset state $\boldsymbol{x}$ from $\mathcal{E}$.
5:   **while** episode not done **do**
6:     **Act:** for each agent $i$, choose $a_i \sim \text{Uniform}(\mathcal{A}_i)$ if step $\leqslant t_{\text{rand}}$, else $a_i = \mu_i(o_i)$.
7:     **Step:** execute $\boldsymbol{a}$, observe $(\mathbf{r}, \boldsymbol{x}')$; store $(\boldsymbol{x}, \boldsymbol{a}, \mathbf{r}, \boldsymbol{x}')$ in $\mathcal{D}$; set $\boldsymbol{x} \leftarrow \boldsymbol{x}'$.
8:     **if** step mod $t_{\text{learn}} = 0$ **and** $\mathcal{D}$ sufficiently full **then**
9:       **Learn:** sample batch $\mathcal{B} \subset \mathcal{D}$.
10:        $(\boldsymbol{\theta}, \boldsymbol{w}) \leftarrow \text{VI-UPDATE}(F, \mathcal{B}, \boldsymbol{\theta}, \boldsymbol{w}; B, \gamma)$
11:        **Targets:** $\bar{\boldsymbol{\theta}} \leftarrow \tau\boldsymbol{\theta} + (1-\tau)\bar{\boldsymbol{\theta}}, \ \bar{\boldsymbol{w}} \leftarrow \tau\boldsymbol{w} + (1-\tau)\bar{\boldsymbol{w}}$.
12:     **end if**
13:   **end while**
14:   **Lookahead:** $(\boldsymbol{\theta}, \boldsymbol{w}) \leftarrow \text{NESTEDLOOKAHEAD}((\boldsymbol{\theta}, \boldsymbol{w}), \mathcal{L})$.
15: **end for**
16: **Output:** $\boldsymbol{\theta}^{(l-1)}, \boldsymbol{w}^{(l-1)}$.

---

**LA-MARL, Algorithm 1.** LA-MARL periodically (at $step = k$) saves snapshots of all agents' actor and critic networks and averages them with the current parameters, using $\alpha$-averaging (Algorithm 6). It can be nested at multiple levels, where higher levels update less frequently. All agents run the LA step synchronously to ensure contraction. Variants of the algorithm specialized for MADDPG and MATD3 are provided in the appendix (Algorithms 7, 8).

*Adaptability.* Although designed for off-policy actor–critic, LA-(MA)RL extends naturally to other MARL settings: (i) policy gradient methods, by removing the critic; and (ii) on-policy methods, by omitting target networks. Importantly, lookahead must be applied in the joint parameter space, since adversarial objectives induce rotational dynamics, and per-agent averaging would be inconsistent.

**(LA-)EG-MARL.** EG-MARL applies the extragradient update rule (EG) to all agents' actor and critic networks (Algorithm 2). LA-EG-MARL further combines extragradient with nested lookahead by using (EG) as the base optimizer.

**Convergence.** For monotone operators (see Appendix B.1.1 for the definition), gradient descent is known to diverge (Korpelevich, 1976). In contrast, EG-MARL and its LA variants converge under this assumption (Korpelevich, 1976; Chavdarova et al., 2021; Gorbunov et al., 2022; Pethick et al., 2023). Lookahead improves the contractiveness of the fixed-point operator, and nesting further strengthens stability, mitigating divergence in competitive (rotational) settings. See Appendix D for details.

## 5 EXPERIMENTS

### 5.1 SETUP

We build on the open-source *PyTorch* MADDPG implementation and extend it to MATD3, using the original papers' hyperparameters (Appendix E). Experiments span two zero-sum games—rock–paper–scissors and matching pennies—and a competitive MPE scenario, using *PettingZoo* (Terry et al., 2021) implementations.

**Rock–paper–scissors (RPS) and Matching Pennies (MP).** These canonical games have closed-form mixed Nash equilibria (MNE) and expose inherent cycling (Zhou, 2015; Wang et al., 2014; Srinivasan et al., 2018) In RPS, two players ($n = 2$) choose among three actions ($m = 3$) with MNE $\pi_{\text{RPS}}^\star(\frac{1}{3}, \frac{1}{3}, \frac{1}{3})$; players observe the opponent's previous action and receive $+1/0/-1$ for win/tie/loss

---

https://github.com/Git-123-Hub/maddpg-pettingzoo-pytorch/tree/master

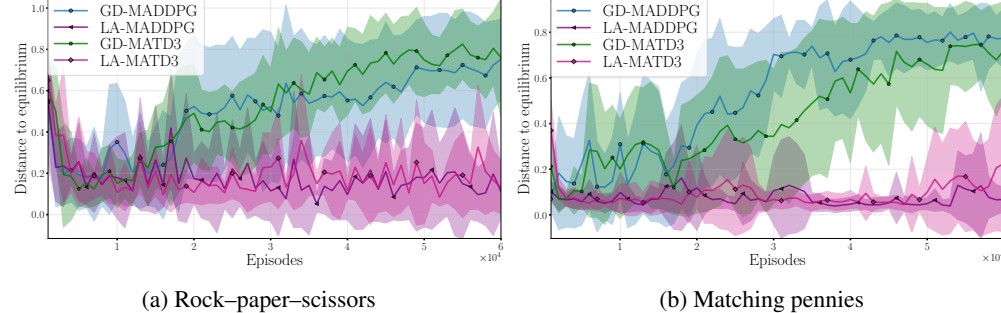

(a) Rock–paper–scissors

(b) Matching pennies

Figure 3: **Comparison between *GD-(MADDPG/MATD3)* and *LA-(MADDPG/MATD3)*, on Rock–paper–scissors and Matching pennies.** $x$-axis: training episodes. $y$-axis:shows the distance between the agents' policies and the equilibrium policy. Curves are averaged over 10 random seeds, and the shaded regions show $\pm 1$ standard deviation over these 10 seeds.

| #Players | GD vs. GD | GD vs. LA | LA vs. LA | LA vs. GD |
|---|---|---|---|---|
| $n = 3$ | $2.99 \pm 1.73$ | $2.14 \pm .91 \downarrow$ | $5.44 \pm 1.27$ | $7.41 \pm 1.75 \uparrow$ |
| $n = 5$ | $15.69 \pm 7.18$ | $15.5 \pm 5.32 \downarrow$ | $14.58 \pm 5.45$ | $22.58 \pm 8.97 \uparrow$ |

Table 1: **Competition between agents trained with different algorithms.** Means and standard deviations (over 5 seeds) of adversary reward in MPE: Predator Prey, on 100 test environments. When GD's opponent is switched to LA, its reward decreases, and vice versa. See Section 5.2.

over $t = 25$ steps. MP is a two-player, two-action game ($m = 2$) with MNE $\pi^{\star}_{\mathrm{MP}} = (\frac{1}{2}, \frac{1}{2})$; *Even* wins on a match, *Odd* on a mismatch. We report $\|\pi - \pi^{\star}\|^2$ for learned policies in both games.

**MPE: Predator-Prey and Physical Deception.** We evaluate one competitive environment from the Multi-Agent Particle Environments (MPE) benchmark (Lowe et al., 2017). *Predator-Prey*, has $p$ *good* agents, $m$ *adversaries*, and $l$ landmarks, where good agents are faster but penalized for being caught or going out of bounds, while adversaries collaborate to capture them. We use $n \equiv (p + m) \in [3, 5]$, and $l = 2$.

**Methods.** We evaluate our methods against the baseline, GD-MARL (GD): MADDPG/MATD3 with Adam (Kingma & Ba, 2015) as base optimizer $B$. When referring to LA-based methods, we will indicate the $k$ values for each lookahead level in brackets. For instance, LA (10, 1000) denotes a two-level lookahead where $k^{(1)} = 10$ and $k^{(2)} = 1000$. *EG* denotes the EG method, and refer to it analogously. Further details on hyperparameters are provided in Appendix E.

### 5.2 RESULTS

**2-player games: RPS & MP.** Figures 3a–3b plot verage distance between learned and equilibrium policies. We show that GD-MARL drifts/diverges, while LA-MARL reliably contracts to near-equilibrium and outperforms GD. MADDPG and MATD3 perform similarly, with MATD3 showing lower seed variance. Figure 1 shows how Lookahead dampens rotations in MP, reducing the distance to equilibrium relative to GD.

**MPE: Predator–Prey.** Table 1 reports mean adversary rewards versus good agents. We train GD-MATD3 (baseline) and LA-MATD3 with five seeds and cross-play them. The results show that LA agents perform well against LA and consistently outperform GD in head-to-head matchups.

**Summary.** Overall, our results indicate the following: *(i)* VI-based methods consistently outperform their respective baselines, by effectively handling the rotational dynamics. *(ii)* LA-VI outperforms the other methods.

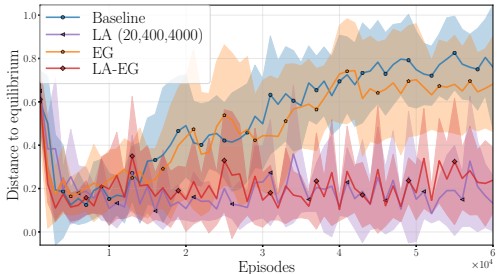

Figure 4: **Comparison between *GD*, *LA*, *EG*, and *LA-EG* optimization methods on the Rock–paper–scissors game.** $x$-axis: training episodes. $y$-axis: squared norm of the learned policy probabilities relative to the equilibrium. Shaded regions indicate one standard deviation across the random seeds used. *EG* uses solely one extrapolation, and thus, as a method, is very close to *GD*; refer to Section 5.2 for a discussion.

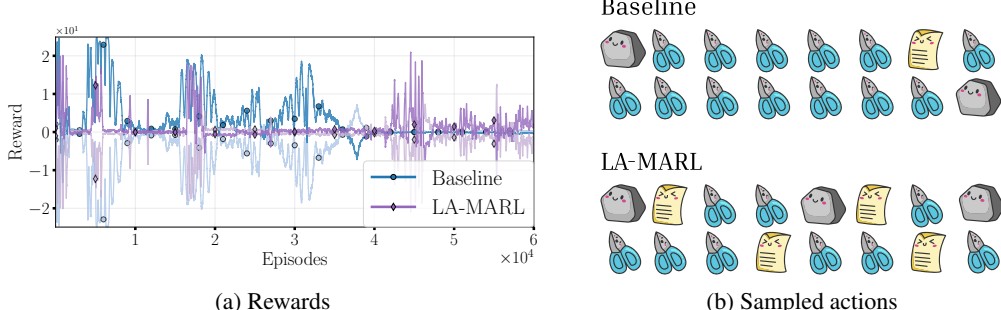

(a) Rewards  (b) Sampled actions

Figure 5: **Rewards (left) vs. sampled actions from learned policies (right), of (LA-)MADDPG in the Rock–paper–scissors game.** The baseline has saturating rewards (in the last part), however, that is not indicative of the agents' performances. Refer to Section 5.2 for a discussion, and Figure 16 for more detailed plots and larger action samples.

**Comparison among VI methods & insights from GANs.** EG ((EG))—while convergent for mono-tone VIs—yields only marginal gains over GD here, reflecting its small local correction. In contrast, Lookahead (LA) imposes stronger contraction, markedly improving stability and convergence; deeper nesting helps—especially preventing last-iterate divergence (Fig. 6)—but too many levels slow learn-ing. Three nested levels offered the best trade-off (Fig. 4). This mirrors GAN results (Chavdarova et al., 2021): EG gives modest benefits, whereas more contractive VI updates consistently perform better. Overall, the evidence indicates strong rotational dynamics; for highly competitive tasks, prefer more contractive VI methods (e.g., multi-level LA).

**Limitations of Rewards as a Metric in MARL.** While saturating rewards are commonly used in MARL, few works challenge their reliability (e.g., Bowling, 2004) as reward convergence doesn't equal optimality. Rewards can saturate with suboptimal policies—e.g., in Fig. 5, baseline agents tie by repeating a subset of actions, masking failure to reach equilibrium. Conversely, LA-MADDPG shows no saturation yet learns near-equilibrium via randomization. For this, stronger metrics are needed, especially when the true equilibrium is unknown (Appendix F.9).

## 6 DISCUSSION

This work studied competitive reinforcement learning through the lens of variational inequalities (VIs). Through second derivative analysis, we showed why even single-agent actor–critic methods inherit game-like dynamics with rotational components, and that these effects amplify in multi-agent settings. Building on this operator view, we unified common multi-agent policy-gradient and actor–critic methods under a VI formulation and instantiated VI-inspired optimizers as *drop-in* updates. In particular, we introduced a general algorithmic framework (Algorithm 1) that augments existing MARL operators with Lookahead (LA) and/or Extragradient (EG) steps, yielding computationally efficient methods tailored to practical CTDE MARL. We provided consistent empirical evidence that integrating VI-based methods improves stability and last-iterate behavior. Conceptually, our LA-MARL and EG-MARL variants act as lightweight, general wrappers that transfer tools from VI theory into practical actor–critic and multi-agent policy-gradient frameworks.

**Limitations.** While rotations offer a useful and tractable lens on instability, they are not the sole driver of failure modes in RL. Function-approximation error, exploration strategies, and environment stochasticity all interact with the rotational dynamics we study. Understanding this interplay more systematically, and isolating when rotational effects dominate versus when other factors are critical, is an important direction for follow-up work.

**Future directions.** Our results suggest several additional promising avenues: further developing VI-inspired optimizers tailored specifically to stochastic and off-policy settings; extending analysis beyond centralized critics to decentralized settings; and designing new evaluation protocols that go beyond reward-based metrics to better capture model performance. This work highlights the importance of optimization dynamics in competitive RL and aims to bridge VI theory and deep MARL practice, opening up further research at the intersection of game theory, optimization, and reinforcement learning.

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

## A    EXTENDED RELATED WORKS DISCUSSION

Our work is primarily grounded in three key areas: Actor critic (AC) methods, Multi-Agent Reinforcement Learning (MARL) and Variational Inequalities (VIs), which we discuss next. Additionally, we extend our discussion on related works on Linear-Quadratic (LQ) games and discuss relevant literature on independent MARL.

**Actor critic.** Many AC algorithms have been proposed for single-agent reinforcement learning (e.g., Lillicrap et al., 2015; Schulman et al., 2015; 2017). In this work, we focus on the rotational, game-like dynamics that can arise in AC. Pfau & Vinyals (2017) formalize the connection between AC and GANs as a bilevel optimization problem, highlighting the adversarial structure. Zheng et al. (2021) make the cycling phenomenon explicit by casting actor–critic updates as a two-player Stackelberg game and show that leader–follower updates can mitigate such rotations. Actor–critic (AC) convergence is well understood in classic two–time-scale settings with linear function approximation (critic faster than actor), where the iterates converge to a stationary point under standard conditions (Konda & Tsitsiklis, 1999). More recent results push beyond linearity and tabular assumptions: Holzleitner et al. (2020) prove (local) convergence of deep actor–critic schemes applied to PPO and RUDDER using a two–time-scale stochastic-approximation framework that accommodates episodic sampling and policies that become greedier over time. On the single-time-scale front, Kumar et al. (2025) analyze the coupled actor/critic recursions and establish global convergence to an $\epsilon$-optimal policy with sample complexity) (finite-state discounted MDPs), refining prior bounds and specifying step-size schedules for both actor and critic. In continuous state–action spaces, Chen & Zhao (2025) give a finite-time convergence analysis for the canonical single-time-scale AC with Markovian sampling via a unified Lyapunov framework, closing a gap between practice and theory. Complementing these, Tian et al. (2023) prove convergence guarantees for AC with deep multi-layer neural networks, quantifying how network width and critic approximation quality control the stationarity gap.

**Multi-Agent Reinforcement Learning (MARL).**    Various MARL algorithms have been developed (Lowe et al., 2017; Iqbal & Sha, 2018; Ackermann et al., 2019; Yu et al., 2021), with some extending existing single-agent reinforcement learning (RL) methods (Rashid et al., 2018; Son et al., 2019; Yu et al., 2022; Kuba et al., 2022). Lowe et al. (2017) extend an actor-critic algorithm to the MARL setting using the *centralized training decentralized execution* framework. In the proposed algorithm, named *multi-agent deep deterministic policy gradient (MADDPG)*, each agent in the game consists of two components: an *actor* and a *critic*. The actor is a policy network that has access only to the local observations of the corresponding agent and is trained to output appropriate actions. The critic is a value network that receives additional information about the policies of other agents and learns to output the Q-value; see Section 2. After a phase of experience collection, a batch is sampled from a replay buffer and used for training the agents. To our knowledge, all deep MARL implementations rely on either stochastic gradient descent or *Adam* optimizer (Kingma & Ba, 2015) to train all networks. Game theory and MARL share many foundational concepts, and several studies explore the relationships between the two fields (Yang & Wang, 2021; Fan, 2024), with some using game-theoretic approaches to model MARL problems (Zheng et al., 2021). This work proposes incorporating game-theoretic techniques into the optimization process of existing MARL methods to determine if these techniques can enhance MARL optimization.

Li et al. (2019) introduced an algorithm called *M3DDPG*, aimed at enhancing the robustness of learned policies. Specifically, it focuses on making policies resilient to worst-case adversarial perturbations, as well as uncertainties in the environment or the behaviors of other agents. Several works rely on two-player zero-sum Markov games to study the regret of an agent relative to a perfect adversary. For instance, Bai & Jin (2020) introduces self-play algorithms for online learning—the *Value Iteration with Upper/Lower Confidence Bound* (VI-ULCB) and an explore-then-exploit algorithm—and show the respective regret bounds. In addition to the online setting, Xie et al. (2020) also consider the offline setting where they propose using Coarse Correlated Equilibria (CCE) instead of Nash Equilibrium (NE) and derive concentration bounds for CCEs. Convergence in MARL is challenging due to complex interactions and non-stationarity among agents. While multi-agent actor-critic methods are widely used (Bettini et al., 2024a), their optimization and convergence properties remain underexplored, making this an open problem.

**Variational Inequalities (VIs).** VIs were first formulated to understand the equilibrium of a dynamical system (Stampacchia, 1964). Since then, they have been studied extensively in mathematics, including operational research and network games (see Facchinei & Pang, 2003b, and references therein). More recently, after the shown training difficulties of GANs (Goodfellow et al., 2014)—which are an instance of VIs—an extensive line of works in machine learning studies the convergence of iterative gradient-based methods to solve VIs numerically. Since the last and average iterates can be far apart when solving VIs (see e.g., Chavdarova et al., 2019), these works primarily aimed at obtaining last-iterate convergence for special cases of VIs that are important in applications, including bilinear or strongly monotone games (e.g., Tseng, 1995; Malitsky, 2015; Facchinei & Pang, 2003b; Daskalakis et al., 2018; Liang & Stokes, 2019; Gidel et al., 2019; Azizian et al., 2020; Thekumparampil et al., 2022), VIs with cocoercive operators (Diakonikolas, 2020), or monotone operators (Chavdarova et al., 2023; Gorbunov et al., 2022). Several works *(i)* exploit continuous-time analyses (Ryu et al., 2019; Bot et al., 2020; Rosca et al., 2021; Chavdarova et al., 2023; Bot et al., 2022), *(ii)* establish lower bounds for some VI classes (e.g., Golowich et al., 2020b;a), and *(iii)* study the constrained setting (Daskalakis & Panageas, 2019; Cai et al., 2022; Yang et al., 2023; Chavdarova et al., 2024), among other. Due to the computational complexities involved in training neural networks, iterative methods that rely solely on first-order derivative computation are the most commonly used approaches for solving variational inequalities (VIs). However, standard gradient descent and its momentum-based variants often fail to converge even on simple instances of VIs. As a result, several alternative methods have been developed to address this issue. Some of the most popular first-order methods for solving VIs include the *extragradient* method (Korpelevich, 1976), *optimistic gradient* method (Popov, 1980), *Halpern* method (Diakonikolas, 2020), and (nested) *Lookahead-VI* method (Chavdarova et al., 2021); these are discussed in detail in Section 2 and Appendix B.1.1. In this work, we primarily focus on the nested Lookahead-VI (LA) method, which has achieved state-of-the-art results on the CIFAR-10 (Krizhevsky, 2009) benchmark for generative adversarial networks (Goodfellow et al., 2014).

**VIs and Markov games.** Variational inequalities (VIs) offer a convenient framework for expressing equilibrium-seeking problems in games and multi-agent learning. Recently, this perspective has been used to analyze learning dynamics in multi-player Markov games. In that line of work, Anagnostides et al. (2023) focus on single-controller Markov games and design a new optimistic policy-gradient algorithm that converges to stationary $\epsilon$-Nash equilibria under a generalized Minty condition (Facchinei & Pang, 2003a). Similarly, Kalogiannis et al. (2024) study two-player adversarial team Markov games, reformulating equilibrium computation as a nonconvex–concave min–max problem and exploiting hidden monotone-operator structure to obtain algorithms to compute the Nash equilibrium in a Markov game with polynomial complexity. In both cases, the main contribution is *algorithmic*: they introduce new learning rules tailored to specific structured classes of Markov games and analyze their convergence. By contrast, our work keeps standard CTDE actor–critic algorithms (e.g., MADDPG, MATD3) fixed and instead changes only the *optimization layer*, using the VI framework to *(i)* reveal and quantify rotational dynamics in existing actor-critic updates and *(ii)* plug in VI-inspired optimizers such as Lookahead and Extragradient as drop-in replacements, thereby isolating the impact of optimization dynamics on stability and convergence behavior.

Another line of work studies simple and optimal methods for stochastic VIs under Markovian noise, with reinforcement-learning policy evaluation as a motivating application. Kotsalis et al. (2021) design several TD-like algorithms—one similar to an extragradient type—for solving VIs with inexact information and under the existence of a stochastic oracle and provide non-asymptotic convergence rates in this setting. In contrast, we work in practical deep multi-agent reinforcement learning setup: we cast practical actor–critic and centralized MARL methods as VI problems in joint parameter space and use VI-based optimization as a generic wrapper to improve their empirical behavior, rather than proposing new TD schemes for policy evaluation.

**General-sum linear quadratic (LQ) games.** In LQ games, each agent's action linearly impacts the state process, and their goal is to minimize a quadratic cost function dependent on the state and control actions of both themselves and their opponents. LQ games are widely studied as they admit global Nash equilibria (NE), which can be analytically computed using coupled algebraic Riccati equations (Lancaster & Rodman, 1995).

Several works establish global convergence for policy gradient methods in zero-sum settings. Zhang et al. (2019a) propose an alternating policy update with projection for deterministic infinite-horizon settings, proving sublinear convergence. Bu et al. (2019) study leader-follower policy gradient in a deterministic setup, and showing sublinear convergence. Zhang et al. (2021) study the sample complexity of policy gradient with alternating policy updates.

For the deterministic $n$-agent setting, Mazumdar et al. (2020) showed that policy gradient methods fail to guarantee even local convergence. Roudneshin et al. (2020) prove global convergence for policy gradient in a *mean-field* LQ game with infinite horizon and stochastic dynamics. Hambly et al. (2023) show that the *natural policy gradient* method achieves global convergence in finite-horizon general-sum LQ games, provided that a certain condition on an added noise to the system is satisfied. Recently, Guan et al. (2024) proposed a policy iteration method for the infinite horizon setting.

**Independent MARL.** In independent MARL, each agent learns its policy independently, without direct access to the observations, actions, or rewards of other agents (Matignon et al., 2012; Foerster et al., 2017). Each agent treats the environment as stationary and ignores the presence of other agents, effectively treating them as part of the environment.

(Daskalakis et al., 2020a) study two-agent zero-sum MARL setting of independent learning algorithms. The authors show that if both players run policy gradient methods jointly, their policies will converge to a min-max equilibrium of the game, as long as their learning rates follow a two-timescale rule. (Arslan & Yüksel, 2015) propose a decentralized $Q$-learning algorithm for MARL setting where agents have limited information and access solely of their local observations and rewards. Jiang & Lu (2022) proposes a decentralized algorithm. Sayin et al. (2021) explore a decentralized $Q$-learning algorithm for zero-sum Markov games, where two competing agents learn optimal policies without direct coordination or knowledge of each other's strategies. Each agent relies solely on local observations and rewards, updating their $Q$-values independently while interacting in a stochastic environment. (Lu et al., 2021) study a decentralized cooperative multi-agent setting with coupled safety constraints.

Wei et al. (2017) rely on the framework of average-reward stochastic games to model single player with a perfect adversary, yielding a two-player zero-sum game, in a Markov environment, and study the regret bound.

## B EXTENDED BACKGROUND

In this section, we further discuss VIs, and provide additional background and relevant algorithms.

### B.1 VI DISCUSSION

**Variational Inequality.** We first recall the definition of VIs. A **VI**$(F, \mathcal{Z})$ is defined as:

$$\text{find } \boldsymbol{z}^\star \in \mathcal{Z} \quad \text{s.t.} \quad \langle \boldsymbol{z} - \boldsymbol{z}^\star, F(\boldsymbol{z}^\star) \rangle \geq 0, \quad \forall \boldsymbol{z} \in \mathcal{Z}, \tag{VI}$$

where $F \colon \mathcal{Z} \to \mathbb{R}^d$, referred to as the *operator*, is continuous, and $\mathcal{Z}$ is a subset of the Euclidean $d$-dimensional space $\mathbb{R}^d$.

When $F \equiv \nabla f$ and $f$ is a real-valued function $f \colon \mathcal{Z} \to \mathbb{R}$, the problem VI is equivalent to standard minimization. However, by allowing $F$ to be a more general vector field, VIs also model problems such as finding equilibria in zero-sum and general-sum games (Cottle & Dantzig, 1968; Rockafellar, 1970). We refer the reader to (Facchinei & Pang, 2003b) for an introduction and examples.

To illustrate the relevance of VIs to multi-agent problems, consider the following example. Suppose we have $n$ agents, each with a strategy $\boldsymbol{z}_i \in \mathbb{R}^{d_i}$, and let us denote the joint strategy with

$$\boldsymbol{z} \equiv \begin{bmatrix} \boldsymbol{z}_1 \\ \vdots \\ \boldsymbol{z}_n \end{bmatrix} \in \mathbb{R}^d, \qquad \text{with} \quad d = \sum_{i=1}^{n} d_i.$$

Each agent $i \in [n]$ aims to optimize its objective $f_i \colon \mathbb{R}^d \to \mathbb{R}$, which, in the general case, depends on all players' strategies. Then, finding an equilibrium in this game is equivalent to solving a VI where the operator $F$ corresponds to:

$$F_{n\text{-agents}}(\boldsymbol{z}) \equiv \begin{bmatrix} \nabla_{\boldsymbol{z}_1} f_1(\boldsymbol{z}) \\ \vdots \\ \nabla_{\boldsymbol{z}_n} f_n(\boldsymbol{z}) \end{bmatrix}. \tag{$F_{n\text{-agents}}$}$$

An instructive way to understand the difference between non-rotational and rotational learning dynamics is to consider the second-derivative matrix $J \colon R^d \to \mathbb{R}^{d \times d}$ referred herein as the *Jacobian*. For the above ($F_{n\text{-agents}}$) problem the Jacobian is as follows:

$$J_{n\text{-agents}}(\boldsymbol{z}) \equiv \begin{bmatrix} \nabla^2_{\boldsymbol{z}_1^2} f_1(\boldsymbol{z}) & \nabla^2_{\boldsymbol{z}_1 \boldsymbol{z}_2} f_1(\boldsymbol{z}) & \dots & \nabla^2_{\boldsymbol{z}_1 \boldsymbol{z}_n} f_1(\boldsymbol{z}) \\ \vdots & \vdots & \dots & \vdots \\ \nabla^2_{\boldsymbol{z}_n \boldsymbol{z}_1} f_n(\boldsymbol{z}) & \nabla^2_{\boldsymbol{z}_n \boldsymbol{z}_2} f_n(\boldsymbol{z}) & \dots & \nabla^2_{\boldsymbol{z}_n^2} f_n(\boldsymbol{z}) \end{bmatrix}. \tag{$J_{n\text{-agents}}$}$$

Notably, unlike in minimization, where the second-derivative matrix—the so-called *Hessian*—is always symmetric, the Jacobian is not necessarily symmetric. Hence, its eigenvalues may belong to the complex plane. In some cases, the Jacobian of the associated vector field can be decomposed into a symmetric and antisymmetric component (Balduzzi et al., 2018), where each behaves as a *potential* (Monderer & Shapley, 1996) and a *Hamiltonian* (purely rotational) game, resp.

In Section D we will also rely on a more general problem, referred to as the *Quasi Variational Inequality*.

**Quasi Variational Inequality.** Given a map $F \colon \mathcal{X} \to \mathbb{R}^n$—herein referred as an *operator*—the goal is to:

$$\text{find } \boldsymbol{x}^\star \quad \text{s.t.} \quad \langle \boldsymbol{x} - \boldsymbol{x}^\star, F(\boldsymbol{x}^\star) \rangle \geq 0, \quad \forall \boldsymbol{x} \in \mathcal{K}(\boldsymbol{x}^\star), \tag{QVI}$$

where $\mathcal{K}(\boldsymbol{x}) \subseteq \mathbb{R}^d$ is a point-to-set mapping from $\mathbb{R}^d$ into subsets of $\mathbb{R}^d$ such that for every $\boldsymbol{x} \in \mathcal{X}$, $\mathcal{K}(\boldsymbol{x}) \subseteq \mathbb{R}^d$ which can be possibly empty.

In other words, the constraint set for QVIs depends on the variable $\boldsymbol{x}$. This contrasts with a standard variational inequality (VI), where the constraint set $\mathcal{K}$ is fixed and does not depend on $\boldsymbol{x}$. QVIs were introduced in a series of works by Bensoussan & Lions (1973a;b; 1974).

### B.1.1 VI CLASSES AND ADDITIONAL METHODS

The following VI class is often referred to as the generalized class for VIs to that of convexity in minimization.

**Definition 1** (monotonicity). An operator $F : \mathbb{R}^d \to \mathbb{R}^d$ is *monotone* if $\langle z - z', F(z) - F(z') \rangle \geq 0$, $\forall z, z' \in \mathbb{R}^d$. $F$ is $\mu$-strongly monotone if: $\langle z - z', F(z) - F(z') \rangle \geq \mu \|z - z'\|^2$ for all $z, z' \in \mathbb{R}^d$.

The following provides an alternative but equivalent formulation of LA. LA was originally proposed for minimization problems (Zhang et al., 2019b).

**LA equivalent formulation.** We can equivalently write (LA) as follows. At a step $t$: *(i)* a copy of the current iterate $\tilde{z}_t$ is made: $\tilde{z}_t \leftarrow z_t$, *(ii)* $\tilde{z}_t$ is updated $k \geq 1$ times using $B$, yielding $\tilde{z}_{t+k}$, and finally *(iii)* the actual update $z_{t+1}$ is obtained as a *point that lies on a line between* the current $z_t$ iterate and the predicted one $\tilde{z}_{t+k}$:

$$z_{t+1} \leftarrow z_t + \alpha(\tilde{z}_{t+k} - z_t), \quad \alpha \in [0, 1]. \tag{LA}$$

In addition to those presented in the main part, we describe the following popular VI method.

**Optimistic Gradient Descent (OGD).** The update rule of Optimistic Gradient Descent OGD ((OGD) Popov, 1980) is:

$$z_{t+1} = z_t - 2\eta F(z_t) + \eta F(z_{t-1}), \tag{OGD}$$

where $\eta \in (0, 1)$ is the learning rate.

### B.1.2 PSEUDOCODE FOR EXTRAGRADIENT

In Algorithm 2 outlines the *Extragradient* optimizer (Korpelevich, 1976), which we employ in EG-MARL. This method uses a gradient-based optimizer to compute the extrapolation iterate, then applies the gradient at the extrapolated point to perform an actual update step. The extragradient optimizer is used to update all agents' actor and critic networks. In our experiments, we use Adam for both the extrapolation and update steps, maintaining the same learning intervals and parameters as in the baseline algorithm.

---

**Algorithm 2** Extragradient optimizer; Can be used as $B$ in algorithm 1.

---

1: **Input:** learning rate $\eta_\psi$, initial weights $\psi$, loss $\ell^\psi$, extrapolation steps $t$
2: $\psi^{copy} \leftarrow \psi$ *(Save current parameters)*
3: **for** $i \in 1, \ldots, t$ **do**
4:     $\psi = \psi - \eta_\psi \nabla_\psi \ell^\psi(\psi)$ *(Compute the extrapolated $\psi$)*
5: **end for**
6: $\psi = \psi^{copy} - \eta_\psi \nabla_\psi \ell^\psi(\psi)$ *(update $\psi$)*
7: **Output:** $\psi$

---

### B.1.3 PSEUDOCODE FOR NESTED LOOKAHEAD FOR A TWO-PLAYER GAME

For completeness, in Algorithm 3 we give the details of adapted version of the nested Lookahead-Minmax algorithm proposed in (Algorithm 6, Chavdarova et al., 2021) with two-levels.

In the given algorithm, the actor and critic parameters are first updated using a gradient-based optimizer. At interval $k^{(1)}$, backtracking is done between the current parameters and first-level copies (slow weights) and they get updated. At interval $k^{(2)} = c_j k^{(1)}$ backtracking is performed again with second-level copies (slower weights), updating both first- and second-level copies with the averaged version.

**Algorithm 3** Pseudocode of Two-Level Nested Lookahead–Minmax. (Chavdarova et al., 2021)

---

1: **Input:** number of episodes $t$, learning rates $\eta_{\boldsymbol{\theta}}, \eta_{\boldsymbol{w}}$, initial weights $\{\boldsymbol{\theta}, \boldsymbol{\theta}^{(1)}, \boldsymbol{\theta}^{(2)}\}$ and $\{(\boldsymbol{w}, \boldsymbol{w}^{(1)}, \boldsymbol{w}^{(2)})\}$, LA hyperparameters: levels $l = 2$, $(k^{(1)}, k^{(2)})$ and $(\alpha_{\boldsymbol{\theta}}, \alpha_{\boldsymbol{w}})$, losses $\ell^{\boldsymbol{\theta}}$, $\ell^{\boldsymbol{w}}$, real–data distribution $p_d$, noise–data distribution $p_z$.

2: **for** $r \in 1, \ldots, t$ **do**

3:     $\boldsymbol{x} \sim p_d, \boldsymbol{z} \sim p_z$

4:     $\boldsymbol{w} \leftarrow \boldsymbol{w} - \eta_{\boldsymbol{w}} \nabla_{\boldsymbol{w}} \ell^{\boldsymbol{w}}(\boldsymbol{w}, \boldsymbol{x}, \boldsymbol{z})$           *(update $\boldsymbol{w}$)*

5:     $\boldsymbol{\theta} \leftarrow \boldsymbol{\theta} - \eta_{\boldsymbol{\theta}} \nabla_{\boldsymbol{\theta}} \ell^{\boldsymbol{\theta}}(\boldsymbol{\theta}, \boldsymbol{x}, \boldsymbol{z})$           *(update $\boldsymbol{\theta}$ )*

6:     **if** $r\%k^{(1)} == 0$ **then**

7:         $\boldsymbol{w} \leftarrow \boldsymbol{w}^{(1)} + \alpha_{\boldsymbol{w}}(\boldsymbol{w} - \boldsymbol{w}^{(1)})$     *(backtracking on interpolated line $\boldsymbol{w}^{(1)}, \boldsymbol{w}$)*

8:         $\boldsymbol{\theta} \leftarrow \boldsymbol{\theta}^{(1)} + \alpha_{\boldsymbol{\theta}}(\boldsymbol{\theta} - \boldsymbol{\theta}^{(1)})$     *(backtracking on interpolated line $\boldsymbol{\theta}^{(1)}, \boldsymbol{\theta}$)*

9:         $(\boldsymbol{\theta}^{(1)}, \boldsymbol{w}^{(1)}) \leftarrow (\boldsymbol{\theta}, \boldsymbol{w})$     *(update slow checkpoints)*

10:     **end if**

11:     **if** $r\%k^{(2)} == 0$ **then**

12:         $\boldsymbol{w} \leftarrow \boldsymbol{w}^{(2)} + \alpha_{\boldsymbol{w}}(\boldsymbol{w} - \boldsymbol{w}^{(2)})$     *(backtracking on interpolated line $\boldsymbol{w}^{(2)}, \boldsymbol{w}$)*

13:         $\boldsymbol{\theta} \leftarrow \boldsymbol{\theta}^{(2)} + \alpha_{\boldsymbol{\theta}}(\boldsymbol{\theta} - \boldsymbol{\theta}^{(2)})$     *(backtracking on interpolated line $\boldsymbol{\theta}^{(2)}, \boldsymbol{\theta}$)*

14:         $(\boldsymbol{\theta}^{(2)}, \boldsymbol{w}^{(2)}) \leftarrow (\boldsymbol{\theta}, \boldsymbol{w})$     *(update super-slow checkpoints)*

15:         $(\boldsymbol{\theta}_{(1)}, \boldsymbol{w}_{(1)}) \leftarrow (\boldsymbol{\theta}, \boldsymbol{w})$     *(update slow checkpoints)*

16:     **end if**

17: **end for**

18: **Output:** $\boldsymbol{\theta}^{(2)}, \boldsymbol{w}^{(2)}$

---

### B.2 MARL ALGORITHMS

#### B.2.1 DETAILS ON THE MADDPG ALGORITHM

The MADDPG algorithm (Lowe et al., 2017) is outlined in Algorithm 4. An empty replay buffer $\mathcal{D}$ is initialized to store experiences. In each episode, the environment is reset and agents choose actions to perform accordingly. After, experiences in the form of *(state, action, reward, next state)* are saved to $\mathcal{D}$.

After a predetermined number of random iterations, learning begins by sampling batches from $\mathcal{D}$. The critic of agent $i$ receives the sampled joint actions $\boldsymbol{a}$ of all agents and the state information of agent $i$ to output the predicted $Q$-value of agent $i$. Deep Q-learning (Mnih et al., 2015) is then used to update the critic network; lines 20–21. Then, the agents' policy network is optimized using policy gradient; refer to 23. Finally, following each learning iteration, the target networks are updated towards current actor and critic networks using a fraction $\tau$. Then the process repeats until the end of training.

All networks are optimized using the Adam optimizer (Kingma & Ba, 2015). Once training is complete, each agent's actor operates independently during execution. This approach is applicable across cooperative, competitive, and mixed environments.

#### B.2.2 MATD3 ALGORITHM

We provide a psuedo code for MATD3 algorithm from (Ackermann et al., 2019) in algorithm 5. As discussed in the main section, MATD3 was introduced as an improvement to MADDPG and follows a similar structure, except for the learning steps. After sampling a batch from the replay buffer $\mathcal{D}$, both critics of each agent are updated using Deep Q-learning, with the target computed using the minimum of the two critic values (notice the difference in lines 20 and 20of the two algorithms). The actor networks are then updated via policy gradient, using only the Q-value from the first critic; see line 24.

#### B.2.3 COUNTERFACTUAL MULTI-AGENT POLICY GRADIENTS (**COMA**, (FOERSTER ET AL., 2018))

COMA is an actor-critic multi-agent algorithm based on the CTDE paradigm, with one centralized critic and $n$ decentralized actors. Additionally, COMA directly addresses the credit assignment problem in multi-agent settings by: *(i)* computing a counterfactual baseline for each agent $b_i(s, \mathbf{a}_{-i})$,

**Algorithm 4** Pseudocode for MADDPG (Lowe et al., 2017).

1: **Input:** Environment $\mathcal{E}$, number of agents $n$, number of episodes $t$, action spaces $\{\mathcal{A}_i\}_{i=1}^n$, number of random steps $t_{\text{rand}}$ before learning, learning interval $t_{\text{learn}}$, actor networks $\{\boldsymbol{\mu}_i\}_{i=1}^n$, with initial weights $\boldsymbol{\theta} \equiv \{\boldsymbol{\theta}_i\}_{i=1}^n$, critic networks $\{\mathbf{Q}_i\}_{i=1}^n$ with initial weights $\boldsymbol{w} \equiv \{\boldsymbol{w}_i\}_{i=1}^n$, learning rates $\eta_{\boldsymbol{\theta}}, \eta_{\boldsymbol{w}}$, optimizer $B$ (e.g., Adam), discount factor $\gamma$, soft update parameter $\tau$.

2: **Initialize:**

3:     Replay buffer $\mathcal{D} \leftarrow \varnothing$

4: **for all** episode $e \in 1, \dots, t$ **do**

5:     $\boldsymbol{x} \leftarrow Sample(\mathcal{E})$                     *(sample from environment $\mathcal{E}$)*

6:     $step \leftarrow 1$

7:     **repeat**

8:       **if** $e \leq t_{\text{rand}}$ **then**

9:         for each agent $i$, $a_i \sim \mathcal{A}_i$            *(sample actions randomly)*

10:      **else**

11:        for each agent $i$, select action $a_i = \boldsymbol{\mu}_i(\boldsymbol{o}_i) + \mathcal{N}_t$ using current policy and exploration noise

12:      **end if**

13:      Execute actions $\boldsymbol{a} = (a_1, \dots, a_n)$, observe rewards $\mathbf{r}$ and new state $\boldsymbol{x}'$ *(apply actions and record results)*

14:      replay buffer $\mathcal{D} \leftarrow (\boldsymbol{x}, \boldsymbol{a}, \mathbf{r}, \boldsymbol{x}')$

15:      $\boldsymbol{x} \leftarrow \boldsymbol{x}'$

16:                                 *(apply learning step if applicable)*

17:      **if** $step \% t_{\text{learn}} = 0$ **then**

18:        **for all** agent $i \in 1, \dots, n$ **do**

19:          sample batch $\mathcal{B} : \{(\boldsymbol{x}^j, \boldsymbol{a}^j, \mathbf{r}^j, \boldsymbol{x}'^j)\}_{j=1}^{|\mathcal{B}|}$ from $\mathcal{D}$

20:          $y^j \leftarrow r_i^j + \gamma \mathbf{Q}^{\bar{\boldsymbol{\mu}}}(\boldsymbol{x}'^j, a_1', \dots, a_n')$, where $a_k' = \bar{\boldsymbol{\mu}}_k(\boldsymbol{o}_k'^j)$

21:          Update critic by minimizing the loss (using optimizer $B$):

$$\ell(\boldsymbol{w}_i) = \tfrac{1}{|\mathcal{B}|} \sum_j \left( y^j - \mathbf{Q}_i^{\boldsymbol{\mu}}(\boldsymbol{x}^j, a_1^j, \dots, a_n^j) \right)^2$$

22:          Update actor policy using policy gradient formula and optimizer $B$

23:          $\nabla_{\boldsymbol{\theta}_i} J \approx \tfrac{1}{|\mathcal{B}|} \sum_j \nabla_{\boldsymbol{\theta}_i} \boldsymbol{\mu}_i(\boldsymbol{o}_i^j) \nabla_{a_i} \mathbf{Q}_i^{\boldsymbol{\mu}}(\boldsymbol{x}^j, a_1^j, \dots, a_i, \dots, a_n^j)$, where $a_i = \boldsymbol{\mu}_i(\boldsymbol{o}_i^j)$

24:        **end for**

25:        **for all** agent $i \in [n]$ **do**

26:         $\bar{\boldsymbol{\theta}}_i \leftarrow \tau \boldsymbol{\theta}_i + (1 - \tau) \bar{\boldsymbol{\theta}}_i$           *(update target networks)*

27:         $\bar{\boldsymbol{w}}_i \leftarrow \tau \boldsymbol{w}_i + (1 - \tau) \bar{\boldsymbol{w}}_i$

28:        **end for**

29:      **end if**

30:      $step \leftarrow step + 1$

31:     **until** environment terminates

32: **end for**

33: **Output:** $\boldsymbol{\theta}, \boldsymbol{w}$

---

**Algorithm 5** Pseudocode for MATD3 (Ackermann et al., 2019).

---

1: **Input:** Environment $\mathcal{E}$, number of agents $n$, number of episodes $t$, action spaces $\{\mathcal{A}_i\}_{i=1}^n$, number of random steps $t_{\text{rand}}$ before learning, learning interval $t_{\text{learn}}$, actor networks $\{\boldsymbol{\mu}_i\}_{i=1}^n$, with initial weights $\boldsymbol{\theta} \equiv \{\boldsymbol{\theta}_i\}_{i=1}^n$, both critic networks, $\{\mathbf{Q}_{i,1}, \mathbf{Q}_{i,2}\}_{i=1}^n$ with initial weights $\boldsymbol{w} \equiv \{\boldsymbol{w}_{i,1}, \boldsymbol{w}_{i,2}\}_{i=1}^n$, learning rates $\eta_{\boldsymbol{\theta}}, \eta_{\boldsymbol{w}}$, optimizer $B$ (e.g., Adam), discount factor $\gamma$, soft update parameter $\tau$, policy update frequency $p$.

2: **Initialize:**

3:      Replay buffer $\mathcal{D} \leftarrow \varnothing$

4: **for all** episode $e \in 1, \ldots, t$ **do**

5:      $\boldsymbol{x} \leftarrow Sample(\mathcal{E})$          *(sample from environment $\mathcal{E}$)*

6:      $step \leftarrow 1$

7:      **repeat**

8:          **if** $e \le t_{\text{rand}}$ **then**

9:              for each agent $i$, $a_i \sim \mathcal{A}_i$          *(sample actions randomly)*

10:          **else**

11:              for each agent $i$, select action $a_i = \boldsymbol{\mu}_i(\boldsymbol{o}_i) + \epsilon$ using current policy with some exploration noise

12:          **end if**

13:      Execute actions $\boldsymbol{a} = (a_1, \ldots, a_n)$, observe rewards $\mathbf{r}$ and new state $\boldsymbol{x}'$ *(apply actions and record results)*

14:      replay buffer $\mathcal{D} \leftarrow (\boldsymbol{x}, \boldsymbol{a}, \mathbf{r}, \boldsymbol{x}')$

15:      $\boldsymbol{x} \leftarrow \boldsymbol{x}'$

16:                                              *(apply learning step if applicable)*

17:      **if** $step \% t_{\text{learn}} = 0$ **then**

18:          **for all** agent $i \in [n]$ **do**

19:              sample batch $\{(\boldsymbol{x}^j, \boldsymbol{a}^j, \mathbf{r}^j, \boldsymbol{x}'^j)\}_{j=1}^{|\mathcal{B}|}$ from $\mathcal{D}$

20:              $y^j \leftarrow r_i^j + \gamma \min_{m=1,2} \mathbf{Q}_{i,m}^{\bar{\boldsymbol{\mu}}}(\boldsymbol{x}'^j, a_1', \ldots, a_n')$, where $a_k' = \bar{\boldsymbol{\mu}}_k(\boldsymbol{o}_k'^j) + \epsilon$

21:              Update both critics, $m = 1, 2$ by minimizing the loss (using optimizer $B$):

$$\ell(\boldsymbol{w}_{i,m}) = \frac{1}{|\mathcal{B}|} \sum_j \left( y^j - \mathbf{Q}_{i,m}^{\boldsymbol{\mu}}(\boldsymbol{x}^j, a_1^j, \ldots, a_n^j) \right)^2$$

22:          **if** $step \% p = 0$ **then**

23:              Update actor policy using policy gradient formula and optimizer $B$

24:              $\nabla_{\boldsymbol{\theta}_i} J \approx \frac{1}{|\mathcal{B}|} \sum_j \nabla_{\boldsymbol{\theta}_i} \boldsymbol{\mu}_i(\boldsymbol{o}_i^j) \nabla_{a_i} \mathbf{Q}_{i,1}^{\boldsymbol{\mu}}(\boldsymbol{x}^j, a_1^j, \ldots, a_i, \ldots, a_n^j)$, where $a_i = \boldsymbol{\mu}_i(\boldsymbol{o}_i^j)$

25:              $\bar{\boldsymbol{\theta}}_i \leftarrow \tau \boldsymbol{\theta}_i + (1 - \tau)\bar{\boldsymbol{\theta}}_i$          *(update target networks)*

26:              $\bar{\boldsymbol{w}}_{i,m} \leftarrow \tau \boldsymbol{w}_{i,m} + (1 - \tau)\bar{\boldsymbol{w}}_{i,m}$

27:          **end if**

28:          **end for**

29:      **end if**

30:      $step \leftarrow step + 1$

31:      **until** environment terminates

32: **end for**

33: **Output:** $\boldsymbol{\theta}, \boldsymbol{w}$

---

*(ii)* using this baseline to estimate the advantage $A_i$ of the chosen action over all others in $\mathcal{A}_i$, and *(iii)* leveraging this advantage to update individual policies. This ensures that policy updates reflect each agent's true contribution to the overall reward.

### B.2.4 MULTI-AGENT TRUST REGION POLICY OPTIMIZATION (**MATRPO**, (LI & HE, 2023))

Trust Region Policy Optimization (TRPO, Schulman et al., 2015) is a policy optimization method that ensures stable updates by constraining policy changes within a trust region. This constraint is enforced using the KL-divergence, and the update step is computed using natural gradient descent.

Extending TRPO to the cooperative multi-agent setting introduces challenges due to non-stationarity. To address this, MATRPO employs a centralized critic, represented by a central value function $V(\boldsymbol{s})$, which leverages shared information among agents to estimate the Generalized Advantage Estimator (GAE) $A_i$. The advantage function is then used in the policy gradient update, while ensuring that the KL-divergence constraint is respected, maintaining stable and coordinated learning across agents.

### B.2.5 MULTI-AGENT PROXIMAL POLICY OPTIMIZATION (MAPPO, YU ET AL., 2021)

One of the widely used algorithms in practice is MAPPO, an extension of Proximal Policy Optimization (PPO, Schulman et al., 2017) to the multi-agent setting. Similar to TRPO, PPO ensures that policy updates remain within a small, stable region, but instead of enforcing a KL-divergence constraint, it uses clipping. This clipping mechanism simplifies the optimization process, allowing updates to be performed efficiently using standard gradient ascent methods.

MAPPO is an on-policy algorithm that employs a centralized critic while maintaining decentralized actor networks for each agent. Its critic update follows the same rule as MATRPO, but for the policy update, it optimizes a clipped surrogate objective, which restricts the policy update step size, ensuring stable and efficient learning.

## C Off-Policy Deterministic Actor-Critic Methods Analysis & Missing Proofs

### C.1 Single-agent Setting

The following describes the considered setting in Lemma 1 in more detail. For simplicity, we consider a single state $s'$ for each trajectory.

**Setting 1** (off-policy deterministic AC setting)**.** *Consider the following off-policy deterministic actor-critic setting (Silver et al., 2014):*

*1. Action $a \in \mathbb{R}$*
*2. State $s \in \mathbb{R}^d$*
*3. State-action features $\phi \in \mathbb{R}^c$ defined as:*

$$\phi(s, a) \triangleq f_\phi(s) + a\, m(s) \in \mathbb{R}^c, \qquad with\ f_\phi \colon \mathbb{R}^d \to \mathbb{R}^c,\ m \colon \mathbb{R}^d \to \mathbb{R}^c.$$

*Notice that:*

$$\frac{\partial \phi(s, a)}{\partial a} = m(s), \qquad \left.\frac{\partial \phi(s, a)}{\partial \theta}\right|_{a\ fixed} = 0.$$

*4. Critic parameters $w \in \mathbb{R}^c$*
*5. Actor parameters $\theta \in \mathbb{R}^d$*
*6. Critic (linear Q-value): $Q_w(s, a) = \langle w, \phi(s, a)\rangle$*
*7. Actor (deterministic linear policy): $\pi_\theta(s) \triangleq \langle \theta, s\rangle$*
*8. Batch of experiences: $\{s_i, a_i, r_i, s_i'\}_{i=1}^B$*

Taking $s \in \mathbb{R}^d$, of the same dimension as $\theta$, is without loss of generality for the following analysis, since one can use a different dimension and a feature map. We also introduce the following shorthands:

$$\phi_i \triangleq \phi(s_i, a_i), \tag{$\phi_i$}$$

$$a_i' \triangleq \pi_\theta(s_i') = \langle \theta, s_i'\rangle, \tag{$a_i'$}$$

$$\phi_i' \triangleq \phi(s_i', a_i') = f_\phi(s_i') + a_i' m(s_i'), \tag{$\phi_i'$}$$

$$\delta_i \triangleq \langle w, \phi_i\rangle - r_i - \gamma\langle w, \phi_i'\rangle. \tag{$\delta_i$}$$

We next state the complete statement of the informal Lemma 1 and provide its proof.

**Lemma 1** (Complex eigenvalues in off-policy deterministic AC)**.** *Consider the off-policy deterministic AC model of setting 1 with $\gamma > 0$. Its associated (simultaneous) gradient operator $F \colon \mathbb{R}^{d+c} \to \mathbb{R}^{d+c}$ is:*

$$F\left(\begin{bmatrix} w \\ \theta \end{bmatrix}\right) = \begin{bmatrix} F_w \\ F_\theta \end{bmatrix} = \begin{bmatrix} \frac{2}{B}\sum_{i=1}^B \delta_i(\phi_i - \gamma\phi_i') \\ -\frac{1}{B}\sum_{i=1}^B s_i \langle w, m(s_i)\rangle \end{bmatrix}. \tag{1}$$

*Its associated Jacobian $J \colon \mathbb{R}^{c+d} \to \mathbb{R}^{(c+d)\times(c+d)}$ has the following block form:*

$$J = \begin{bmatrix} J_{ww} & J_{w\theta} \\ J_{\theta w} & J_{\theta\theta} \end{bmatrix}, \qquad with\ J_{ww} \succeq 0,$$

*with:*

$$
\textbf{(1)} \ J_{\boldsymbol{ww}} = \frac{\partial F_{\boldsymbol{w}}}{\partial \boldsymbol{w}} = \frac{2}{B} \sum_{i=1}^{B} (\phi_i - \gamma \phi_i')(\phi_i - \gamma \phi_i')^{\mathsf{T}} \in \mathbb{R}^{c \times c}
$$

$$
\textbf{(2)} \ J_{\boldsymbol{w\theta}} = \frac{\partial F_{\boldsymbol{w}}}{\partial \boldsymbol{\theta}} = -\frac{2\gamma}{B} \sum_{i=1}^{B} \left[ (\phi_i - \gamma \phi_i') \, \boldsymbol{s}_i'^{\mathsf{T}} \, \boldsymbol{w}^{\mathsf{T}} m(\boldsymbol{s}_i') \ + \ m(\boldsymbol{s}_i') \, \boldsymbol{s}_i'^{\mathsf{T}} \, \delta_i \right] \in \mathbb{R}^{c \times d}
$$

$$
\textbf{(3)} \ J_{\boldsymbol{\theta w}} = \frac{\partial F_{\boldsymbol{\theta}}}{\partial \boldsymbol{w}} = -\frac{1}{B} \sum_{i=1}^{B} \boldsymbol{s}_i \, m(\boldsymbol{s}_i)^{\mathsf{T}} \in \mathbb{R}^{d \times c}
$$

$$
\textbf{(4)} \ J_{\boldsymbol{\theta\theta}} = \frac{\partial F_{\boldsymbol{\theta}}}{\partial \boldsymbol{\theta}} = \mathbf{0}_{d \times d}.
$$

*Then:*

(i) *(Pure cross-term case)* If $J_{\boldsymbol{ww}} = \mathbf{0}$, then $\mathrm{Spec}(J) = \{\pm\sqrt{\lambda_k(J_{\boldsymbol{w\theta}} J_{\boldsymbol{\theta w}})}\}_k$. In particular, if $J_{\boldsymbol{w\theta}} J_{\boldsymbol{\theta w}}$ has a negative eigenvalue, $J$ has a purely imaginary conjugate eigenpair.

(ii) *(Persistence)* If $J_{\boldsymbol{w\theta}} J_{\boldsymbol{\theta w}}$ has a negative eigenvalue, there exists $\varepsilon > 0$ such that for all $\|J_{\boldsymbol{ww}}\| < \varepsilon$ (operator norm), $J$ has a non-real conjugate eigenpair.

(iii) *(Symmetry case)* If $J_{\boldsymbol{w\theta}} = J_{\boldsymbol{\theta w}}^{\mathsf{T}}$ and $J_{\boldsymbol{ww}} = J_{\boldsymbol{ww}}^{\mathsf{T}} \succeq 0$, then $J$ is symmetric and all eigenvalues are real.

*Moreover, $J_{\boldsymbol{w\theta}} \neq J_{\boldsymbol{\theta w}}^{\mathsf{T}}$ unless $\gamma = 0$ or simultaneously $\delta_i \equiv 0$ and $\boldsymbol{w}^{\mathsf{T}} m(\boldsymbol{s}_i') \equiv 0$ for all $i$. Hence, for $\gamma > 0$ and generic data/parameters, $J_{\boldsymbol{w\theta}} J_{\boldsymbol{\theta w}}$ is sign-indefinite and $J$ exhibits a non-real eigenpair.*

*Proof of Lemma 1.* STEP 1: CRITIC LOSS AND DERIVATIVES

**Loss.** Given a batch $\{\boldsymbol{s}_i, a_i, r_i, \boldsymbol{s}_i'\}_{i=1}^{B}$, consider the *temporal difference* (TD) loss for the critic, parametrized by $\boldsymbol{w}$, defined as:

$$
\mathcal{L}(\boldsymbol{w}) = \frac{1}{B} \sum_{i=1}^{B} \Big( \underbrace{\langle \boldsymbol{w}, \phi(\boldsymbol{s}_i, a_i) \rangle}_{\hat{y}_i} - \underbrace{r_i - \gamma \langle \boldsymbol{w}, \phi(\boldsymbol{s}_i', \pi_{\boldsymbol{\theta}}(\boldsymbol{s}_i')) \rangle}_{y_i} \Big)^2
$$

$$
= \frac{1}{B} \sum_{i=1}^{B} \Big( \langle \boldsymbol{w}, \phi_i \rangle - r_i - \gamma \langle \boldsymbol{w}, \phi_i' \rangle \Big)^2 . \tag{2}
$$

Notice that the $\boldsymbol{\theta}$-dependence in the critic loss is implicit, via $a_i'$ in $\phi_i'$.

**First derivative.** The first derivative of the critic loss with respect to $\boldsymbol{w}$ is then:

$$
F_{\boldsymbol{w}}(\boldsymbol{w}, \boldsymbol{\theta}) \equiv \nabla_{\boldsymbol{w}} \mathcal{L}(\boldsymbol{w}) = \frac{2}{B} \sum_{i=1}^{B} \delta_i \big( \phi_i - \gamma \phi_i' \big) \in \mathbb{R}^c. \tag{$F_{\boldsymbol{w}}$}
$$

**Second derivative.** The second derivative matrix for $\boldsymbol{w}$ is then:

$$
J_{\boldsymbol{ww}} = \nabla_{\boldsymbol{w}}^2 \mathcal{L}(\boldsymbol{w}) = \frac{2}{B} \sum_{i=1}^{B} \big( \phi_i - \gamma \phi_i' \big) \big( \phi_i - \gamma \phi_i' \big)^{\mathsf{T}} \in \mathbb{R}^{c \times c}. \tag{$J_{\boldsymbol{ww}}$}
$$

$J_{\boldsymbol{ww}}$ is symmetric, and due to the outer product structure of $J_{\boldsymbol{ww}}$, it is positive semi-definite (PSD); thus, it has non-negative eigenvalues.

**Mixed second derivative: derivative of critic's first derivative w.r.t. actor.** Since:

$$
\frac{\partial \, \phi(\boldsymbol{s}_i', \pi_{\boldsymbol{\theta}}(\boldsymbol{s}_i'))}{\partial \boldsymbol{\theta}} = \frac{\partial \phi(\boldsymbol{s}_i', a_i')}{\partial a} \, \boldsymbol{s}_i'^{\mathsf{T}} = m(\boldsymbol{s}_i') \, \boldsymbol{s}_i'^{\mathsf{T}} \in \mathbb{R}^{c \times d} ,
$$

differentiating ($F_{\boldsymbol{w}}$) w.r.t. $\boldsymbol{\theta}$, and applying the product and chain rules, gives:

$$
J_{\boldsymbol{w\theta}} = \nabla_{\boldsymbol{\theta}} F_{\boldsymbol{w}} = -\frac{2\gamma}{B} \sum_{i=1}^{B} \left[ (\phi_i - \gamma \phi_i') \, \boldsymbol{s}_i'^{\mathsf{T}} \, \boldsymbol{w}^{\mathsf{T}} m(\boldsymbol{s}_i') \ + \ m(\boldsymbol{s}_i') \, \boldsymbol{s}_i'^{\mathsf{T}} \, \delta_i \right] \in \mathbb{R}^{c \times d}. \tag{$J_{\boldsymbol{w\theta}}$}
$$

STEP 2: ACTOR DERIVATIVES

**First derivative.** We adopt the convention that both actor and critic run descent step; thus, for consistency, we use a minus sign to the derivative for $\boldsymbol{\theta}$; i.e., the actor block is the negative policy gradient. Due to the policy gradient theorem for the deterministic setting, it holds:

$$F_{\boldsymbol{\theta}}(\boldsymbol{\theta}) = -\frac{1}{B} \sum_{i=1}^{B} \nabla_{\boldsymbol{\theta}} \pi_{\boldsymbol{\theta}}(\boldsymbol{s}_i) \, \nabla_a Q_{\boldsymbol{w}}(\boldsymbol{s}_i, a)\big|_{a=\pi_{\boldsymbol{\theta}}(\boldsymbol{s}_i)} . \tag{3}$$

Since in the considered setting $Q_{\boldsymbol{w}}(\boldsymbol{s}, a) = \langle \boldsymbol{w}, \phi(\boldsymbol{s}, a) \rangle = \langle \boldsymbol{w}, f_\phi(\boldsymbol{s}) \rangle + a \langle \boldsymbol{w}, m(\boldsymbol{s}) \rangle$, we have: $\nabla_a Q_{\boldsymbol{w}}(\boldsymbol{s}, a) = \boldsymbol{w}^\mathsf{T} m(\boldsymbol{s})$ . Thus, for the first derivative w.r.t. $\boldsymbol{\theta}$ for the considered setting we have:

$$\begin{aligned} F_{\boldsymbol{\theta}}(\boldsymbol{w}, \boldsymbol{\theta}) &= -\frac{1}{B} \sum_{i=1}^{B} \nabla_{\boldsymbol{\theta}} \pi_{\boldsymbol{\theta}}(\boldsymbol{s}_i) \, \nabla_a Q_{\boldsymbol{w}}(\boldsymbol{s}_i, a)\big|_{a=\pi_{\boldsymbol{\theta}}(\boldsymbol{s}_i)} \\ &= -\frac{1}{B} \sum_{i=1}^{B} \boldsymbol{s}_i \, \langle \boldsymbol{w}, m(\boldsymbol{s}_i) \rangle . \end{aligned} \tag{$F_{\boldsymbol{\theta}}$}$$

**Second derivative.** Since the actor gradient in equation ($F_{\boldsymbol{\theta}}$) doesn't have $\boldsymbol{\theta}$ terms:

$$J_{\boldsymbol{\theta}\boldsymbol{\theta}} = \nabla_{\boldsymbol{\theta}} F_{\boldsymbol{\theta}} \; = \; \mathbf{0}_{d \times d} \in \mathbb{R}^{d \times d} . \tag{$J_{\boldsymbol{\theta}\boldsymbol{\theta}}$}$$

**Mixed second derivative: derivative of actor's first derivative w.r.t. critic.** Differentiating ($F_{\boldsymbol{\theta}}$) with respect to $\boldsymbol{w}$ yields:

$$J_{\boldsymbol{\theta}\boldsymbol{w}} = \nabla_{\boldsymbol{w}} F_{\boldsymbol{\theta}} = -\frac{1}{B} \sum_{i=1}^{B} \boldsymbol{s}_i \, m(\boldsymbol{s}_i)^\mathsf{T} \in \mathbb{R}^{d \times c} . \tag{$J_{\boldsymbol{\theta}\boldsymbol{w}}$}$$

STEP 3: SPECTRAL ANALYSIS OF $J$

(i) (*Pure cross-term case*) If $J_{\boldsymbol{w}\boldsymbol{w}} = \mathbf{0}$, then $J^2 = \mathrm{diag}(J_{\boldsymbol{w}\boldsymbol{\theta}} J_{\boldsymbol{\theta}\boldsymbol{w}}, J_{\boldsymbol{\theta}\boldsymbol{w}} J_{\boldsymbol{w}\boldsymbol{\theta}})$, so the nonzero eigenvalues of $J$ are the signed square roots of those of $J_{\boldsymbol{w}\boldsymbol{\theta}} J_{\boldsymbol{\theta}\boldsymbol{w}}$ and $J_{\boldsymbol{\theta}\boldsymbol{w}} J_{\boldsymbol{w}\boldsymbol{\theta}}$ (where $J_{\boldsymbol{w}\boldsymbol{\theta}} J_{\boldsymbol{\theta}\boldsymbol{w}}$ and $J_{\boldsymbol{\theta}\boldsymbol{w}} J_{\boldsymbol{w}\boldsymbol{\theta}}$ share the same nonzero spectrum). Negative eigenvalues of $J_{\boldsymbol{w}\boldsymbol{\theta}} J_{\boldsymbol{\theta}\boldsymbol{w}}$ yield purely imaginary eigenvalues of $J$.

(ii) (*Persistence under small symmetric $J_{\boldsymbol{w}\boldsymbol{w}} \succeq 0$*) Let

$$J(\epsilon) = \begin{bmatrix} \epsilon J_{\boldsymbol{w}\boldsymbol{w}} & J_{\boldsymbol{w}\boldsymbol{\theta}} \\ J_{\boldsymbol{\theta}\boldsymbol{w}} & \mathbf{0} \end{bmatrix} ,$$

where $\epsilon J_{\boldsymbol{w}\boldsymbol{w}}$ is symmetric semidefinite and $\epsilon = \|J_{\boldsymbol{w}\boldsymbol{w}}\|$ (operator norm) is small. By classical matrix perturbation theory, eigenvalues depend continuously (analytically) on its matrix entries (e.g., Kato, 1966); a simple imaginary pair at $J_{\boldsymbol{w}\boldsymbol{w}} = \mathbf{0}$ cannot become real under a sufficiently small symmetric perturbation $J_{\boldsymbol{w}\boldsymbol{w}} \succeq 0$ without crossing the real axis. Thus the non-real pair persists for small $\|J_{\boldsymbol{w}\boldsymbol{w}}\|$, that is for all $0 \le \epsilon \le \epsilon_0$ small enough.

(iii) If $J_{\boldsymbol{w}\boldsymbol{\theta}} = J_{\boldsymbol{\theta}\boldsymbol{w}}^\mathsf{T}$ and $J_{\boldsymbol{w}\boldsymbol{w}} = J_{\boldsymbol{w}\boldsymbol{w}}^\mathsf{T}$, the Jacobian $J$ is symmetric, so its spectrum is real.

The following shows that (iii) forces degenerate conditions. In our setting,

$$J_{\boldsymbol{w}\boldsymbol{\theta}} = -\frac{2\gamma}{B} \sum_{i=1}^{B} \Big[ (\phi_i - \gamma\phi_i') \, \boldsymbol{s}_i'^\mathsf{T} \, \boldsymbol{w}^\mathsf{T} m(\boldsymbol{s}_i') + m(\boldsymbol{s}_i') \, \boldsymbol{s}_i'^\mathsf{T} \delta_i \Big], \qquad J_{\boldsymbol{\theta}\boldsymbol{w}} = -\frac{1}{B} \sum_{i=1}^{B} \boldsymbol{s}_i m(\boldsymbol{s}_i)^\mathsf{T},$$

and

$$J_{\boldsymbol{\theta}\boldsymbol{w}}^\mathsf{T} = -\frac{1}{B} \sum_{i=1}^{B} m(\boldsymbol{s}_i) \boldsymbol{s}_i^\mathsf{T} .$$

Thus, $J_{\boldsymbol{w\theta}} = J_{\boldsymbol{\theta w}}^{\mathsf{T}}$ would require *(i)* $\gamma = 0$ (which kills the dependence on the next-state) and that $J_{\boldsymbol{\theta w}} = \boldsymbol{0}$, or *(ii)* the vanishing of both the factors $\delta_i$ and $\boldsymbol{w}^{\mathsf{T}} m(\boldsymbol{s}_i')$ for all $i$; these are stringent/degenerate conditions. So generically $J_{\boldsymbol{w\theta}} \neq J_{\boldsymbol{\theta w}}^{\mathsf{T}}$.

$\square$

*Remark* 1 (Sign convention). Above, we used the sign convention for descending in $F$, which, due to the negative sign in the derivative of $\boldsymbol{\theta}$, implies that the actor follows the ascent direction. Even if the alternative sign convention is used, the above lemma is invariant under the actor sign flip.

### C.1.1 AN INSTANCE OF SINGLE ACTOR-CRITIC SETTING 1: FULL DESCRIPTION AND STATIONARY POINT

This section describes an instance setting 1 in Appendix C.1, which is used for the numerical experiment in Figure 2. The specific example uses $d=c=2$, and a fixed off–policy batch of size $B=2$. The purpose of the example is to give an intuition of the rotational dynamics in actor-critic and that LA can indeed help with mitigating them.

**Setup.** Suppose we have state and action spaces

$$\boldsymbol{s} \in \mathbb{R}^2, \qquad a \in \mathbb{R},$$

and consider a deterministic linear policy

$$\pi_{\boldsymbol{\theta}}(\boldsymbol{s}) = \langle \boldsymbol{\theta}, \boldsymbol{s} \rangle, \qquad \boldsymbol{\theta} \in \mathbb{R}^2,$$

together with a linear critic

$$Q_{\boldsymbol{w}}(\boldsymbol{s}, a) = \langle \boldsymbol{w}, \phi(\boldsymbol{s}, a) \rangle, \qquad \boldsymbol{w} \in \mathbb{R}^2.$$

The state-action features are defined as in Setting 1 with

$$\phi(\boldsymbol{s}, a) = f_\phi(\boldsymbol{s}) + a\, m(\boldsymbol{s}), \qquad f_\phi(\boldsymbol{s}) \equiv \boldsymbol{0}, \qquad m(\boldsymbol{s}) = M\,\boldsymbol{s}, \quad M = \begin{bmatrix} 0 & -1 \\ 1 & 0 \end{bmatrix}.$$

Thus, $m$ maps any vector to a $+90°$ rotation.

We fix a small off-policy batch of size $B=2$:

$$\left\{ (\boldsymbol{s}_i, a_i, r_i, \boldsymbol{s}_i') \right\}_{i=1}^{2} = \left\{ (e_1, 1, r_1, e_2), (e_2, 1, r_2, e_1) \right\}, \qquad r_1 > 0,\ r_2 > 0, \qquad \gamma \in [0, 1),$$

where $e_1 = [1, 0]^{\top}$ and $e_2 = [0, 1]^{\top}$, hence, the future states are *swapped*: $\boldsymbol{s}_1' = e_2$ and $\boldsymbol{s}_2' = e_1$. We use the shorthand notations from $(\phi_i)$–$(\delta_i)$ defined previously.

**Deriving the operator.** We first compute the feature maps induced by the chosen batch. Using $m(e_1) = Me_1 = e_2$, $m(e_2) = Me_2 = -e_1$, and $f_\phi \equiv \boldsymbol{0}$, the features of the sampled state-action pairs are

$$\phi_1 = \phi(e_1, 1) = m(e_1) = e_2, \qquad \phi_2 = \phi(e_2, 1) = m(e_2) = -e_1.$$

Next, we compute the actions at the next states under the current policy:

$$a_1' = \pi_{\boldsymbol{\theta}}(e_2) = \langle \boldsymbol{\theta}, e_2 \rangle = \theta_2, \qquad a_2' = \pi_{\boldsymbol{\theta}}(e_1) = \langle \boldsymbol{\theta}, e_1 \rangle = \theta_1.$$

Using these, the features of the next states and actions are

$$\phi_1' = \phi(e_2, a_1') = \theta_2\, m(e_2) = -\theta_2\, e_1, \qquad \phi_2' = \phi(e_1, a_2') = \theta_1\, m(e_1) = \theta_1\, e_2.$$

Substituting those formulas and using direct omputations, the difference terms $\phi_i - \gamma\phi_i'$, can be written as:

$$\phi_1 - \gamma\phi_1' = e_2 - \gamma(-\theta_2 e_1) = \begin{bmatrix} \gamma\theta_2 \\ 1 \end{bmatrix}, \qquad \phi_2 - \gamma\phi_2' = -e_1 - \gamma(\theta_1 e_2) = \begin{bmatrix} -1 \\ -\gamma\theta_1 \end{bmatrix}.$$

We previously defined the TD-error in ($\delta_i$):
$$\delta_i = \langle \boldsymbol{w}, \boldsymbol{\phi}_i \rangle - r_i - \gamma \langle \boldsymbol{w}, \boldsymbol{\phi}_i' \rangle.$$
Writing $\boldsymbol{w} = (w_1, w_2)^\top$, we obtain
$$\delta_1 = \langle \boldsymbol{w}, \boldsymbol{\phi}_1 \rangle - r_1 - \gamma \langle \boldsymbol{w}, \boldsymbol{\phi}_1' \rangle = \langle \boldsymbol{w}, e_2 \rangle - r_1 - \gamma \langle \boldsymbol{w}, -\theta_2 e_1 \rangle$$
$$= w_2 - r_1 + \gamma\,\theta_2\,w_1,$$
$$\delta_2 = \langle \boldsymbol{w}, \boldsymbol{\phi}_2 \rangle - r_2 - \gamma \langle \boldsymbol{w}, \boldsymbol{\phi}_2' \rangle = \langle \boldsymbol{w}, -e_1 \rangle - r_2 - \gamma \langle \boldsymbol{w}, \theta_1 e_2 \rangle$$
$$= -w_1 - r_2 - \gamma\,\theta_1\,w_2.$$

Recall from the proof of Lemma 1 setting that:
$$F_{\boldsymbol{w}}(\boldsymbol{w}, \boldsymbol{\theta}) = \frac{2}{B} \sum_{i=1}^{B} \delta_i \big( \boldsymbol{\phi}_i - \gamma \boldsymbol{\phi}_i' \big), \qquad F_{\boldsymbol{\theta}}(\boldsymbol{w}, \boldsymbol{\theta}) = -\frac{1}{B} \sum_{i=1}^{B} \boldsymbol{s}_i \, \langle \boldsymbol{w}, m(\boldsymbol{s}_i) \rangle.$$

For our toy example, we have $B = 2$, so the factor $2/B$ in $F_{\boldsymbol{w}}$ equals 1.

Substituting the expressions above, we get the operator of our toy example:
$$F_{\boldsymbol{w}}(\boldsymbol{w}, \boldsymbol{\theta}) = \delta_1 \big( \boldsymbol{\phi}_1 - \gamma \boldsymbol{\phi}_1' \big) + \delta_2 \big( \boldsymbol{\phi}_2 - \gamma \boldsymbol{\phi}_2' \big)$$
$$= \delta_1 \begin{bmatrix} \gamma\theta_2 \\ 1 \end{bmatrix} + \delta_2 \begin{bmatrix} -1 \\ -\gamma\theta_1 \end{bmatrix} = \begin{bmatrix} \delta_1(\gamma\theta_2) - \delta_2 \\ \delta_1 - \delta_2(\gamma\theta_1) \end{bmatrix}, \tag{4}$$

$$F_{\boldsymbol{\theta}}(\boldsymbol{w}, \boldsymbol{\theta}) = -\tfrac{1}{2} \Big( e_1 \, \langle \boldsymbol{w}, m(e_1) \rangle + e_2 \, \langle \boldsymbol{w}, m(e_2) \rangle \Big)$$
$$= -\tfrac{1}{2} \Big( e_1 \, \langle \boldsymbol{w}, e_2 \rangle + e_2 \, \langle \boldsymbol{w}, -e_1 \rangle \Big) = -\tfrac{1}{2} \begin{bmatrix} w_2 \\ -w_1 \end{bmatrix} = -\tfrac{1}{2} M^\top \boldsymbol{w}. \tag{5}$$

**Closed-form stationary point $(\boldsymbol{w}^*, \boldsymbol{\theta}^*)$.**  We now solve for a stationary point $(\boldsymbol{w}^*, \boldsymbol{\theta}^*)$ satisfying
$$F_{\boldsymbol{\theta}}(\boldsymbol{w}^*, \boldsymbol{\theta}^*) = \boldsymbol{0}, \qquad F_{\boldsymbol{w}}(\boldsymbol{w}^*, \boldsymbol{\theta}^*) = \boldsymbol{0}.$$

**Actor block.**  From (5), the condition $F_{\boldsymbol{\theta}}(\boldsymbol{w}^*, \boldsymbol{\theta}^*) = \boldsymbol{0}$ reads
$$-\tfrac{1}{2} \begin{bmatrix} w_2^* \\ -w_1^* \end{bmatrix} = \boldsymbol{0} \quad \implies \quad \boldsymbol{w}^* = (0,0)^\top.$$
Thus, at any stationary point of $F$, the critic parameters must be zero in this toy setup.

**Critic block.**  With $\boldsymbol{w}^* = \boldsymbol{0}$, the TD errors simplify to
$$\delta_1 = -r_1, \qquad \delta_2 = -r_2.$$
Plugging these values into (4) gives
$$F_{\boldsymbol{w}}(\boldsymbol{0}, \boldsymbol{\theta}) = \begin{bmatrix} -r_1\,(\gamma\theta_2) + r_2 \\ -r_1 + r_2\,(\gamma\theta_1) \end{bmatrix}.$$
Setting $F_{\boldsymbol{w}}(\boldsymbol{0}, \boldsymbol{\theta}^*) = \boldsymbol{0}$ yields the linear system
$$-r_1\,(\gamma\theta_2^*) + r_2 = 0, \qquad -r_1 + r_2\,(\gamma\theta_1^*) = 0,$$
whose solution is
$$\theta_1^* = \frac{r_1}{\gamma\,r_2}, \qquad \theta_2^* = \frac{r_2}{\gamma\,r_1}.$$

Collecting the two blocks, we obtain the unique stationary point of $F$ in this toy example:
$$\boxed{\boldsymbol{w}^* = (0,0)^\top, \qquad \boldsymbol{\theta}^* = \left( \frac{r_1}{\gamma r_2}, \frac{r_2}{\gamma r_1} \right).}$$

In the symmetric case $r_1 = r_2 = r$, this reduces to
$$\boldsymbol{\theta}^* = (1/\gamma, \ 1/\gamma).$$
In the numerical example used for Figure 2, we chose $r_1 = r_2 = 3$ and $\gamma = 0.99$ which gives the stationary points marked by the dashed lines in the figure.

## C.2 Multi-agent Setting with a Centralized Critic

We extend the single-agent linear setting to $n$ agents with a shared state and scalar actions. Each agent $i$ has its own *centralized* critic with access to the full state and the joint action, and a linear deterministic actor. We keep one unrolled next-state $s'$ per transition, as in the single-agent analysis. We rely on similar shorthand definitions as in Appendix C.1.

**Setting 2** (off-policy deterministic $n$-agent AC; full-access critics). *Consider the following:*

1. *State and joint action:* $s \in \mathbb{R}^d$, $a = (a^{(1)}, \ldots, a^{(n)}) \in \mathbb{R}^n$ *with scalar* $a^{(i)} \in \mathbb{R}$.
2. *Per-agent centralized critic (linear in $a$):*

$$Q_i(s, a; w_i) = \langle w_i, \phi_i(s, a) \rangle, \qquad \phi_i(s, a) = f_{\phi,i}(s) + \sum_{p=1}^{n} a^{(p)} m_i^{(p)}(s) \in \mathbb{R}^{c_i},$$

*where* $m_i^{(p)} : \mathbb{R}^d \to \mathbb{R}^{c_i}$ *and* $w_i \in \mathbb{R}^{c_i}$*. Then* $\frac{\partial \phi_i(s, a)}{\partial a^{(p)}} = m_i^{(p)}(s)$ *and* $\frac{\partial \phi_i(s,a)}{\partial \theta_p}\Big|_{a \text{ fixed}} = 0$.

3. *Per-agent actor (linear deterministic):*

$$a^{(i)} = \pi_i(s; \theta_i) \triangleq \langle \theta_i, s \rangle, \qquad \theta_i \in \mathbb{R}^d.$$

4. *Batch and unrolling:* $\mathcal{B} = \{(s^j, a^j, r^j, s'^j)\}_{j=1}^{|\mathcal{B}|}$ *with* $r^j = (r_1^j, \ldots, r_n^j)$*; next actions* $a^{(p)'j} = \langle \theta_p, s'^j \rangle$*, joint* $a'^j = (a^{(p)'j})_{p=1}^n$*. Define the shorthands*

$$\phi_i^j = \phi_i(s^j, a^j), \qquad \phi_i'^j = \phi_i(s'^j, a'^j).$$

5. *TD residual (per critic):*

$$\delta_i^j \triangleq \langle w_i, \phi_i^j \rangle - r_i^j - \gamma \langle w_i, \phi_i'^j \rangle.$$

**Per-critic $i$ TD loss and derivatives.**

$$\mathcal{L}_i(w_i) = \frac{1}{|\mathcal{B}|} \sum_{j=1}^{|\mathcal{B}|} \Big( \underbrace{\langle w_i, \phi_i(s^j, a^j) \rangle}_{\hat{y}_i^j} - \underbrace{(r_i^j + \gamma \langle w_i, \phi_i(s'^j, a'^j) \rangle)}_{y_i^j} \Big)^2$$

$$= \frac{1}{|\mathcal{B}|} \sum_{j=1}^{|\mathcal{B}|} \Big( \langle w_i, \phi_i^j \rangle - r_i^j - \gamma \langle w_i, \phi_i'^j \rangle \Big)^2. \tag{6}$$

$$F_{w_i}(w_i, \theta_{1:n}) \equiv \nabla_{w_i} \mathcal{L}_i(w_i) = \frac{2}{|\mathcal{B}|} \sum_{j=1}^{|\mathcal{B}|} \delta_i^j (\phi_i^j - \gamma \phi_i'^j) \in \mathbb{R}^{c_i}, \tag{7}$$

$$J_{w_i w_i} = \nabla_{w_i}^2 \mathcal{L}_i(w_i) = \frac{2}{|\mathcal{B}|} \sum_{j=1}^{|\mathcal{B}|} (\phi_i^j - \gamma \phi_i'^j)(\phi_i^j - \gamma \phi_i'^j)^\mathsf{T} \succeq 0 \in \mathbb{R}^{c_i \times c_i}. \tag{8}$$

Using

$$\frac{\partial \phi_i(s'^j, a'^j)}{\partial \theta_p} = \frac{\partial \phi_i(s'^j, a)}{\partial a^{(p)}}\Big|_{a=a'^j} \frac{\partial a^{(p)'j}}{\partial \theta_p} = m_i^{(p)}(s'^j)(s'^j)^\mathsf{T} \in \mathbb{R}^{c_i \times d},$$

the mixed critic–actor block is

$$J_{w_i \theta_p} = \nabla_{\theta_p} F_{w_i} = -\frac{2\gamma}{|\mathcal{B}|} \sum_{j=1}^{|\mathcal{B}|} \Big[ (\phi_i^j - \gamma \phi_i'^j)(s'^j)^\mathsf{T} \underbrace{w_i^\mathsf{T} m_i^{(p)}(s'^j)}_{\alpha_{i,p}^j} + m_i^{(p)}(s'^j)(s'^j)^\mathsf{T} \delta_i^j \Big] \in \mathbb{R}^{c_i \times d}. \tag{9}$$

**Actor derivatives (descent-style operator).** Each actor $p$ follows the negative policy gradient using its *own* critic $Q_p$:

$$\nabla_{a^{(p)}} Q_p(s, a; w_p) = w_p^\mathsf{T} m_p^{(p)}(s), \qquad \nabla_{\theta_p} \pi_p(s; \theta_p) = s.$$

Thus

$$F_{\theta_p}(w_p, \theta_p) \equiv -\nabla_{\theta_p} J^{(p)} = -\frac{1}{|\mathcal{B}|} \sum_{j=1}^{|\mathcal{B}|} s^j \langle w_p, m_p^{(p)}(s^j) \rangle \in \mathbb{R}^d, \tag{10}$$

$$J_{\theta_p \theta_q} = \nabla_{\theta_q} F_{\theta_p} = \mathbf{0}_{d \times d} \qquad \text{(for all } p, q), \tag{11}$$

$$J_{\theta_p w_i} = \nabla_{w_i} F_{\theta_p} = \begin{cases} -\dfrac{1}{|\mathcal{B}|} \sum_{j=1}^{|\mathcal{B}|} s^j \, m_p^{(p)}(s^j)^\mathsf{T} \in \mathbb{R}^{d \times c_p}, & \text{if } i = p, \\ \mathbf{0}_{d \times c_i}, & \text{if } i \neq p. \end{cases} \tag{12}$$

Hence, the actor–critic block $J_{\theta w}$ is block-diagonal across agents.

**Operator and Jacobian.** Stack parameters as $\big[ w_1; \ldots; w_n; \theta_1; \ldots; \theta_n \big] \in \mathbb{R}^{c_{\text{tot}} + nd}$ with $c_{\text{tot}} = \sum_i c_i$. Stack the operator as

$$F = \begin{bmatrix} F_{w_1} \\ \vdots \\ F_{w_n} \\ F_{\theta_1} \\ \vdots \\ F_{\theta_n} \end{bmatrix}.$$

The Jacobian has the block structure

$$J = \begin{bmatrix} J_{ww} & J_{w\theta} \\ J_{\theta w} & \mathbf{0} \end{bmatrix}, \qquad J_{ww} = \operatorname{diag}\big( J_{w_1 w_1}, \ldots, J_{w_n w_n} \big),$$

with $J_{w_i w_i}$ from (8); $J_{w\theta}$ the $c_{\text{tot}} \times nd$ block matrix whose $(i, p)$ block is (9); and $J_{\theta w}$ the $nd \times c_{\text{tot}}$ block matrix whose $(p, i)$ block is (12).

We now state the spectral result and relate it to the single-agent case (same one-step unrolling).

**Lemma 2** (Complex eigenvalues and scaling vs. single agent). *Consider Setting 2 with $\gamma > 0$ and the Jacobian*

$$J = \begin{bmatrix} J_{ww} & J_{w\theta} \\ J_{\theta w} & 0 \end{bmatrix}.$$

*Let $B_{i,p} \triangleq J_{w_i \theta_p} \in \mathbb{R}^{c_i \times d}$ and $C_{p,i} \triangleq J_{\theta_p w_i} \in \mathbb{R}^{d \times c_i}$; define*

$$\mathcal{M}_n \triangleq J_{w\theta} J_{\theta w} = \sum_{p=1}^{n} \begin{bmatrix} B_{1,p} \\ \vdots \\ B_{n,p} \end{bmatrix} \begin{bmatrix} C_{p,1} & \cdots & C_{p,n} \end{bmatrix} \in \mathbb{R}^{c_{\text{tot}} \times c_{\text{tot}}}.$$

*Then:*

(i) *(Pure cross-term case) If $J_{ww} = \mathbf{0}$, then*

$$\operatorname{spec}(J) = \Big\{ \pm \sqrt{\lambda_k(\mathcal{M}_n)} \Big\}_k.$$

*In particular, if $\mathcal{M}_n$ has a negative eigenvalue, $J$ has a purely imaginary conjugate pair. Because $B_{i,p}$ couples critic $i$ with* every *actor $p$ via the next-state joint action, while $C_{p,i}$ is nonzero only for $i = p$ (actor $p$ uses its own critic $p$), $\mathcal{M}_n$ is generically sign-indefinite, hence complex pairs are typical.*

(ii) *(Persistence) If $\mathcal{M}_n$ has a negative eigenvalue, there exists $t_0 > 0$ such that for all $0 \leqslant t < t_0$*

$$J(t) = \begin{bmatrix} t\, J_{ww} & J_{w\theta} \\ J_{\theta w} & 0 \end{bmatrix}$$

*has a non-real conjugate eigenpair.*

*(iii) (*Relation to single agent*) Let $\mathcal{M}_1$ denote the single-agent product (with one $\boldsymbol{s}'$ unroll). Then*

$$\mathcal{M}_n = \mathcal{M}_1 + \sum_{p=2}^{n} \begin{bmatrix} B_{1,p} \\ \vdots \\ B_{n,p} \end{bmatrix} \begin{bmatrix} C_{p,1} & \cdots & C_{p,n} \end{bmatrix}.$$

*Hence*

$$\max_{p} \rho\left( \begin{bmatrix} B_{1,p} \\ \vdots \\ B_{n,p} \end{bmatrix} \begin{bmatrix} C_{p,1} & \cdots & C_{p,n} \end{bmatrix} \right) \leqslant \rho(\mathcal{M}_n) \leqslant \sum_{p=1}^{n} \left\| \begin{bmatrix} B_{1,p} \\ \vdots \\ B_{n,p} \end{bmatrix} \right\| \left\| \begin{bmatrix} C_{p,1} & \cdots & C_{p,n} \end{bmatrix} \right\|,$$

*so in the pure cross-term case*

$$\rho(J) = \sqrt{\rho(\mathcal{M}_n)} \geqslant \max_{p} \sqrt{\rho\left( \begin{bmatrix} B_{1,p} \\ \vdots \\ B_{n,p} \end{bmatrix} \begin{bmatrix} C_{p,1} & \cdots & C_{p,n} \end{bmatrix} \right)}.$$

*Under alignment (shared invariant directions / commuting contributions / PSD terms) the spectral radius grows with $n$ (at least as the square root of the sum of aligned eigenvalues).*

*(iv) (*Triangular/degenerate real-spectrum cases*) If $J_{\boldsymbol{w}_i \boldsymbol{\theta}_p} = \boldsymbol{0}$ for all $i, p$ (e.g., $\gamma = 0$, or $\delta_i^j \equiv 0$ and $\boldsymbol{w}_i^{\mathsf{T}} m_i^{(p)}(\boldsymbol{s}'^j) \equiv 0$), $J$ is block upper triangular with real spectrum $\mathrm{spec}(J_{\boldsymbol{ww}}) \cup \{0\}$. If instead $J_{\boldsymbol{\theta}_p \boldsymbol{w}_i} = \boldsymbol{0}$ for all $p, i$ (e.g., $m_p^{(p)} \equiv 0$), $J$ is block lower triangular, again yielding a real spectrum. If $J_{\boldsymbol{ww}} = J_{\boldsymbol{ww}}^{\mathsf{T}} \succeq 0$ and $J_{\boldsymbol{w}_i \boldsymbol{\theta}_p} = (J_{\boldsymbol{\theta}_p \boldsymbol{w}_i})^{\mathsf{T}}$ for all $i, p$, then $J$ is symmetric and all eigenvalues are real (stringent/degenerate off-policy conditions).*

*Proof sketch.* (i) With $J_{\boldsymbol{ww}} = 0$, $J^2 = \mathrm{diag}\big(\mathcal{M}_n, \ J_{\boldsymbol{\theta w}} J_{\boldsymbol{w \theta}}\big)$, so eigenvalues of $J$ are $\pm\sqrt{\lambda_k(\mathcal{M}_n)}$. (ii) Analytic perturbation theory implies a simple imaginary pair at $t = 0$ cannot become real for small $t > 0$ without crossing the real axis. (iii) Bounds follow from subadditivity of the operator norm and $\rho(\sum_p X_p) \geqslant \max_p \rho(X_p)$ for square matrices $\{X_p\}$ of common size; alignment/commutation yields additive growth of dominant eigenvalues, hence $\rho(\mathcal{M}_n)$ (and thus $\rho(J)$) increases with $n$. (iv) Triangular and symmetric cases yield real spectra as stated. $\square$ $\square$

*Remark* 2 (How adding agents changes the Jacobian). Relative to the single-agent Jacobian: (a) $J_{\boldsymbol{ww}}$ becomes block-diagonal across critics; (b) $J_{\boldsymbol{w \theta}}$ gains $n$ columns of blocks $J_{\boldsymbol{w}_i \boldsymbol{\theta}_p}$ since each critic $i$ depends on the *full* next-state joint action; (c) $J_{\boldsymbol{\theta w}}$ becomes block-diagonal because actor $p$ depends only on its own critic $p$. These changes enlarge $\mathcal{M}_n = J_{\boldsymbol{w \theta}} J_{\boldsymbol{\theta w}}$ by a sum of $n$ player-specific terms, making negative eigenvalues (and thus complex pairs) more likely and typically *increasing* the spectral radius compared to the single-agent case with one $\boldsymbol{s}'$ unroll.

*Remark* 3 (Sign convention). We use a descent-style operator for both critic and actors, matching the single-agent section. Flipping all actor signs multiplies $J_{\boldsymbol{\theta w}}$ by $-1$ and leaves conclusions about non-symmetry and complex pairs unchanged, since they depend on $J_{\boldsymbol{w \theta}} J_{\boldsymbol{\theta w}}$.

## D    VI MARL Convergence, Perspectives & Details on the Proposed Algorithms

In this section we extend our discussion on the convergence on VI-MARL operator, then we present the VI operators of additional MARL algorithms within the centralized critic CTDE paradigm. After, we provide detailed versions of Algorithm 1 for MADDPG and MATD3, outlining the full training process when incorporating LA or LA-EG.

### D.1    VI MARL Convergence

We first recall the abstract multi-player operator definition from Appendix B.1. Each agent $i \in [n]$ aims to optimize its objective $f_i \colon \mathbb{R}^d \to \mathbb{R}$, which, in the general case, depends on all players' strategies. Then, we have the following operator $F$:

$$F_{n\text{-agents}}(\boldsymbol{z}) \equiv \begin{bmatrix} \nabla_{\boldsymbol{z}_1} f_1(\boldsymbol{z}) \\ \vdots \\ \nabla_{\boldsymbol{z}_n} f_n(\boldsymbol{z}) \end{bmatrix}, \tag{$F_{n\text{-agents}}$}$$

with the game Jacobian as follows:

$$J_{n\text{-agents}}(\boldsymbol{z}) \equiv \begin{bmatrix} \nabla^2_{\boldsymbol{z}_1^2} f_1(\boldsymbol{z}) & \nabla^2_{\boldsymbol{z}_1 \boldsymbol{z}_2} f_1(\boldsymbol{z}) & \dots & \nabla^2_{\boldsymbol{z}_1 \boldsymbol{z}_n} f_1(\boldsymbol{z}) \\ \vdots & \vdots & \dots & \vdots \\ \nabla^2_{\boldsymbol{z}_n \boldsymbol{z}_1} f_n(\boldsymbol{z}) & \nabla^2_{\boldsymbol{z}_n \boldsymbol{z}_2} f_n(\boldsymbol{z}) & \dots & \nabla^2_{\boldsymbol{z}_n^2} f_n(\boldsymbol{z}) \end{bmatrix}. \tag{$J_{n\text{-agents}}$}$$

More precisely, for multi-agent actor-critic RL we have the following operator:

$$F_{\text{MAAC}}\left( \begin{bmatrix} \vdots \\ \boldsymbol{w}_i \\ \boldsymbol{\theta}_i \\ \vdots \end{bmatrix} \right) \equiv \begin{bmatrix} \vdots \\ \nabla_{\boldsymbol{w}_i}\left( \frac{1}{|\mathcal{B}|} \sum_{j=1}^{|\mathcal{B}|} \ell_i^{\boldsymbol{w}}(\cdot; \boldsymbol{w}_i, \boldsymbol{\theta}) \right) \\ \nabla_{\boldsymbol{\theta}_i}\left( \frac{1}{|\mathcal{B}|} \sum_{j=1}^{|\mathcal{B}|} \ell_i^{\boldsymbol{\theta}}(\cdot; \boldsymbol{w}_i, \boldsymbol{\theta}_i) \right) \\ \vdots \end{bmatrix}, \tag{$F_{\text{MAAC}}$}$$

where the parameter space is $\mathcal{Z} \equiv \mathbb{R}^d$, with $d = \sum_{i=1}^{n}(d_i^Q + d_i^\mu)$; and *MAAC* stands for *multi-agent-actor-critic*.

Then, we can notice by computing the Jacobian of the above operator that the eigenvalues are in the complex plane. Applying lookahead results in interpolating the largest eigenvalue (in magnitude) with the point (1,0) in the complex plane, thus reducing the spectral radius of the Jacobian. Furthermore, applying this recursively (nested Lookahead) leads to larger contraction.

To make this more precise, consider the gradient descent operator as a base optimizer

$$T_{GD} \equiv I - \alpha F,$$

where $\alpha$ is the step size vector.

Let $\lambda$ denote the eigenvalue of $J^{base} \triangleq \nabla T_{GD}(\cdot)$ with largest modulus, *i.e.* $\rho(J^{base}(\cdot) = |\lambda|$, let $\boldsymbol{u}$ be its associated eigenvector: $J^{base}\boldsymbol{u} = \lambda\boldsymbol{u}$.

The Jacobian of Lookahead is then:

$$J^{LA} = \nabla F^{LA}(\cdot) = (1 - \alpha)I + \alpha(J^{base})^k.$$

The power $k$ rotates the eigenvector in the complex plane; see (Chavdarova et al., 2021). By noticing that:

$$J^{LA}\boldsymbol{u} = ((1 - \alpha)I + \alpha(J^{base})^k)\boldsymbol{u}$$
$$= ((1 - \alpha) + \alpha\lambda^k)\boldsymbol{u},$$

we deduce $\boldsymbol{u}$ is an eigenvector of $J^{LA}$ with eigenvalue $1 - \alpha + \alpha\lambda^k$. Thus, this is strictly closer to the unit ball in the complex plane, increasing the contractiveness.

## D.2 VI MARL PERSPECTIVES

In the main text, we introduced the general VI operator for multi-agent actor-critic algorithms ($F_{\text{MAAC}}$) and provided the specific equations for MADDPG in ($\ell^{\boldsymbol{w}}_{\text{MADDPG}}$ & $\ell^{\boldsymbol{\theta}}_{\text{MADDPG}}$), with the operator corresponding to:

$$
F_{\text{MADDPG}}\left(\begin{bmatrix} \vdots \\ \boldsymbol{w}_i \\ \boldsymbol{\theta}_i \\ \vdots \end{bmatrix}\right) \equiv \begin{bmatrix} \vdots \\ \nabla_{\boldsymbol{w}_i}\left(\frac{1}{|\mathcal{B}|}\sum_{j=1}^{|\mathcal{B}|}\left(r_i^j + \gamma\mathbf{Q}_i^{\bar{\boldsymbol{\mu}}}\left(\boldsymbol{x}'^j, \boldsymbol{a}'; \boldsymbol{w}_i'\right)\big|_{\boldsymbol{a}'=\bar{\boldsymbol{\mu}}(\boldsymbol{o}'^j)} - \mathbf{Q}_i^{\boldsymbol{\mu}}(\boldsymbol{x}^j, \boldsymbol{a}^j; \boldsymbol{w}_i)\right)^2\right) \\ \nabla_{\boldsymbol{\theta}_i}\left(\frac{1}{|\mathcal{B}|}\sum_{j=1}^{|\mathcal{B}|}\boldsymbol{\mu}_i(\boldsymbol{o}_i^j; \boldsymbol{\theta}_i)\nabla_{a_i}\ \mathbf{Q}_i^{\boldsymbol{\mu}}(\boldsymbol{x}^j, a_1^j, \ldots, a_i, \ldots, a_n^j; \boldsymbol{w}_i)\big|_{a_i=\boldsymbol{\mu}_i(\boldsymbol{o}_i^j)}\right) \\ \vdots \end{bmatrix},
$$

$$(F_{\text{MADDPG}})$$

where the parameter space is $\mathcal{Z} \equiv \mathbb{R}^d$, with $d = \sum_{i=1}^n(d_i^Q + d_i^{\mu})$.

We now show how update equations for several well-known MARL algorithms—that follow the CTDE paradigm with a centralized critic—can be written as a VI. Our VI-based methods can also be applied to these algorithms using the operators below.

For a more general notation, for each agent $i \in [n]$ we assume: *(i)* central critic network (one or multiple) that estimates either action value $Q$–Network$(\boldsymbol{s}, \boldsymbol{a})$: $\mathbf{Q}_i(\boldsymbol{x}_t, \boldsymbol{a}_t; \boldsymbol{w}_i)$, or state value $V$–network(s): $\mathbf{V}_i(\boldsymbol{x}_t; \boldsymbol{w}_i)$, and *(ii)* a decentralized policy network that can be deterministic $\boldsymbol{\mu}_i(\boldsymbol{o}_i; \boldsymbol{\theta}_i)$ or stochastic $\boldsymbol{\pi}_i(\boldsymbol{o}_i; \boldsymbol{\theta}_i)$, depending on the algorithm. Given a batch of experiences $\mathcal{B}$: $(\boldsymbol{x}^j, \boldsymbol{a}^j, \mathbf{r}^j, \boldsymbol{x}'^j)$, sampled from a replay buffer $(\mathcal{D})$, we provide the necessary equations and the final operator $(F)$ for each of the following popular MARL algorithms.

### D.2.1 MATD3

The VI formulation for MATD3 is very similar to MADDPG, except here, for each agent, we have two critic networks; we write: $\boldsymbol{w}_i \equiv \{\boldsymbol{w}_{i,1}, \boldsymbol{w}_{i,2}\}$. Accordingly, target computation for the critic $(Q_{i,m})$ is calculated by taking the minimum of both critic networks, but only the value of critic 1 is used for the actor (policy network) update. We have:

$$
F_{\text{MATD3}}\left(\begin{bmatrix} \vdots \\ \boldsymbol{w}_{i,1} \\ \boldsymbol{w}_{i,2} \\ \boldsymbol{\theta}_i \\ \vdots \end{bmatrix}\right) \equiv
$$

$$
\begin{bmatrix}
\vdots \\
\nabla_{\boldsymbol{w}_{i,1}}\left(\frac{1}{|\mathcal{B}|}\sum_{j=1}^{|\mathcal{B}|}\left(\underbrace{r_i^j + \gamma\min_{m\in\{1,2\}}\mathbf{Q}_{i,m}^{\bar{\boldsymbol{\mu}}}\left(\boldsymbol{x}'^j, a_1', \ldots, a_n'\right)\big|_{\boldsymbol{a}'=\bar{\boldsymbol{\mu}}(\boldsymbol{o}'^j)}}_{\text{target } y_i} - \mathbf{Q}_{i,1}^{\boldsymbol{\mu}}(\boldsymbol{x}^j, \boldsymbol{a}^j; \boldsymbol{w}_{i,1})\right)^2\right) \\
\nabla_{\boldsymbol{w}_{i,2}}\left(\frac{1}{|\mathcal{B}|}\sum_{j=1}^{|\mathcal{B}|}\left(\underbrace{r_i^j + \gamma\min_{m\in\{1,2\}}\mathbf{Q}_{i,m}^{\bar{\boldsymbol{\mu}}}\left(\boldsymbol{x}'^j, a_1', \ldots, a_n'\right)\big|_{\boldsymbol{a}'=\bar{\boldsymbol{\mu}}(\boldsymbol{o}'^j)}}_{\text{target } y_i} - \mathbf{Q}_{i,2}^{\boldsymbol{\mu}}(\boldsymbol{x}^j, \boldsymbol{a}^j; \boldsymbol{w}_{i,2})\right)^2\right) \\
\left(\frac{1}{|\mathcal{B}|}\sum_{j=1}^{|\mathcal{B}|}\nabla_{\boldsymbol{\theta}_i}\boldsymbol{\mu}_i(\boldsymbol{o}_i^j; \boldsymbol{\theta}_i)\nabla_{a_i}\ \mathbf{Q}_{i,1}^{\boldsymbol{\mu}}(\boldsymbol{x}^j, a_1^j, \ldots, a_i, \ldots, a_n^j)\big|_{a_i=\boldsymbol{\mu}_i(\boldsymbol{o}_i^j)}\right) \\
\vdots
\end{bmatrix}.
$$

$$(F_{\text{MATD3}})$$

### D.2.2 COMA

In COMA, critic is trained using a $TD(\lambda)$ target $(y^\lambda)$ computed from a target network parameterized by $\bar{\boldsymbol{w}}$ that get updated to main network weights every couple iterations. Given the following Advantage $A_i$ calculations:

$$A_i(\boldsymbol{x}, \boldsymbol{a}) = Q(\boldsymbol{x}, \boldsymbol{a}) - b_i(\boldsymbol{x}, \boldsymbol{a}_{-i})$$

$$b_i(\boldsymbol{x}, \boldsymbol{a}_{-i}) = \sum_{a_i} \boldsymbol{\pi}_i(a_i|\boldsymbol{o}_i) Q(\boldsymbol{x}, (a_i, \boldsymbol{a}_{-i})),$$

the operator for *COMA* corresponds to:

$$F_{\text{COMA}}\left(\begin{bmatrix} \vdots \\ \boldsymbol{w}_i \\ \boldsymbol{\theta}_i \\ \vdots \end{bmatrix}\right) \equiv \begin{bmatrix} \vdots \\ \nabla_{\boldsymbol{w}_i} \mathbb{E}\left[\left(y_i^\lambda - Q_i(\boldsymbol{x}^j, \boldsymbol{a}; \boldsymbol{w}_i)\right)^2\right] \\ \mathbb{E}\left[\nabla_{\boldsymbol{\theta}_i} \sum_i A_i(\boldsymbol{x}, \boldsymbol{a}) \log \boldsymbol{\pi}_{\boldsymbol{\theta}_i}(a_i|\boldsymbol{o}_i)\right] \\ \vdots \end{bmatrix}. \quad (F_{\text{COMA}})$$

### D.2.3 MAPPO

As previously noted, MAPPO can be seen as a simplified version of MATRPO. It shares a similar critic loss with MATRPO but simplifies the actor loss by using a clipped objective instead of a KL constraint, making the optimization problem more tractable. This allows it to be formulated as a VI, as shown below:

$$\hat{V}_t = (1-\lambda) \sum_{n=1}^T \lambda^{n-1} \left(\sum_{k=0}^{n-1} \gamma^k r_{t+k} + \gamma^n \mathbf{V}(\boldsymbol{o}_i')\right),$$

$$F_{\text{MAPPO}}\left(\begin{bmatrix} \vdots \\ \boldsymbol{w}_i \\ \boldsymbol{\theta}_i \\ \vdots \end{bmatrix}\right) \equiv \begin{bmatrix} \vdots \\ \nabla_{\boldsymbol{w}_i} \mathbb{E}\left[\left(\mathbf{V}(\boldsymbol{x}; \boldsymbol{w}_i) - \hat{V}_t\right)^2\right] \\ \nabla_{\boldsymbol{\theta}_i} \mathbb{E}\left[\min\left\{\frac{\boldsymbol{\pi}_{\boldsymbol{\theta}_i}(a_i|\boldsymbol{o}_i)}{\boldsymbol{\pi}_{\boldsymbol{\theta}_i^{old}}(a_i|\boldsymbol{o}_i)} A_i^{\theta_i^{old}}, \text{clip}\left(\frac{\boldsymbol{\pi}_{\boldsymbol{\theta}_i}(a_i|\boldsymbol{o}_i)}{\boldsymbol{\pi}_{\boldsymbol{\theta}_i^{old}}(a_i|\boldsymbol{o}_i)}, 1-\epsilon, 1+\epsilon\right) A_i^{\theta_i^{old}}\right\}\right] \\ \vdots \end{bmatrix}. \quad (F_{\text{MAPPO}})$$

### D.3 DETAILED ALGORITHMS

Herein we provide procedure NestedLookahead called from algorithm 1 to compute the extrapolations and after present two pseudocodes considered as extended versions of the main algorithm in algorithm 1; in which we detail how the lookahead approach can be integrated in the training process of MADDPG and MATD3.

#### D.3.1 NESTED LOOKAHEAD ALGORITHM

In algorithm 6 below we share a detailed version of Nested lookahead procedure called from algorithms 1, 7 and 8.

#### D.3.2 EXTENDED VERSION OF LA-MADDPG PSEUDOCODE

We include an extended version for the LA-MADDPG algorithm without VI notations in algorithm 7.

#### D.3.3 EXTENDED VERSION OF LA-MATD3 PSEUDOCODE

We include an extended version for the LA-MATD3 algorithm without VI notations in algorithm 8.

---

**Algorithm 6** Pseudocode for LA-VI, called from Algorithm 1. Updates the parameters in-place.

---

1: **procedure** NESTEDLOOKAHEAD:
2:  **Input:**  #agents $n$, episode counter $e$, actor and critic weights and snapshots: $\{(\boldsymbol{\theta}_i, \boldsymbol{\theta}_i^{(1)}, \ldots, \boldsymbol{\theta}_i^{(l)})\}_{i=1}^n$ and $\{(\boldsymbol{w}_i, \boldsymbol{w}_i^{(1)}, \ldots, \boldsymbol{w}_i^{(l)})\}_{i=1}^n$, LA hyperparameters: levels $l$, $(k^{(1)}, \ldots, k^{(l)})$ and $(\alpha_{\boldsymbol{\theta}}, \alpha_{\boldsymbol{w}})$.
3:  **for all** $j \in [l]$ **do**
4:    **if** $e\%k^{(j)} == 0$ **then**
5:      **for all** agent $i \in [n]$ **do**
6:        $\boldsymbol{w}_i \leftarrow \boldsymbol{w}_i^{(j)} + \alpha_{\boldsymbol{w}}(\boldsymbol{w}_i - \boldsymbol{w}_i^{(j)})$                    *LA $j^{th}$ level*
7:        $\boldsymbol{\theta}_i \leftarrow \boldsymbol{\theta}_i^{(j)} + \alpha_{\boldsymbol{\theta}}(\boldsymbol{\theta}_i - \boldsymbol{\theta}_i^{(j)})$
8:        $(\boldsymbol{\theta}_i^{(1)}, \ldots, \boldsymbol{\theta}_i^{(j)}, \boldsymbol{w}_i^{(1)}, \ldots, \boldsymbol{w}_i^{(j)}) \leftarrow (\{\boldsymbol{\theta}_i\}_{\times j}, \{\boldsymbol{w}_i\}_{\times j})$        *Update copies up to $j^{th}$*
9:      **end for**
10:   **end if**
11:  **end for**
12: **end procedure**

---

## E   DETAILS ON THE IMPLEMENTATION

We used the configurations and hyperparameters from the original MADDPG paper for our implementation. For completeness, these are listed in Table 2. We ran $t = 60000$ training episodes for all environments, with a maximum of 25 environment steps ($step$) per episode.

In all experiments, we used a 2-layer MLP with $64$ units per layer. ReLU activation was applied between layers for both the policy and value networks of all agents.

### E.1   HYPERPARAMETER SELECTION FOR LOOKAHEAD

In this section, we discuss and share guidelines for hyperparameter selection based on our experiments.

**Summary.**

- We observed two- or three-level of Lookahead outperform single-level Lookahead (figure 6).

- Each level $j \in [l]$ has different $k$, denoted here with $k^{(j)}$. These should be selected as multiple of the selected $k$ for the level before, that is, $k^{(j)} = c_j \cdot k^{(j-1)}$, where $c_j$ is positive integer.

- We observed that for the innermost lookahead, small values for $k^{(1)}$, such as smaller than or equal to 50, perform better than using large values. For the outer $k^{(j)}, j > 1$ large values work well, such as in the range between $5 - 10$ for the $c_j$,.

- We typically used $\alpha = 0.5$, and we observed lower values, such as $\alpha = 0.3$, give better performances then $\alpha > 0.5$.

**Discussion.**

- To give an intuition regarding the above-listed conclusions, small values for $k^{(1)}$ help because the MARL setting is very noisy and the vector field is rotational. If large values are used for $k_s$, then the algorithm will diverge away. It is known that the combination of noise and rotational vector field can cause methods to diverge away (Chavdarova et al., 2019).

- Relative to the analogous conclusions for GANs (Chavdarova et al., 2021), the differences is that:

  - The better-performing values for $k^{(1)}$ are of a similar range as for Lookahead with GD for GANs; however they are smaller than those used for Lookahead with EG for GANs.

**Algorithm 7** Pseudocode for LA–MADDPG: MADDPG with (Nested) Lookahead.

---

1: **Input:** Environment $\mathcal{E}$, number of agents $n$, number of episodes $t$, action spaces $\{\mathcal{A}_i\}_{i=1}^n$, number of random steps $t_{\text{rand}}$ before learning, learning interval $t_{\text{learn}}$, actor networks $\{\boldsymbol{\mu}_i\}_{i=1}^n$, with initial weights $\boldsymbol{\theta} \equiv \{\boldsymbol{\theta}_i\}_{i=1}^n$, critic networks $\{\mathbf{Q}_i\}_{i=1}^n$ with initial weights $\boldsymbol{w} \equiv \{\boldsymbol{w}_i\}_{i=1}^n$, learning rates $\eta_{\boldsymbol{\theta}}, \eta_{\boldsymbol{w}}$, base optimizer $B$ (e.g., Adam), discount factor $\gamma$, lookahead hyperparameters $\mathcal{L} \equiv (l, \{k^{(j)}\}_{j=1}^l, \alpha_{\boldsymbol{\theta}}, \alpha_{\boldsymbol{w}})$, soft update parameter $\tau$.
2: **Initialize:**
3:     Replay buffer $\mathcal{D} \leftarrow \varnothing$
4:     LA parameters: $\boldsymbol{\phi} \leftarrow \{\boldsymbol{\theta}\}_{\times l}, \{\boldsymbol{w}\}_{\times l}$                     *(store snapshots for LA)*
5: **for all** episode $e \in 1, \ldots, t$ **do**
6:     $\boldsymbol{x} \leftarrow Sample(\mathcal{E})$                     *(sample from environment $\mathcal{E}$)*
7:     $step \leftarrow 1$
8:     **repeat**
9:         **if** $e \le t_{\text{rand}}$ **then**
10:             for each agent $i$, $a_i \sim \mathcal{A}_i$                     *(sample actions randomly)*
11:         **else**
12:             for each agent $i$, select action $a_i$ using current policy and exploration
13:         **end if**
14:                             *(apply actions and record results)*
15:         Execute actions $\boldsymbol{a} = (a_1, \ldots, a_n)$, observe rewards $\mathbf{r}$ and new state $\boldsymbol{x}'$
16:         replay buffer $\mathcal{D} \leftarrow (\boldsymbol{x}, \boldsymbol{a}, \mathbf{r}, \boldsymbol{x}')$
17:         $\boldsymbol{x} \leftarrow \boldsymbol{x}'$
18:                             *(apply learning step if applicable)*
19:         **if** $step \% t_{\text{learn}} = 0$ **then**
20:             **for all** agents $i \in 1, \ldots, n$ **do**
21:                 sample batch $\{(\boldsymbol{x}^j, \boldsymbol{a}^j, \mathbf{r}^j, \boldsymbol{x}'^j)\}_{j=1}^{|\mathcal{B}|}$ from $\mathcal{D}$
22:                 $y^j \leftarrow r_i^j + \gamma \mathbf{Q}^{\bar{\boldsymbol{\mu}}}(\boldsymbol{x}'^j, a_1', \ldots, a_n')$, where $a_k' = \bar{\boldsymbol{\mu}}_k(\boldsymbol{o}_k'^j)$
23:                 Update critic by minimizing the loss $\ell(\boldsymbol{w}_i) = \frac{1}{|\mathcal{B}|} \sum_j \left( y^j - \mathbf{Q}_i^{\boldsymbol{\mu}}(\boldsymbol{x}^j, a_1^j, \ldots, a_n^j) \right)^2$ using $B$
24:                 Update actor policy using policy gradient formula $B$
25:                 $\nabla_{\boldsymbol{\theta}_i} J \approx \frac{1}{|\mathcal{B}|} \sum_j \nabla_{\boldsymbol{\theta}_i} \boldsymbol{\mu}_i(\boldsymbol{o}_i^j) \nabla_{a_i} \mathbf{Q}_i^{\boldsymbol{\mu}}(\boldsymbol{x}^j, a_1^j, \ldots, a_i, \ldots, a_n^j)$, where $a_i = \boldsymbol{\mu}_i(\boldsymbol{o}_i^j)$
26:             **end for**
27:             **for all** agents $i \in [n]$ **do**
28:                 $\bar{\boldsymbol{\theta}}_i \leftarrow \tau \boldsymbol{\theta}_i + (1 - \tau) \bar{\boldsymbol{\theta}}_i$                     *(update target networks)*
29:                 $\bar{\boldsymbol{w}}_i \leftarrow \tau \boldsymbol{w}_i + (1 - \tau) \bar{\boldsymbol{w}}_i$
30:             **end for**
31:         **end if**
32:         $step \leftarrow step + 1$
33:     **until** environment terminates
34:     NESTEDLOOKAHEAD$(n, e, \boldsymbol{\phi}, \mathcal{L})$
35: **end for**
36: **Output:** $\boldsymbol{\theta}, \boldsymbol{w}$

---

**Algorithm 8** Pseudocode for LA–MATD3: MATD3 with (Nested) Lookahead.

1: **Input:** Environment $\mathcal{E}$, number of agents $n$, number of episodes $t$, action spaces $\{\mathcal{A}_i\}_{i=1}^n$, number of random steps $t_{\mathrm{rand}}$ before learning, learning interval $t_{\mathrm{learn}}$, actor networks $\{\boldsymbol{\mu}_i\}_{i=1}^n$, with initial weights $\boldsymbol{\theta} \equiv \{\boldsymbol{\theta}_i\}_{i=1}^n$, both critic networks, $\{\mathbf{Q}_{i,1}, \mathbf{Q}_{i,2}\}_{i=1}^n$ with initial weights $\boldsymbol{w} \equiv \{\boldsymbol{w}_{i,1}, \boldsymbol{w}_{i,2}\}_{i=1}^n$, learning rates $\eta_{\boldsymbol{\theta}}, \eta_{\boldsymbol{w}}$, base optimizer $B$ (e.g., Adam), discount factor $\gamma$, lookahead hyperparameters $\mathcal{L} \equiv (l, \{k^{(j)}\}_{j=1}^l, \alpha_{\boldsymbol{\theta}}, \alpha_{\boldsymbol{w}})$, soft update parameter $\tau$, policy update frequency $p$.

2: **Initialize:**

3:    Replay buffer $\mathcal{D} \leftarrow \varnothing$

4:    LA parameters: $\boldsymbol{\phi} \leftarrow \{\boldsymbol{\theta}\}_{\times l}, \{\boldsymbol{w}\}_{\times l}$         *(store snapshots for LA)*

5: **for all** episode $e \in 1, \ldots, t$ **do**

6:    $\boldsymbol{x} \leftarrow Sample(\mathcal{E})$         *(sample from environment $\mathcal{E}$)*

7:    $step \leftarrow 1$

8:    **repeat**

9:       **if** $e \le t_{\mathrm{rand}}$ **then**

10:          for each agent $i$, $a_i \sim \mathcal{A}_i$         *(sample actions randomly)*

11:       **else**

12:          for each agent $i$, select action $a_i$ using current policy and exploration

13:       **end if**

14:         *(apply actions and record results)*

15:       Execute actions $\boldsymbol{a} = (a_1, \ldots, a_n)$, observe rewards $\mathbf{r}$ and new state $\boldsymbol{x}'$

16:       replay buffer $\mathcal{D} \leftarrow (\boldsymbol{x}, \boldsymbol{a}, \mathbf{r}, \boldsymbol{x}')$

17:       $\boldsymbol{x} \leftarrow \boldsymbol{x}'$

18:         *(apply learning step if applicable)*

19:       **if** $step\%t_{\mathrm{learn}} = 0$ **then**

20:          **for all** agent $i \in [n]$ **do**

21:             sample batch $\{(\boldsymbol{x}^j, \boldsymbol{a}^j, \mathbf{r}^j, \boldsymbol{x}'^j)\}_{j=1}^{|\mathcal{B}|}$ from $\mathcal{D}$

22:             $y^j \leftarrow r_i^j + \gamma \min_{m=1,2} \mathbf{Q}_{i,l}^{\bar{\boldsymbol{\mu}}}(\boldsymbol{x}'^j, a_1', \ldots, a_n')$, where $a_k' = \bar{\boldsymbol{\mu}}_k(\boldsymbol{o}_k'^j) + \epsilon$

23:             Update both critics, $m = 1, 2$ by minimizing the loss (using optimizer $B$):
$$\ell(\boldsymbol{w}_{i,m}) = \tfrac{1}{|\mathcal{B}|} \sum_j \left( y^j - \mathbf{Q}_{i,m}^{\boldsymbol{\mu}}(\boldsymbol{x}^j, a_1^j, \ldots, a_n^j) \right)^2$$

24:             **if** $step\%p = 0$ **then**

25:                Update actor policy using policy gradient formula and optimizer $B$

26:                $\nabla_{\boldsymbol{\theta}_i} J \approx \frac{1}{|\mathcal{B}|} \sum_j \nabla_{\boldsymbol{\theta}_i} \boldsymbol{\mu}_i(\boldsymbol{o}_i^j) \nabla_{a_i} \mathbf{Q}_{i,1}^{\boldsymbol{\mu}}(\boldsymbol{x}^j, a_1^j, \ldots, a_i, \ldots, a_n^j)$, where $a_i = \boldsymbol{\mu}_i(\boldsymbol{o}_i^j)$

27:                $\bar{\boldsymbol{\theta}}_i \leftarrow \tau \boldsymbol{\theta}_i + (1 - \tau)\bar{\boldsymbol{\theta}}_i$         *(update target networks)*

28:                $\bar{\boldsymbol{w}}_{i,m} \leftarrow \tau \boldsymbol{w}_{i,m} + (1 - \tau)\bar{\boldsymbol{w}}_{i,m}$

29:             **end if**

30:          **end for**

31:       **end if**

32:       $step \leftarrow step + 1$

33:    **until** environment terminates

34:    NESTEDLOOKAHEAD$(n, e, \boldsymbol{\phi}, \mathcal{L})$

35: **end for**

36: **Output:** $\boldsymbol{\theta}, \boldsymbol{w}$

Table 2: Hyperparameters used for LA-MADDPG experiments.

| Name | Description |
|------|-------------|
| Adam $lr$ | 0.01 |
| Adam $\beta_1$ | 0.9 |
| Adam $\beta_2$ | 0.999 |
| Batch-size | 1024 |
| Update ratio $\tau$ | 0.01 |
| Discount factor $\gamma$ | 0.95 |
| Replay Buffer | $1.5 \times 10^6$ |
| learning step $t_{\text{learn}}$ | 100 |
| $t_{\text{rand}}$ | 1024 |
| Policy update ratio (MATD3) $p$ | 2 |
| Noise std (MATD3) | 0.2 |
| Noise clip (MATD3) | 0.5 |
| Lookahead $\alpha$ | 0.5 |

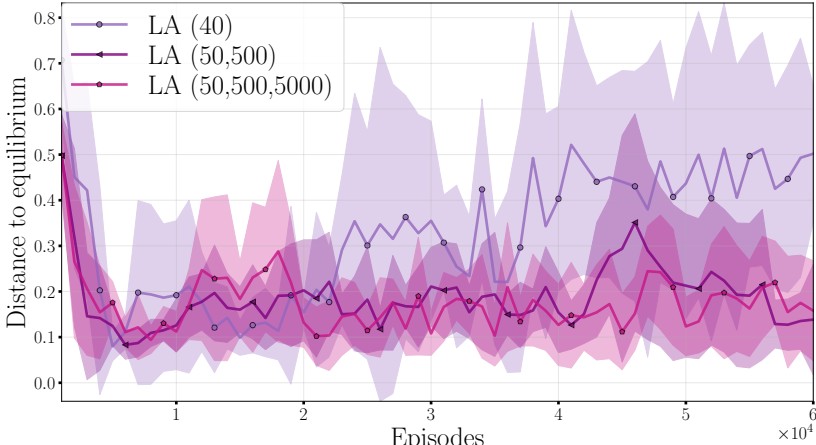

Figure 6: **Comparison of LA-MATD3 with different levels in Rock–paper–scissors**. $x$-axis: training episodes. $y$-axis: 5-seed average norm between the two players' policies and equilibrium policy $(\frac{1}{3}, \frac{1}{3}, \frac{1}{3})^2$.

### E.2 COMPUTE RESOURCES

We ran the multi-agent experiments (RPS, MP, MPE) on Google Colab enterprise using an e2-standard-8 type machine with 100 GB Standard disk (pd-standard).

## F ADDITIONAL EMPIRICAL RESULTS

### F.1 MPE: PHYSICAL DECEPTION: COOPERATIVE-COMPETITIVE ENVIRONMENT

In *Physical deception*, we have $p$ good agents, one adversary, and $p$ landmarks, with one landmark designated as the *target*. Good agents aim to get close to the target landmark while misleading the adversary, which must infer the target's location. Unlike Predator-Prey, this environment does not

| Method | Adversary Win Rate |
|--------|-------------------|
| Baseline | $0.45 \pm .16$ |
| LA-MADDPG | $0.53 \pm .11$ |
| EG-MADDPG | $0.56 \pm .27$ |
| LA-EG-MADDPG | $\mathbf{0.51 \pm .14}$ |

Table 3: *Equilibrium reached?* Average adversary win rate for MPE: Physical deception on 100 test environments. The *win rate* is the fraction of times the adversary was closer to the target at the end of episode. *Closer to* 0.5 *is better.* Refer to Section 5.2.

involve direct competition for the adversary–its reward depends solely on its own policy. In our experiments, we set $p = 2$.

Table F.1 presents the mean and standard deviation of the adversary's win rate, measuring how often it successfully reaches the target. In this setting, equilibrium is achieved when both teams win with equal probability across multiple instances. Given the *cooperative* nature of the game, the baseline performs relatively well, with EG-MADDPG achieving similar performance. However, both LA-MADDPG and LA-EG-MADDPG outperform their respective base optimizers (MADDPG and EG-MADDPG), demonstrating improved stability and effectiveness.

## F.2    ROCK–PAPER–SCISSORS: BUFFER STRUCTURE

For the Rock–paper–scissors (RPS) game, using a buffer size of 1M wasn't sufficient to store all experiences from the 60K training episodes. We observed a change in algorithm behavior around 40K episodes. To explore the impact of buffer configurations, we experimented with different sizes and structures, as experience storage plays a critical role in multi-agent reinforcement learning.

**Full buffer.** The buffer is configured to store all experiences from the beginning to the end of training without any loss.

**Buffer clearing.** In this setup, a smaller buffer is used, and once full, the buffer is cleared completely, and new experiences are stored from the start.

**Buffer shifting.** Similar to the small buffer setup, but once full, old experiences are replaced by new ones in a first-in-first-out (FIFO) manner.

**Results.** Figure 7 depicts the results when using different buffer options for the RPS game.

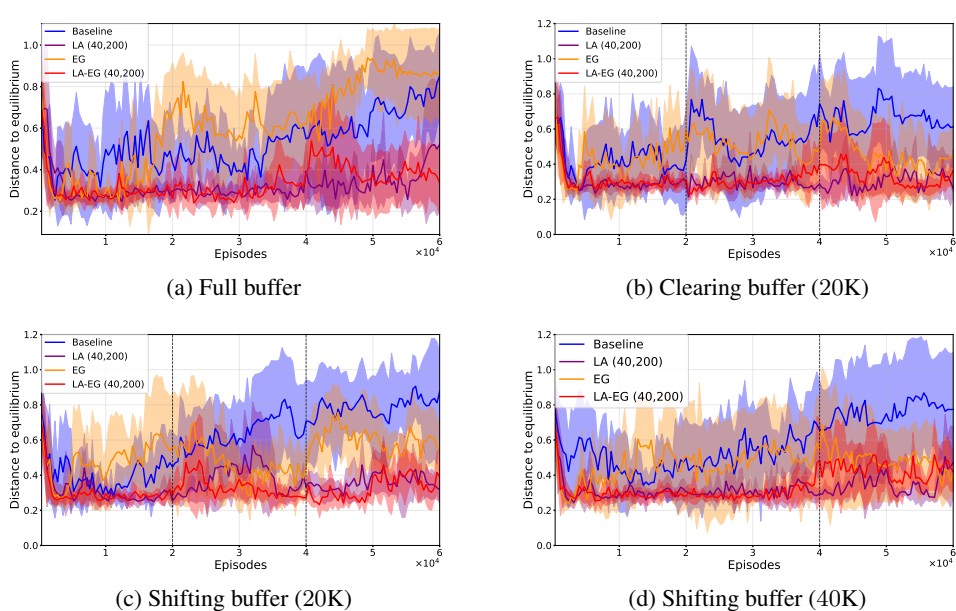

(a) Full buffer

(b) Clearing buffer (20K)

(c) Shifting buffer (20K)

(d) Shifting buffer (40K)

Figure 7: **Comparison of different buffer configurations (see Appendix F.2) and methods on Rock–paper–scissors game.** $x$-axis: training episodes. $y$-axis: 5-seed average norm between the two players' policies and equilibrium policy $(\frac{1}{3}, \frac{1}{3}, \frac{1}{3})^2$. The dotted line indicates the point at which the buffer begins to change, either through shifting or clearing.

## F.3 ROCK–PAPER–SCISSORS: SCHEDULED LEARNING RATE

We experimented with gradually decreasing the learning rate (LR) during training to see if it would aid convergence to the optimal policy in RPS. While this approach reduced noise in the results, it also led to increased variance across all methods except for LA-MADDPG.

Figure 8 depicts the average distance to the equilibrium policy over 5 different seeds for each methods, using periodically decreased step sizes.

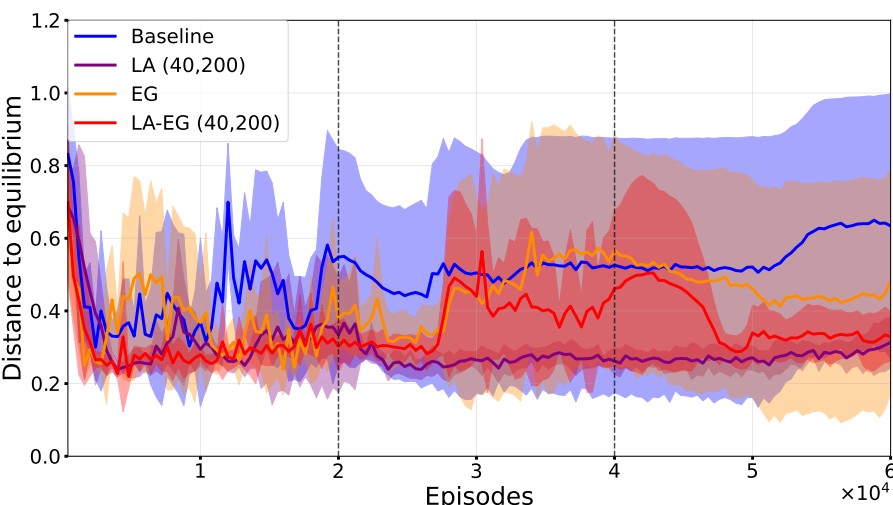

Figure 8: **Compares MADDPG with different *LA-MADDPG* configurations to the baseline MADDPG with (*Adam*) in Rock–paper–scissors with a scheduled learning rate**. $x$-axis: training episodes. $y$-axis: 5-seed average norm between the two players' policies and equilibrium policy $(\frac{1}{3}, \frac{1}{3}, \frac{1}{3})^2$. The dotted lines depict the times when the learning rate was decreased by a factor of 10.

## F.4 MPE: PREDATOR-PREY FULL RESULTS

We also evaluated the trained models of all methods on an instance of the environment that runs for 50 steps to compare learned policies. We present snapshots from it in Figure 10. Here, you can clearly anticipate the difference between the policies from baseline and our optimization methods. As in the baseline, only one agent will chase at the beginning of episode. Moreover, for the baseline (topmost row), the agents move further away from the landmarks and the good agent, which is suboptimal. This can be noticed from the decreasing agents' size in the figures. While in ours, both adversary agents engage in chasing the good agent until the end.

## F.5 MPE: PREDATOR-PREY AND PHYSICAL DECEPTION TRAINING FIGURES

In figures 11a and 11b we include the rewards achieved during the training of GD-MADDPG and LA-MADDPG resp. for MPE: Predator-prey. The figures show individual rewards for the agent (prey) and one adversary (predator). Blue and green show the individual rewards received at each episode while the orange and red lines are the respective running averages with window size of 100 of those rewards.

Figures 12a and 12b demonstrate same results but for MPE: Physical deception. In this game, We have two good agents, 'Agent 0 and 1' but since they are both receive same rewards, we only show agent 0.

## F.6 ADDITIONAL METRICS

For completeness, we include the NashConv and Exploitability results for Rock-paper-scissors (RPS). NashConv and Exploitability are commonly used in game-theoretic papers to measure the incentive of players to deviate quantized by the gain they get compared to best-response utility. In the optimal

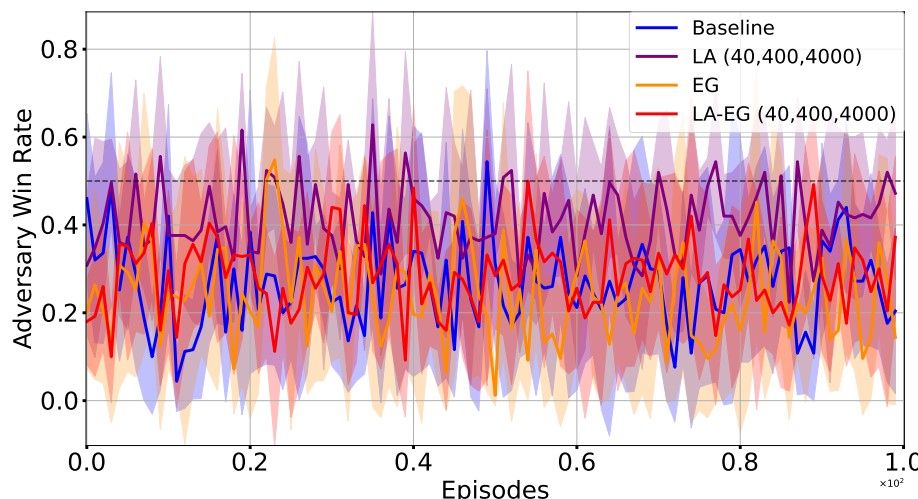

Figure 9: **Comparison on the MPE–Predator-prey game between the *GD-MADDPG*, *LA-MADDPG*, *EG-MADDPG* and *LA-EG-MADDPG* optimization methods, denoted as *Baseline*, *LA*, *EG*, *LA-EG*, resp.** $x$-axis: evaluation episodes. $y$-axis: mean adversaries win rate, averaged over 5 runs with different seeds.

case, both measures are equal to zero and closer to zero means less ability to deviate and less chance of being exploited.

In figure 13 we can see the results for using GD vs. LA in RPS, we can clearly see that LA has much lower values for the metrics compared to the baseline. Which confirms that it finds better policies.

## F.7    VARIANCE OF OUTERMOST LOOKAHEAD LEVEL

We also ran an additional experiment with longer episodes, where we evaluated and recorded the distance to equilibrium only after the outermost lookahead step, depicted in Figure 14. In these runs, the variance across seeds is noticeably smaller and the trajectories are more stable (compared to Figure 3a, further supporting the claim that Lookahead stabilizes training. This also reinforces the design choice in our pseudocode and in (Chavdarova et al., 2021) to always use the iterate obtained immediately after the outermost lookahead level (Algorithm 1 & 3).

## F.8    ABLATION: NUMBER OF SEEDS

In Figure 15, we compare the mean and variance of the distance to equilibrium in Rock–paper–scissors for *LA-MADDPG* trained with 5 versus 10 runs. The curves closely overlap, indicating that the results are consistent and robust to the number of seeds.

## F.9    ON THE REWARDS AS CONVERGENCE METRIC

Based on our experiments and findings from the multi-agent literature (Bowling, 2004), we observe that average rewards offer a weaker measure of convergence compared to policy convergence in multi-agent games. This implies that rewards can reach a target value even when the underlying policy is suboptimal. For example, in the Rock–paper–scissors game, the Nash equilibrium policy leads to nearly equal wins for both players, resulting in a total reward of zero. However, this same reward can also be achieved if one player always wins while the other consistently loses, or if both players repeatedly select the same action, leading to a tie. As such, relying solely on rewards during training can be misleading.

Figure 16 (top row) depicts a case with the baseline where, despite rewards converging during training, the agents ultimately learned to play the same action repeatedly, resulting in ties. Although this matched the expected reward, it falls far short of equilibrium and leaves the agents vulnerable to

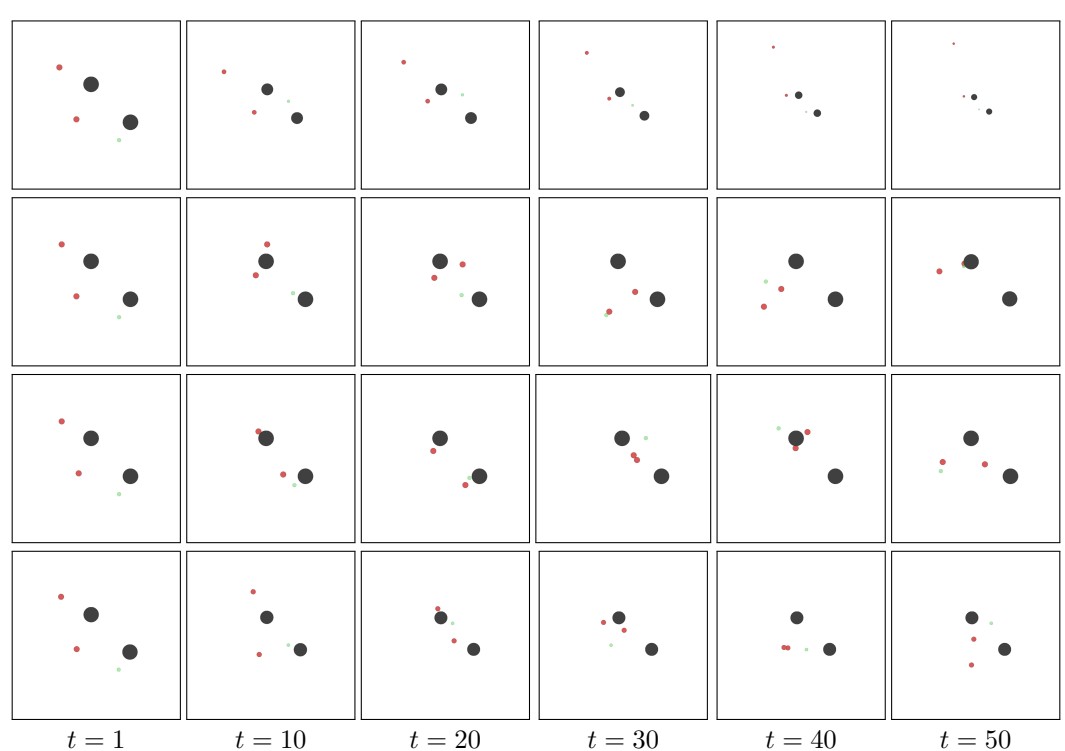

$t = 1$    $t = 10$    $t = 20$    $t = 30$    $t = 40$    $t = 50$

Figure 10: **Agents' trajectories of fully trained models with all considered optimization methods on the same environment seed of MPE: Predator-prey.** Snapshots show the progress of agents as time progresses in a 50 steps long environment. Each row contains snapshots of one method, from top to bottom: *GD-MADDPG*, *LA-MADDPG*, *EG-MADDPG* and *LA-EG-MADDPG*. Big dark circles represent landmarks, small red circles are adversary agents and green one is the good agent.

.

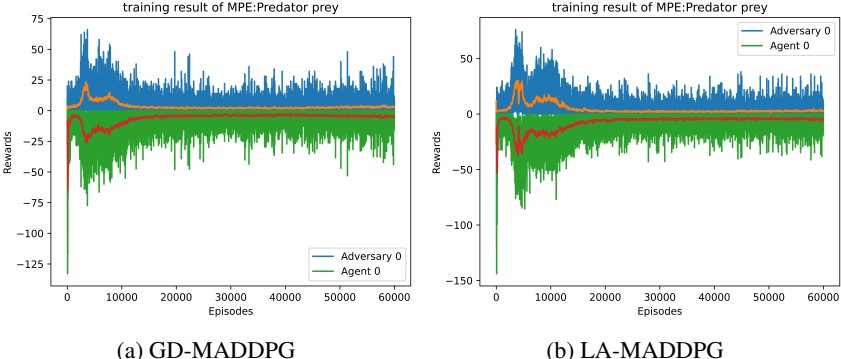

(a) GD-MADDPG           (b) LA-MADDPG

Figure 11: **The figure shows the learning curves during training of GD-MADDPG and LA-MADDPG for MPE: Predator-Prey.** $x$-axis: training episodes. $y$-axis: agents' rewards and their moving average with a window size of 100, calculated over 5-seeds over 5 seeds.

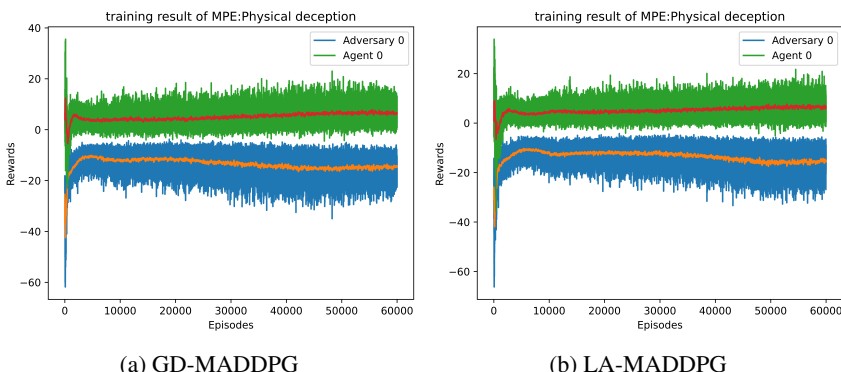

(a) GD-MADDPG                    (b) LA-MADDPG

Figure 12: **The figure shows the learning curves during training of GD-MADDPG and LA-MADDPG for MPE: Physical deception.** $x$-axis: training episodes. $y$-axis: agents' rewards and their moving average with a window size of 100, calculated over 5-seeds over 5 seeds.

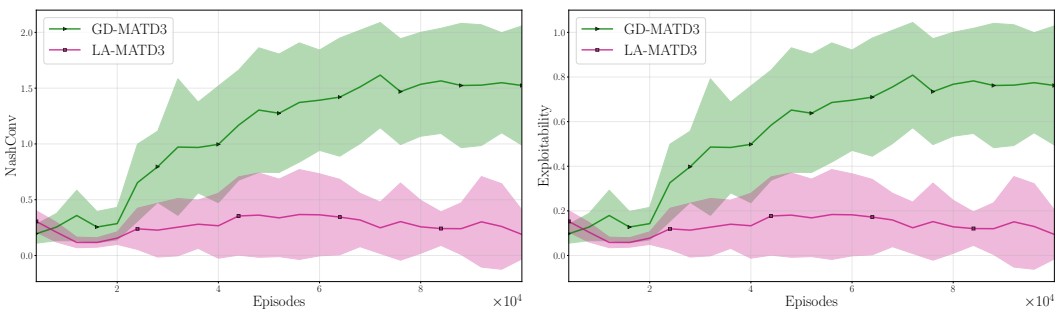

Figure 13: **NashConv (left) and exploitability (right) during training episodes of MATD3 on Rock–Paper–Scissors for GD-MATD3 and LA-MATD3.** Curves show the mean over 10 random seeds, and shaded regions indicate one standard deviation across seeds.

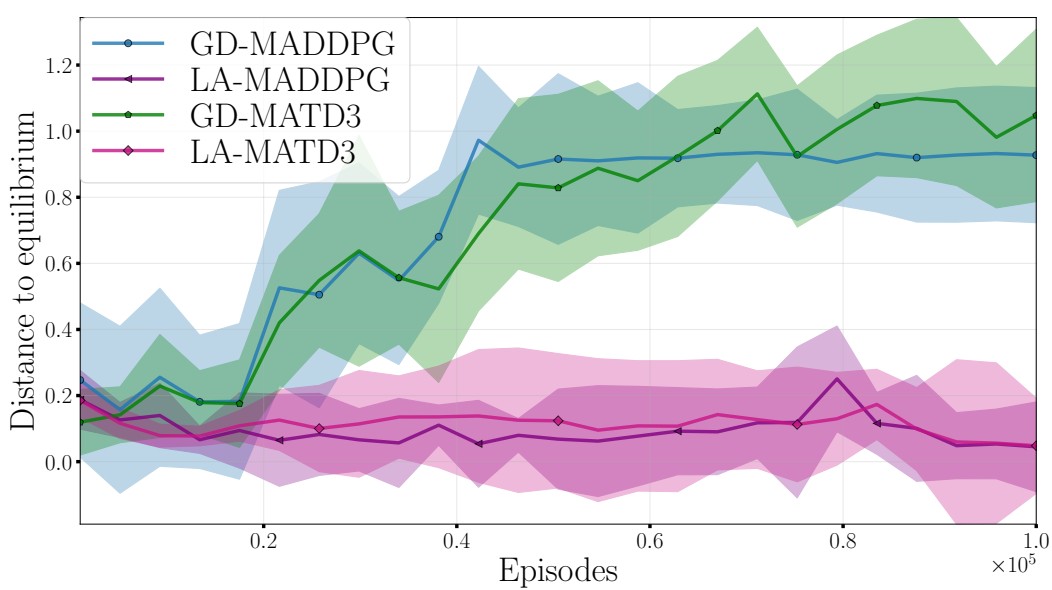

Figure 14: **Comparison of *GD-(MADDPG/MATD3)* and *LA-(MADDPG/MATD3)* on Rock–paper–scissors.** The $x$-axis shows training episodes, and the $y$-axis shows the distance between the agents' policies and the equilibrium policy. Results are measured after the outermost lookahead level. Curves are averaged over 10 random seeds, and shaded regions indicate $\pm 1$ standard deviation across these 10 seeds.

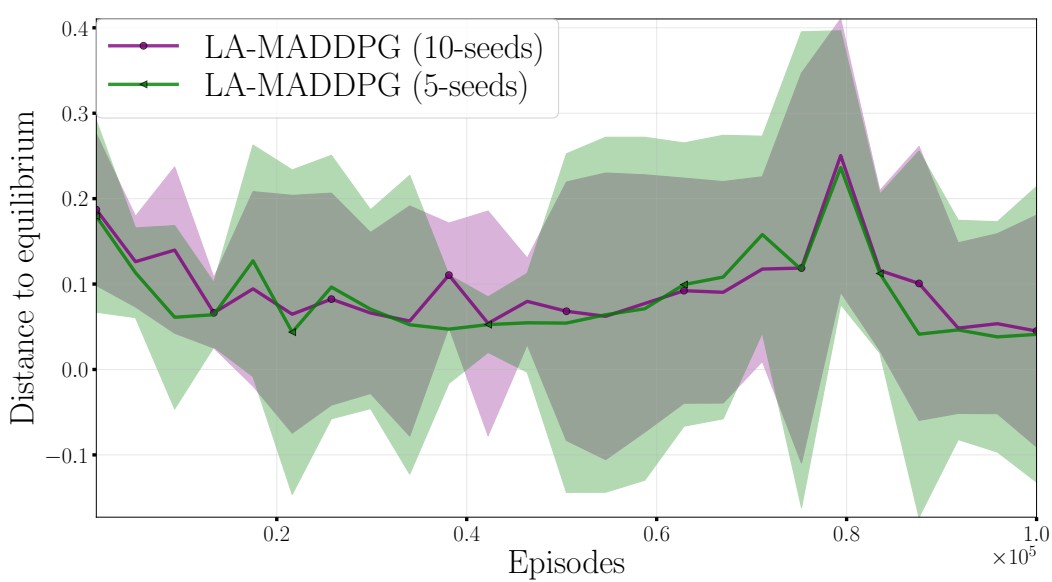

Figure 15: **Effect of the number of random seeds on *LA-MADDPG* in Rock–Paper–Scissors.** The $x$-axis shows training episodes, and the $y$-axis shows the distance between the learned joint policy and the Nash equilibrium. The two curves correspond to *LA-MADDPG* trained with 10 and 5 random seeds, respectively, illustrating the consistency of the method under different levels of averaging across seeds.

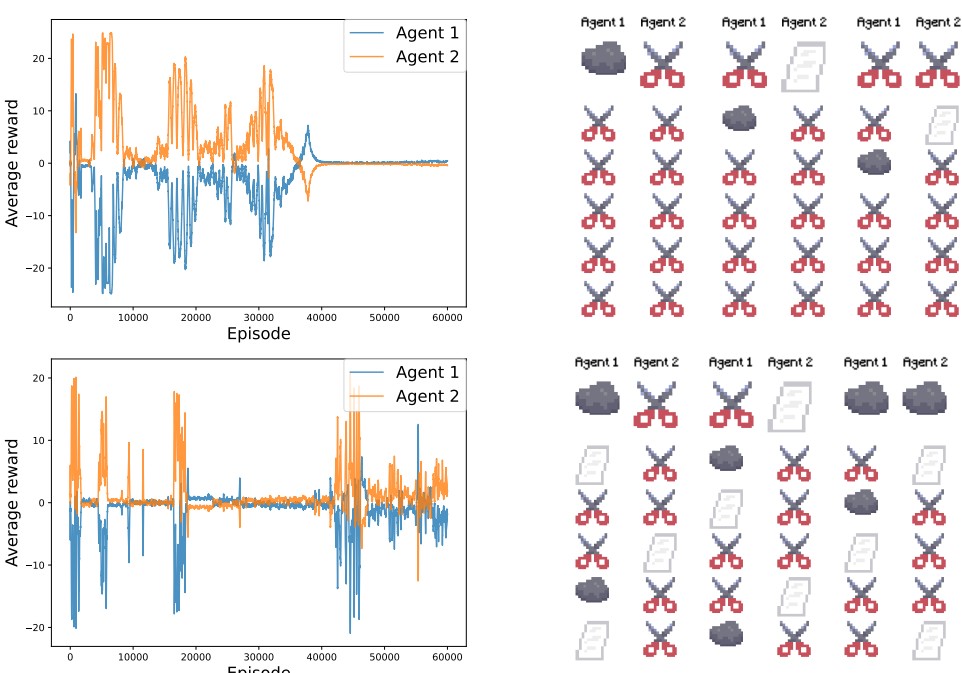

Figure 16: **Detailed Figure 5.** Saturating rewards (left) versus actions of the learned policies at the end (right) in the Rock–paper–scissors game. **Top row:** *GD-MADDPG*; **bottom row:** *LA-MADDPG*. In the left column, blue and orange show the running average of rewards through a window of 100 episodes. In the right column, we depict actions from the respective learned policies evaluation after training is completed, where each row represents what actions players have chosen in one step of the episode. Saturating rewards do not imply good performance, as evidenced by the top row; refer to Section 5.2 for discussion.

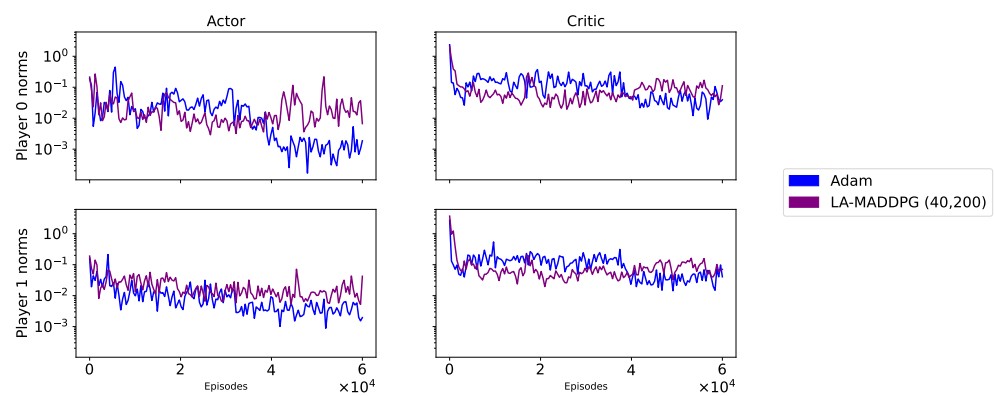

Figure 17: **Gradient norms across training in the *Rock–paper–scissors* game.**

exploitation by more skilled opponents. In contrast, the same figure shows results from LA-MADDPG under the same experimental conditions. Notably, while the rewards did not fully converge, the agents learned a near-optimal policy during evaluation, alternating between all three actions as expected. These results also align with the findings shown in Figure 3a.

We explored the use of gradient norms as a potential metric in these scenarios but found them to be of limited utility, as they provided no clear indication of convergence for either method. We include those results in Figure 17, where we compare the gradient norms of Adam and LA across the networks of different players.

This work highlights the need for more robust evaluation metrics in multi-agent reinforcement learning, a point also emphasized in (Lanctot et al., 2023), as reward-based metrics alone may be inadequate, particularly in situations where the true equilibrium is unknown.

