# OpenReview forum: "Learning in Circles: Rotational Dynamics in Competitive Reinforcement Learning"
_ICLR.cc/2026/Conference — Submitted to ICLR 2026_

### Official Review · Reviewer_jb55 · 2025-10-19

**Soundness:** 3
**Presentation:** 4
**Contribution:** 3
**Rating:** 8
**Confidence:** 3

**Summary:**

The paper investigates fundamental instabilities in competitive and multi-agent reinforcement learning (MARL), showing that when agents learn jointly (for example via actor–critic methods), the optimization dynamics are more properly viewed not as a minimization problem but as an equilibrium-seeking problem. In particular, the authors argue that the learning dynamics exhibit rotational flows (i.e., cycling around equilibria) rather than simple descent, which partly explains the observed instability or non-convergence in MARL. To address this, they recast the problem in the framework of Variational Inequality (VI) and draw on operator-theoretic methods from VI theory. Concretely, they propose using the Lookahead method (originally developed for VIs) to suppress rotational dynamics and integrate it into MARL, yielding the algorithm they call LA‑(MA)RL (Lookahead Multi-Agent Reinforcement Learning). Empirical results on classical two-player games (e.g., Rock-Paper-Scissors, Matching Pennies) and multi-agent benchmarks (e.g., Multi-Agent Particle Environments) show that their method improves stability and convergence toward equilibrium strategies compared to standard gradient-based MARL methods. Overall, the contribution lies in: (1) highlighting the rotational component of MARL dynamics, (2) connecting MARL with VI theory, and (3) proposing a practical algorithmic remedy through a Lookahead approach.

**Strengths:**

I find this paper very well-motivated and insightful. It builds a clear bridge between variational inequality (VI) theory and multi-agent reinforcement learning (MARL), showing that many of the stability issues in competitive learning come from the rotational nature of the underlying optimization field. This perspective feels both intuitive and mathematically grounded.

What I especially like is that the proposed solution — introducing Lookahead and Extragradient updates — is simple yet principled. It doesn’t require redesigning the RL algorithm, only rethinking how gradient steps are taken, and the resulting stability improvements are convincing.

Finally, the paper is very well presented: the visuals clearly illustrate the rotational dynamics, and the theory connects nicely to the experiments. Overall, it’s a rare example of a paper that deepens our conceptual understanding of MARL while also offering a practical fix.

**Weaknesses:**

While the paper is strong conceptually, its experimental scope is somewhat limited. Most results focus on small-scale or two-player settings, and it’s not entirely clear how well the proposed approach scales to more complex multi-agent or high-dimensional RL environments. The theoretical discussion of rotational dynamics is compelling, but it would be even stronger if the authors provided quantitative analyses or ablations showing how Lookahead affects convergence rates or equilibrium quality in larger systems.

**Questions:**

- The paper shows clear benefits in two-player and small-scale MARL settings. Do you anticipate any challenges or modifications needed for scaling the VI-based Lookahead approach to environments with many agents or continuous control tasks?

- The analysis decomposes learning dynamics into symmetric and antisymmetric components. Could the authors elaborate on whether this decomposition can inform adaptive algorithms — for example, automatically adjusting the update step or damping based on the observed rotational strength?

---

> ### Author Response · Authors · 2025-11-27
>
> Thank you very much for the thoughtful and positive review, and for engaging with both the conceptual and practical aspects of the paper on a deep level.
>
> **On scaling Lookahead to many agents and continuous control.**
>
> Indeed, there is no inherent barrier to scaling the VI-based Lookahead operator to many-agent or continuous-action settings. Because Lookahead acts purely at the optimizer level, it remains agnostic to the underlying policy/critic architectures. In practice, transitioning from standard gradient updates to Lookahead mostly involves:
> - keeping an additional parameter snapshot per agent and
> - performing one extra averaging step between “fast’’ and “slow’’ parameters.
>
> Importantly, no additional gradient evaluations are introduced beyond those required by the base optimizer — so the computational order remains unchanged. As you noted, standard methods tend to degrade as competitive pressure rises due to rotational dynamics in parameter space; Lookahead directly suppresses precisely these rotations when applied jointly across agents, as in our pseudocode.
>
> The same applies to continuous control: Lookahead sees only the gradient field in parameter space, so it is indifferent to whether actions are discrete or continuous. In practice, we expect that larger, high-dimensional environments may require some additional care in tuning the Lookahead depth and step sizes. Overall, we expect the extension to many agents and continuous control to be straightforward, we appreciate the opportunity to clarify.
>
> **On using the symmetric/antisymmetric decomposition for adaptive methods.**
>
> We were glad to see your interest in this point: it’s one of the more forward-looking contributions of the paper. As you inferred, the decomposition suggests natural directions for adaptivity: the antisymmetric component captures rotational strength, while the symmetric component reflects contraction. Monitoring quantities tied to the antisymmetric component (e.g., norms or dominant complex eigenvalues) could inform when to:
> - increase damping or Lookahead nesting under strong rotations, or
> - relax such corrections when dynamics become more potential-like.
>
> The difficulty, as you might have anticipated, is computational: tracking Jacobian structure in high-dimensional RL is expensive, and even approximations must handle a moving operator as policies evolve. Any adaptive scheme would therefore need lightweight proxies rather than full second-order information. Still, your intuition is right: Jacobian-aware but tractable adaptivity is a promising next step that builds directly on the operator-theoretic view we presented.
>
> Thank you again for your review, the accurate summary of the paper’s strengths, and the insightful questions. We’re glad the conceptual framing and practical impact resonated, and we appreciate the time you put into the assessment.

---

### Official Review · Reviewer_q2DT · 2025-10-29

**Soundness:** 3
**Presentation:** 3
**Contribution:** 3
**Rating:** 4
**Confidence:** 3

**Summary:**

This work presents an operator-theoretic viewpoint on actor-critic learning in MARL to show that AC methods exhibit rotational dynamics during learning, resulting in cycling around equilibria. They use a VI framework to study MARL as an equilibrium-seeking problem, and adopt the lookahead method for VIs, resulting in an algorithm which mitigates rotational dynamics. They test this method on some classical MULTI AGENT benchmarks.

**Strengths:**

1. The work is well-motivated. Instability is a well-known but understudied phenomenon in MARL, and the VI perspective sheds a light most MARL practitioners have probably not seen before. The work also presents a good overview of current literature with respect to VIs in the appendix.
2. The proofs of off-policy AC methods as VIs appears correct in Lemma 1 and Appendix C (Have not checked Appendix D).
3. The results for LAMARL applied to MADDPG and MATD3 are convincing, especially RPS and MP.

**Weaknesses:**

1. What are the error bars in Figures 3 and 4? This should be clearly noted such that the statistical significance of these results can be understood.
2. Why were more off-policy baselines not compared against? The Soft Actor Critic is generally considered a superior OP-AC baselines to the ones presented.
3. The benchmarks are relevant, but in my opinion not sufficient. MPE and classical games are necessary to indicate the mitigation of rotational dynamics, but what about more complex multi-agent benchmarks such as SMAC, Mamujoco, of Overcooked?

**Questions:**

1. The authors claim that an extension to on-policy methods is simple. Why was this not tried? Can the authors provide more insights, perhaps through an ablation, that is indeed feasible?

---

> ### Author Response · Authors · 2025-11-27
>
> Thank you very much for the careful and constructive review. We appreciate the time you took to examine the paper in detail.
>
> **Error bars in Figures 3 and 4.**
> The shaded regions in Figures 3 and 4 represent one standard deviation across the 5 random seeds used in each experiment. We have updated the figure captions in the revised version to state this explicitly. We appreciate the reviewer noting that this clarification was needed.
>
>
> **Choice of off-policy baselines (and SAC).**
> Our empirical focus is to isolate and study rotational dynamics arising in competitive multi-agent actor–critic training, and to evaluate whether VI-based optimization mitigates these effects while keeping the underlying algorithms unchanged. Thus, we selected MADDPG and MATD3 [1,2], which:
>
> * are designed for mixed competitive settings with centralized critics, and
>
> * are known to exhibit pronounced instability and cycling—making them a natural and informative testbed for an optimizer-centric study.
>
> Including SAC is indeed an interesting direction, but it is beyond the scope of the current paper.
>
>
> **Benchmark choice and absence of larger MARL domains (SMAC, Mamujoco, Overcooked).**
> The paper specifically targets instability driven by *competitive* interactions, where rotational dynamics are the most prominent. Classical matrix games and competitive MPE provide well-understood, controlled settings that allow us to:
>
> * connect empirical behavior directly to the VI-based analysis (e.g., through known equilibria), and
>
> * visualize and quantify rotational phenomena without confounding factors.
>
> Many popular benchmarks—SMAC, Mamujoco, Overcooked—are primarily cooperative, and competitive variants are not standardized. Constructing and validating competitive reward structures for these tasks is itself non-trivial and would constitute a separate empirical contribution.
>
> We fully agree that extending our approach to richer and more realistic competitive domains is an important direction, and we plan to explore this in future work. The current experiments provide a first controlled step that cleanly validates the operator-theoretic analysis and the impact of game optimizers on rotational dynamics.
>
> **On extending the method to on-policy algorithms.**
>
> Our statement that extending Lookahead-(MA)RL to on-policy methods is “simple” was meant at the operator level: the Lookahead update depends on the structure of the joint operator, not on whether it is estimated on-policy or off-policy. Switching to on-policy actor–critic changes the estimator, not the fixed point or the Lookahead operator.
> In practice, one can incorporate on-policy policy-gradient and value updates into the same Lookahead template with minimal structural modification.
>
> In this submission, however, we focused on off-policy methods for relevance: centralized off-policy critics (MADDPG/MATD3) are widely used in competitive MARL and are particularly susceptible to the instabilities studied here.
>
>
> Thank you again for the thoughtful feedback and for highlighting points that strengthened the clarity and positioning of the paper. We would be glad to continue the discussion should any additional clarification be useful.
>
>
> ----
> [1] Johann J.H. Ackermann, Volker Gabler, Takayuki Osa, and Masashi Sugiyama. Reducing overesti-
> mation bias in multi-agent domains using double centralized critics. ArXiv:1910.01465, 2019.
> [2] Ryan Lowe, Yi Wu, Aviv Tamar, Jean Harb, Pieter Abbeel, and Igor Mordatch. Multi-agent actor-
> critic for mixed cooperative-competitive environments. NIPS, 2017.

---

### Official Review · Reviewer_x4NB · 2025-10-31

**Soundness:** 2
**Presentation:** 1
**Contribution:** 1
**Rating:** 2
**Confidence:** 4

**Summary:**

This paper discusses actor-critic methods in single and multi-agent RL settings. The work adopts the variational inequality viewpoint formulation of the problems to adapt the lookahead VI algorithm and actor-critic methods to the Markov game setting and mitigate cycling behavior due to rotational dynamics. Experiments on rock-paper-scissors, Matching pennies and Multi-Agent Particle environments illustrate the performance of the proposed methods in mitigating cycling behavior.

**Strengths:**

1. The paper is well-organized.
2. The VI viewpoint is interesting to be further explored in MARL.

**Weaknesses:**

Overall this paper does not provide any substantially new contributions for the following reasons:

1. **VI formulation.** This formulation is well-known in game theory. It has also been exploited in Markov games (see e.g. Giannou et al. 2022).

2. **Algorithms using VI formulation for Markov games.** The algorithms for VIs to address divergence issues of gradient dynamics are well known and well studied. They have also been introduced and studied in the context of Markov games, see e.g. the paper below among others.

Anagnostides, I., Panageas, I., Farina, G., & Sandholm, T. (2024, March). Optimistic policy gradient in multi-player markov games with a single controller: Convergence beyond the minty property. In Proceedings of the AAAI Conference on Artificial Intelligence (Vol. 38, No. 9, pp. 9451-9459).

3. **Theory.** Beyond the formulation which is not genuinely new, the paper does not provide any theoretical convergence guarantees for the proposed algorithms (unlike some existing works, see examples above).

4. **Experiments.** Simulations are also very limited in scope. Given that there are new substantially new theoretical contributions, one would expect a more comprehensive evaluation of the proposed algorithms on more challenging benchmarks to support the proposed algorithms:
- Simulations are conducted for small scale toy problems such as Rock-Paper-Scissors and Matching Pennies, MPE is also small scale.
- The considered settings do not even seem to include state transitions, while the title and the preliminaries and notation insist on the RL/MARL setting. The evaluation is mostly on matrix games.
- It is not clear why we need AC methods in these settings, especially the simple ones.
- It is well known that algorithms such as extragradient and lookahead mitigate the divergence behaviors of gradient descent dynamics. The experiments provide little additional insights.

5. **Clarity.** The presentation lacks clarity in many parts of the paper and this makes reading the paper difficult, writing needs substantial improvements:
- What is exactly setting 1? Is the paper back to single agent RL after introducing Markov games in section 2? The setting and the operators are not explained, I guess some TD learning like algorithms are considered.
- Actor-critic convergence and convergence in mixing competitive 2 player zero-sum RL involve two separate set of theoretical questions as the rotational dynamics already show up for matrix games (independently of the dynamic RL setting). The game formulation for actor-critic methods (in single-agent RL) is unclear. If there is no game, what do you mean exactly by rotational game-like dynamics. I think the paper would benefit from stating the policy optimization problem and the Markov game problems separately and then introduce actor-critic methods for each one of them separately, then analyse the convergence behavior of AC methods for each one of the settings.
-  l. 239: ‘The above lemma shows setting 1 has game structure’: what is the considered game here? Actor-critic methods are used to solve policy optimization problems which are optimization problems, once can see them as solving a bilevel optimization formulation where an actor optimizes the expected return whereas a critic minimizes a policy evaluation error. It is unclear what the paper means by game here. Appendix C looks a little clearer but the main part requires entire rewriting in my opinion (sections 3, 4) and new developments.
-  In Markov games there is no observation space (l. 138-140) in the classical setting unless you consider a partially observable setting which does not seem to be the case in the rest of the paper.
- The introduction is too descriptive and does not sufficiently emphasize the gaps in the literature that the paper is addressing.

**Questions:**

- Can you clarify the novelty with respect to prior work?
- Can you clarify what is game setting considered in section 3 for single-agent RL?

---

> ### Author Response · Authors · 2025-11-27
> **addressing questions part (1/2)**
>
> Thank you for the detailed and critical review. Your comments helped us make the paper’s contributions more explicit and strengthen the positioning of the work.
>
> ### **1. Overall response**
>
> While the generality of the VI framework is well known, **the gap between this theory and the way standard deep actor–critic methods are actually used in RL/MARL practice remains surprisingly wide.** Classical VI results are rarely connected to the behavior of commonly used algorithms such as DDPG/MADDPG, MATD3, and other off-policy actor–critic variants. Conversely, VI-based theoretical works typically **assume settings or operators that do not align with how modern RL is implemented**, making it difficult for practitioners to translate those insights into stable training procedures.
>
> At the same time, multi-agent RL continues to suffer from well-documented instability and reproducibility issues, even on small benchmarks. This suggests that the existing theory—while elegant—**has not yet translated into practical guidance for commonly used algorithms.**
>
> The goal of our paper is therefore to explicitly **bridge this gap.** Rather than proposing yet another new MARL/VI algorithm, we intentionally chose to:
>
> * **derive the exact operator** corresponding to standard off-policy actor–critic (and its CTDE multi-agent form),
> * **analyze its Jacobian** to further explain why rotational dynamics arise even in what practitioners consider “standard RL,” and
> * **demonstrate experimentally** that VI-style lookahead optimizers can improve stability **without modifying the algorithms themselves.**
>
> This means the novelty is not in inventing new VI theory, but in **making the VI viewpoint operational for the RL community** (as used, without stringent simplifications), with a clean formulation that clarifies when and why rotations arise, how RL differs from classical VI settings, and how Lookahead and other VI methods act directly on the induced operator.
>
> In this sense, the contribution trades “novelty” in the sense of introducing a new algorithmic family for **novelty in bridging two communities** whose tools have remained surprisingly disconnected.
>
> ---
>  ### **2. Clarifying the core contributions**
>
> We agree the original submission did not highlight these points clearly enough and that the VI connection of Markov games are well knwon, and we do not claim novelty for the abstract VI framework itself.
> Our contributions are instead:
>
> 1. **Operator-theoretic formulation of standard actor–critic (single-agent and CTDE multi-agent).**
>    We derive the specific joint operator and Jacobian for widely used MA/RL actor–critic methods and show that, even for these standard algorithms, the induced dynamics are generically *rotational* due to the non-symmetric structure of the Jacobian. This is not discussed in prior VI–Markov game papers, which focus on different operators and assumptions. We would like to clarify that this is non-trivial; and the theoretical setting required multiple iterations so that it captures the core issues of real world competitive RL and can be used in future work to build on.
>
> 2. **Demonstrating why RL settings differ from classical VI optimization.**
>    By deriving the operator explicitly, we show how the “loss landscape” itself changes at every step due to policy-dependent data, even in full-batch settings. This explains why direct transfer of VI results to RL has been difficult in practice.
>
> 3. **Practical integration of Lookahead as an optimizer for deep MARL methods.**
>    We propose (nested) Lookahead as a *drop-in optimizer* for MADDPG/MATD3 and show that it improves stability and last-iterate performance **without altering critics, policies, or objectives.**
>
> 4. **Empirical evidence isolating the effect of the optimizer.**
>    Using clean settings (matrix games + MPE), we isolate the optimizer effect without confounding architectural or algorithmic choices. This provides actionable insight for practitioners.

---

> ### Author Response · Authors · 2025-11-27
> **addressing questions part (2/2)**
>
> ### **3. Clarifications addressing the reviewer’s technical questions**
>
> **(i) Novelty relative to prior VI / MG works (e.g., Anagnostides et al. 2024).**
> Those works design new algorithms with guarantees under structural assumptions (single controller, equilibrium collapse, generalized Minty property)  in a more tabular/value-based setting. For instance, note that Anagnostides et al. 2024 discuss the need for specific product structure that is not always veasible in practice. We instead analyze **existing deep actor–critic** as used in practice and connect their practical instabilities to the operator’s rotational component, showing how VI-based optimizers directly address this.
> The goals and settings are fundamentally different but the work can be complementary. We further modified related work discussion (section A in Appendix) and added a section (VIs and Markov games)  to incorporate the work on VIs for Markov games and how those compare to our work.
>
> **(ii) Game setting in Section 3.**
> Section 3 considers a single-agent MDP with an off-policy deterministic actor–critic algorithm. The “game” there is not between environment agents, but between **actor and critic viewed as two players in parameter space** with operator:
>
> [
> F(w, \theta) = (\nabla_w L_w, -\nabla_\theta J_\theta)
> ]
>
> * the critic minimizes a Bellman-error loss in ( w ),
> * the actor maximizes return in ( \theta ),
>   and the joint operator ( F(w, \theta) ) has a saddle structure with rotational components.
>   We previously provided longer explanations in appendix but we agree clarity was lacking here and have revised the section accordingly.
>
> **(iii) Markov games and observation space**
>
> You are correct that in the *classical* Markov game definition, one typically writes only the state and joint action spaces. In modern MARL, especially in CTDE settings, it is standard to introduce an **observation function** (o_i = f_i(s)) when agents have partial views of the state (e.g., in MPE, agents see only relative positions of nearby entities) (see [1,2]). We followed this MARL convention:
>
> * In Sec. 2, we explicitly state that each agent receives an observation (o_i \in O_i) that is a function of the global state (s), with examples where (f_i) is the identity (fully observable) or a coordinate projection / more complex mapping (partially observable).
> * In our MPE experiments, this is precisely the case: agents observe only parts of the global state (e.g., local positions), consistent with the PettingZoo MPE specification.
>
>
> ---
>
> ### **4. Revisions made**
>
> To address the reviewer’s concerns, we revised the manuscript as follows:
>
> * **Introduction:** rewritten contributions to more explicitly state the literature gap we address and clarify the paper’s methodological contribution (connecting VI theory to practical deep MARL).
> * **Sections 3:** reorganized to clearly define the single-agent and multi-agent settings; and the precise meaning of the “game” in the single-agent case.
> * **Conclusion:** strengthened to emphasize the conceptual bridge and outline how this operator view can guide future adaptive or theoretically grounded algorithms.
> * **Extended Related Work:** added a section summarizing works on VIs and Markov games and how they differ from our work.
>
> We believe the revisions clarify the contribution and resolve the issues identified. Should further clarification be helpful, we are happy to provide it.
>
>
>  ---
>  [1] Johann J.H. Ackermann, Volker Gabler, Takayuki Osa, and Masashi Sugiyama. Reducing overesti-
> mation bias in multi-agent domains using double centralized critics. ArXiv:1910.01465, 2019.
> [2] Ryan Lowe, Yi Wu, Aviv Tamar, Jean Harb, Pieter Abbeel, and Igor Mordatch. Multi-agent actor-critic for mixed cooperative-competitive environments. NIPS, 2017.

---

### Official Review · Reviewer_Nu3Z · 2025-11-04

**Soundness:** 2
**Presentation:** 2
**Contribution:** 3
**Rating:** 4
**Confidence:** 3

**Summary:**

The pervasive instability in competitive actor–critic RL stems less from algorithm design and more from the optimization dynamics. From a variational inequality perspective, the authors show that both single-agent actor–critic and multi-agent systems are equilibrium-seeking games whose Jacobians naturally induce rotational dynamics near equilibria, which causes the cycling. Then the authors introduce Lookahead-(MA)RL, a computationally efficient VI-based optimizer applied directly in the joint parameter space. Experiments on matrix games and competitive MPE benchmarks demonstrate improved stability and convergence over GD-style baselines, without changing agents or objectives.

**Strengths:**

- This paper offers a clear framing by identifying rotational dynamics as the fundamental cause of instability in competitive RL and formalizing this through a variational inequality perspective.

- This paper is theoretically rigorous, establishing a direct link between rotations in actor–critic methods and the non-symmetric Jacobian / complex eigenpairs structure.

- This paper proposes Lookahead-based VI methods as a practical and computationally efficient solution that can scale across actor-critic variants, single- and multi-agent cases with both on- and off-policy implementations, demonstrating consistent empirical superiority and stability on multiple tasks.

**Weaknesses:**

- The baseline comparisons are limited, failing to include state-of-the-art MARL approaches.

- The environments are mostly toy games (RPS / MP / simple MPE), missing more complex and realistic MARL benchmarks (e.g., SMAC, football, ...).

- Missing ablation or sensitivity study for the key hyperparameters ($k^{(j)}$, $\alpha$ , ...).

**Questions:**

- Why aren't standard evaluation metrics like exploitability or NashConv included in experiment results?

- Why in MPE only 5 seeds are used when the environment is more complex?

---

> ### Author Response · Authors · 2025-11-26
>
> Thanks for the thoughtful review on both the theory and experiments.
>
> ## Questions:
> **On evaluation metrics (exploitability / NashConv)**.
>
> For the normal-form games (Rock–paper–scissors and Matching Pennies), there is a *single, known Nash equilibrium*. Hence, the distance of the agents’ policies to the equilibrium policy is a natural and easy-to-interpret metric: it directly measures how far the learning dynamics are from the target solution at each point in training. For these games, other common metrics, such as exploitability or NashConv, are transformations of the equilibrium distance, so we opted for it in the main text to keep the presentation focused.
>
> In response to your suggestion, we have added plots (Figure 13 in the appendix F6) for exploitability and NashConv for Rock–paper–scissors (RPS) and confirmed that they indicate same conclusions as our distance-to-equilibrium metric.
>
> **On the number of seeds in MPE (5 seeds).**
>
> As environment complexity increases (e.g., from matrix games to MPE), the computational cost per seed grows, so in deep RL/MARL it is common to use between 3 and 10 seeds; for MPE in particular, 5 seeds is both common practice (e.g., [1,2,3]) and identified as the typical choice in the commonly used protocols in [3]. We thus followed this benchmarking to balance statistical robustness and computational budget.
>
> Importantly, in our plots the confidence intervals are relatively tight and the performance gaps between Gradient Descent (GD) and Lookahead (LA) are already large and consistent across runs. Our additional results support this: Figure 15 (appendix) shows that LA on RPS with 5 vs. 10 seeds yields essentially identical error bands, and Figure 14 (appendix) (measuring distance after the outermost lookahead) shows even lower and more stable variance than Figure 1(a), further confirming the stabilizing effect of LA.
> In response to your concern, we will add MPE experiments with more seeds for the camera-ready version.
>
> Thanks again for careful reading, constructive questions, and your time.
>
> ----
> [1] Johann J.H. Ackermann, Volker Gabler, Takayuki Osa, and Masashi Sugiyama. Reducing overesti-
> mation bias in multi-agent domains using double centralized critics. ArXiv:1910.01465, 2019.
>
> [2] Ryan Lowe, Yi Wu, Aviv Tamar, Jean Harb, Pieter Abbeel, and Igor Mordatch. Multi-agent actor-
> critic for mixed cooperative-competitive environments. NIPS, 2017.
>
> [3] Rihab Gorsane, Omayma Mahjoub, Ruan John de Kock, Roland Dubb, Siddarth Singh, and Arnu Pretorius. Towards a standardised performance evaluation protocol for cooperative marl. Advances in Neural Information Processing Systems, 35:5510–5521, 2022.**

---

### Meta-Review · Area_Chair_YwDG · 2026-01-06

**Summary:**

This paper discusses actor-critic methods in single and multi-agent RL settings. The authors try to mitigate the cycling behaviors in games.
Reviewers are all challenging the scalability and practicality of this paper. Some reviewers have concerns about missing theoretical analysis or guarantees.

**Reviewer Concerns:**

The main outstanding concern is still the toyish test environments and limited baselines. While the authors show genuine efforts in solving it during rebuttal, the concerns still remain.

**Reviewer Scores:**

8 4 4 4

---

### Decision · Program_Chairs · 2026-01-26

Reject